# Robustness of Graph Neural Networks at Scale

**Simon Geisler, Tobias Schmidt, Hakan Şirin,**
**Daniel Zügner, Aleksandar Bojchevski, and Stephan Günnemann**
Department of Informatics
Technical University of Munich
{geisler, schmidt, sirin, zuegnerd, bojchevs, guennemann}@in.tum.de

## Abstract

Graph Neural Networks (GNNs) are increasingly important given their popularity and the diversity of applications. Yet, existing studies of their vulnerability to adversarial attacks rely on relatively small graphs. We address this gap and study how to attack and defend GNNs at scale. We propose two sparsity-aware first-order optimization attacks that maintain an efficient representation despite optimizing over a number of parameters which is quadratic in the number of nodes. We show that common surrogate losses are not well-suited for global attacks on GNNs. Our alternatives can double the attack strength. Moreover, to improve GNNs' reliability we design a robust aggregation function, Soft Median, resulting in an effective defense at all scales. We evaluate our attacks and defense with standard GNNs on graphs more than 100 times larger compared to previous work. We even scale one order of magnitude further by extending our techniques to a scalable GNN.

## 1 Introduction

The evidence that Graph Neural Networks (GNNs) are not robust to adversarial perturbations is compelling [14, 20, 50]. However, the graphs in previous robustness studies are tiny. This is worrying, given that GNNs are already deployed in many real-world Internet-scale applications [5, 44]. For example, PubMed [33] (19,717 nodes) is often considered to be a large-scale graph and around 20 GB of memory is required for an attack based on its dense adjacency matrix. Such memory requirements are impractical and limit advancements of the field. In this work, we set the foundation for the holistic study of adversarial robustness of GNNs at scale. We study graphs with up to 111 million nodes for local attacks (i.e. attacking a single node) and 2.5 million nodes for global attacks (i.e. attacking all nodes at once). As it turns out,

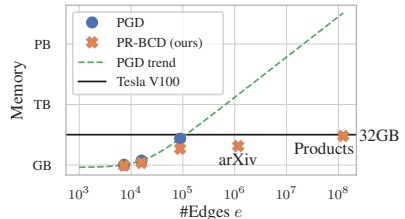

Figure 1: GPU memory consumption for a global attack with Projected Gradient Descent (PGD) [43], its quadratic extrapolation, and our *Projected Randomized Block Coordinate Descent (PR-BCD)* (§ 3). Both yield similar adversarial accuracy. Beyond attacks, our defense (§ 4) also scales to these graphs.

GNNs at scale are also highly vulnerable to adversarial attacks. In Fig. 1, we show the substantial improvement of memory efficiency of our attack over a popular prior work for attacking a GNN globally. Analogous to our attacks, our effective defense scales to graphs with 111 million nodes and beyond.

**Scope.** We focus on adversarial robustness w.r.t. structure attacks on GNNs for node classification

$$\max_{\tilde{\boldsymbol{A}} \text{ s.t. } \|\tilde{\boldsymbol{A}} - \boldsymbol{A}\|_0 < \Delta} \mathcal{L}(f_\theta(\tilde{\boldsymbol{A}}, \boldsymbol{X})) \tag{1}$$

with loss function $\mathcal{L}$ (or its surrogate $\mathcal{L}'$), budget $\Delta$, and *fixed* model parameters $\theta$. The GNN $f_\theta(\boldsymbol{A}, \boldsymbol{X})$ is applied to a graph $\mathcal{G} = (\boldsymbol{A}, \boldsymbol{X})$ with node attributes $\boldsymbol{X} \in \mathbb{R}^{n \times d}$, the adjacency matrix

35th Conference on Neural Information Processing Systems (NeurIPS 2021).

$\boldsymbol{A} \in \{0, 1\}^{n \times n}$, and $m$ edges. We focus on *evasion* (test time) attacks, but our methods can be used in *poisoning* (train time) attacks [49]. We distinguish between *local* attacks on a single node and *global* attacks that target a large fraction of nodes with a shared budget $\Delta$. We study white-box attacks since they have the most powerful threat model and can be used to understand the robustness w.r.t. "worst-case noise" of a model, as well as to assess the efficacy of defenses. For these reasons *white-box attacks are important and practical from the perspective of a defender*.

**Broader impact.** Since we enable the study of robustness at scale, which previously was practically infeasible, an adversary can potentially abuse our attacks. The risk is minimized given that we assume perfect knowledge about the graph, model, and labels. Nonetheless, our findings suggest that one should be careful when deploying GNNs, and highlight that further research is needed. To mitigate this risk, we must be able to evaluate it. We also propose a scalable defense that shows strong performance empirically but we urge practitioners to consider potential trade-offs, e.g. improving robustness at the expense of accuracy for different groups of users.

**Contributions.** We address three major challenges hindering the study of GNNs' adversarial robustness at scale and propose viable solutions including an extensive empirical evaluation: (1) Previous losses are not well-suited for global attacks on GNNs; (2) Attacks on GNNs scale quadratically in the number of nodes or worse; (3) Similarly, previous robust GNNs are typically not scalable.

**(1) Surrogate loss.** We study the limitation of state-of-the-art surrogate losses for attacking the accuracy of a GNN over all nodes [8, 9, 17, 27, 41, 43, 49] in § 2. Especially in combination with small/realistic budgets $\Delta$ and on large graphs, previous surrogate losses lead to weak attacks. In particular, Cross Entropy (CE) or the widely used Carlini-Wagner loss [6, 43] are weak surrogates for such global attacks. Our novel losses that overcome these limitations easily improve the strength of the attack by 100% on common datasets. For larger graphs, this gap even becomes more significant.

**(2) Attacks.** Attacks solving a discrete optimization problem easily become computationally infeasible because of the vast amount of potential adjacency matrices ($\mathcal{O}(2^{n^2})$). An approximate solution can be found with first-order optimization but we then still optimize over a quadratic number of parameters ($n^2$). There is no trivial way to sparsify existing attacks as we need to represent each edge explicitly to obtain its gradient (i.e. space complexity $\Theta(n^2)$). Nevertheless, we overcome this limitation and propose two strategies to apply first-order optimization without the burden of a dense adjacency matrix. In § 3, we describe how to add/remove edges between existing nodes based on Randomized Block Coordinate Descent (R-BCD) at an additional memory requirement of $\mathcal{O}(\Delta)$. Due to the limited scalability of traditional GNNs, we also consider the case where we attack PPRGo [5], a scalable GNN. Here, we even obtain an algorithm with constant complexity in the nodes $n$.

**(3) Defense.** We propose *Soft Median* in § 4 – a computationally efficient, robust, differentiable aggregation function inspired by Geisler et al. [18], by taking advantage of recent advancements in differentiable sorting [31]. Using our Soft Median we observe similar robustness to [18], but with a significantly lower memory footprint, which enables us to defend GNNs at scale.

## 2   Surrogate Losses for Global Attacks

During training and in first-order attacks, we ideally wish to optimize a target metric which is often discontinuous (e.g. accuracy and 0/1 loss $\mathcal{L}_{0/1}$). However, for gradient-based optimization we commonly substitute the actual target loss by a *surrogate* $\mathcal{L}' \approx \mathcal{L}$ (e.g. cross entropy for $\mathcal{L}_{0/1}$). In the context of i.i.d. samples (e.g. images), a single example is attacked in isolation with its own budget, which is similar to a *local* attack for GNNs. When a single node's prediction is attacked, it is often sufficient to *maximize* the cross entropy for the attacked node/image (untargeted attack): $\mathrm{CE}(y, \boldsymbol{p}) = -\log(p_{c^*})$. where $y$ is the label and $\boldsymbol{p}$ is the vector of confidence scores.

Many *global* attacks for GNNs [8, 41, 43, 49] maximize the average CE to attack all nodes with a combined budget $\Delta$. However, a loss like CE can be ineffective, particularly when the number of nodes is large in comparison to the budget $\Delta/n \to 0$. While experimenting on large graphs, we often observed that the CE loss increases even though the accuracy does not decline (see § B). As we can see in Fig. 2, this is due to CE's bias towards nodes that have a low confidence score. With CE and a sufficiently small budget $\Delta \ll n$ we primarily attack nodes that are already misclassified, which means that the classification margin $\psi = \min_{c \neq c^*} p_{c^*} - p_c$ is already negative.

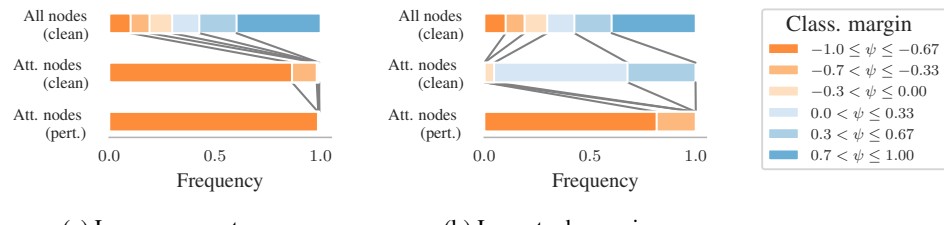

(a) Loss: cross entropy  (b) Loss: tanh margin

Figure 2: Margin $\psi$ of test nodes vs. attacked nodes, before ("clean") and after perturbation ("perturbed"). We attack the PubMed graph (Table 1) and a single-layer GCN with one percent of edges ($\epsilon = 0.01$) as a budget. In stark contrast to the tanh margin (b), the CE loss (a) spends a lot of its budget on misclassified nodes (i.e. $\psi < 0$). See § B for more variants and details.

**Global attack.** In contrast to attacking a single image/node, a global attack on a GNN has to (1) keep house with the budget $\Delta$ and (2) find edges that degrade the overall accuracy maximally (i.e. target "fragile" nodes). Without additional information, intuitively, one would first attack low-confidence nodes close to the decision boundary. Hence, the surrogate loss should have a minimal (maximally negative) gradient at $\psi \to 0^+$ (i.e. approaching $\psi \to 0$ from $\psi \geq 0$). Moreover, if we solely want to lower the accuracy, then we can stop attacking a node once it is misclassified[1]:

**Definition 1** *A surrogate loss $\mathcal{L}'$ for global attacks* **(I)** *should only incentivize perturbing nodes that are correctly classified:* $\partial\mathcal{L}'/\partial z_{c^*}|_{\psi<0} = 0$ *and* **(II)** *should favour nodes close to the decision boundary:* $\partial\mathcal{L}'/\partial z_{c^*}|_{\psi_0} < \partial\mathcal{L}'/\partial z_{c^*}|_{\psi_1}$ *for any* $0 < \psi_0 < \psi_1$.

Since Eq. 1 is in general a discrete and non-convex optimization problem that is often NP-complete [1, 3, 38, 45], we propose to study the surrogate loss under the subsequent simplifying assumptions. Note that in an actual attack other influences (e.g. node degree) are still considered while solving the optimization problem. *Assumption 1*: The set of attacked nodes is independent (their receptive fields do not overlap). Particularly on large graphs with small budgets, $\Delta/n \to 0$, deciding which node to attack becomes an increasingly local decision since the receptive field becomes insignificant in comparison to the rest of the graph. *Assumption 2*: The budget required to change the prediction of node $i$ depends (only) on the margin: $\Delta_i = g(|\psi_i|)$ for some increasing and non-negative function $g(\cdot)$. That is, the larger the margin $\psi_i$, the harder it is to attack node $i$. As stated in Proposition 1, with these assumptions, an optimizer with a surrogate loss compliant with Definition 1 is also optimizing the 0/1 loss $\mathcal{L}_{0/1}$ (for proof see § B.4).

**Proposition 1** *Let $\mathcal{L}'$ be the surrogate for the 0/1 loss $\mathcal{L}_{0/1}$ used to attack a node classification algorithm $f_\theta(\boldsymbol{A}, \boldsymbol{X})$ with a global budget $\Delta$. Suppose we greedily attack nodes in order of $\partial\mathcal{L}'/\partial z_{c^*}(\psi_0) \leq \partial\mathcal{L}'/\partial z_{c^*}(\psi_1) \leq \cdots \leq \partial\mathcal{L}'/\partial z_{c^*}(\psi_l)$ until the budget is exhausted $\Delta < \sum_{i=0}^{l+1} \Delta_i$. Under Assumptions 1 & 2, we then obtain the global optimum of $\max_{\tilde{\boldsymbol{A}} \text{ s.t. } \|\tilde{\boldsymbol{A}}-\boldsymbol{A}\|_0 < \Delta} \mathcal{L}_{0/1}(f_\theta(\tilde{\boldsymbol{A}}, \boldsymbol{X}))$ if $\mathcal{L}'$ has the properties* **(I)** $\partial\mathcal{L}'/\partial z_{c^*}|_{\psi<0} = 0$ *and* **(II)** $\partial\mathcal{L}'/\partial z_{c^*}|_{\psi_0} < \partial\mathcal{L}'/\partial z_{c^*}|_{\psi_1}$ *for any* $0 < \psi_0 < \psi_1$.

Even under the simplifying Assumptions 1 & 2, the Cross Entropy (CE) is not guaranteed to obtain the global optimum. The (CE) violates property (I) and in the worst case only perturbs nodes that are already misclassified (see Fig. 2). The Carlini-Wagner (CW) [6, 43] loss CW = $\min(\max_{c \neq c^*} z_c - z_{c^*}, 0)$ violates property (II). It is also not guaranteed to obtain the global optimum, i.e. CW loss lacks focus on nodes close to the decision boundary. In the worst case, an attack with CW spends all budget on confident nodes—without flipping a single one.

We propose the Masked Cross Entropy MCE = $1/|\mathbb{V}^+| \sum_{i \in \mathbb{V}^+} -\log(p_{c^*}^{(i)})$ which fulfills both properties by only considering correctly classified nodes $\mathbb{V}^+$ and, hence, reaches the global optimum under the stated assumptions. Empirically, for a greedy gradient-based attack the MCE comes with gains of more than 200% in strength (see Fig. 6). Surprisingly, if we apply MCE to a Projected Gradient Descent (PGD) attack, we observe hardly any improvement over CE. We identify two potential reasons for that. The first is due to the learning dynamics of PGD. Suppose a misclassified node does

---

[1]We simply write $\boldsymbol{p} = f_\theta(\boldsymbol{A}, \boldsymbol{X})$ omitting that $\boldsymbol{p}$ belongs to specific node, i.e. $\boldsymbol{p}_i$ of node $i$. We also overload this with the logits / pre-softmax activation as $\boldsymbol{z} = f_\theta(\boldsymbol{A}, \boldsymbol{X})$. See § A for an overview of the notation.

not receive any weight in the gradient update, now if the budget is exceeded after the update it is likely to be down-weighted. This can lead to nodes that oscillate around the decision boundary (for more details see § B). A similar behavior occurs for to the Carlini-Wagner loss in e.g. Fig. B.1 (e).

In Definition 2, we relax properties (I)/(II) and propose to overcome these limitations via enforcing confidently misclassified nodes, i.e. we want the attacked nodes to be at a "safe" distance from the decision boundary. We propose the tanh of the margin in logit space, i.e. tanh margin $= \tanh(\max_{c \neq c^*} z_c - z_{c^*})$. It obeys Definition 2 and its effectiveness is apparent from Fig. 2. For the empirical evaluation see § 5 and for more results as well as details on all selected losses see § B. In the appendix we also study further losses to deepen the understanding about the required properties. Additionally, in § B.5, we give an alternative Proposition 1 for a relaxed Assumption 2 s.t. $\mathbb{E}[\Delta_i | \psi_i] = g(|\psi_i|)$

**Definition 2** *A surrogate loss $\mathcal{L}'$ for global attacks that encourages confident misclassification* (**A**) *should saturate* $\lim_{\psi \to -1^+} \mathcal{L}' < \infty$ *and* (**B**) *should favor points close to the decision boundary:* $\partial \mathcal{L}'/\partial z_{c^*}|_{\psi_0} < \partial \mathcal{L}'/\partial z_{c^*}|_{\psi_1} < 0$ *for any* $0 < \psi_0 < \psi_1 < 1$ *or* $-1 < \psi_1 < \psi_0 < 0$.

## 3 Scalable Attacks

Beginning with [14, 50], many adversarial attacks on the graph structure have been proposed. As discussed, gradient-based attacks such as [9, 41, 49, 50] aim for lower computational cost by approximating the corresponding discrete optimization problem. However, they optimize all possible entries in the *dense* adjacency matrix $A$ which comes with quadratic space complexity $\Theta(n^2)$. Since previous attacks come with limited scalability (e.g. see Fig. 1), GNNs robustness on larger graphs is largely unexplored. First, we propose a family of attacks that does not require a dense adjacency matrix and comes with linear complexity w.r.t. the budget $\Delta$. Then, we further improve the complexity of our attack for a scalable GNN called PPRGo [5].

**Related work.** Li et al. [26] evaluate their *local* adversarial attack SGA only on a graph with around 200k nodes and SGA is specifically designed for Simplified Graph Convolution (SGC) [40]. Note that a two-layer SGC is identical to Nettack's surrogate model. We consider arbitrary Graph Neural Networks and we even scale our *global* attack to a graph ten times larger. With PPRGo we even outscale them by factor 500. Feng et al. [17] partition the graph to lower the attack's memory footprint but still have a time complexity of $\mathcal{O}(n^2)$. Dai et al. [14] scale their reinforcement learning approach to a graph for financial transactions with 2.5 million nodes. In contrast to our work, they scale their *local* attack only using a tiny budget $\Delta$ of *a single edge deletion* and only need to consider the receptive field of a single node. We scale our local attack to 111M nodes and allow large budgets $\Delta$.

**Large scale optimization.** In some big data use cases, the cost to calculate the gradient towards all variables can be prohibitively high. For this reason, coordinate descent has gained importance in machine learning and large-scale optimization [39]. Nesterov [29] proposed (and analyzed the convergence of) Randomized Block Coordinate Descent (R-BCD). In R-BCD only a subset (called a block) of variables is optimized at a time and, hence, only the gradients towards those variables are required. In many cases, this allows for a lower memory footprint and in some settings even converges faster than standard methods [30].

For clarity, we model the perturbations $P \in \{0, 1\}^{n \times n}$ explicitly ($P_{ij} = 1$ denotes an edge flip):

$$\max_{P \text{ s.t. } P \in \{0,1\}^{n \times n}, \ \sum P \leq \Delta} \mathcal{L}(f_\theta(A \oplus P, X)). \tag{2}$$

Here, $\oplus$ stands for an element-wise exclusive or and $\Delta$ denotes the edge budget (i.e. the number of altered entries in the perturbed adjacency matrix). Naively, applying R-BCD to optimize towards the dense adjacency matrix would only save some computation on obtaining the respective gradient. It still has a space complexity of $\mathcal{O}(n^2)$ on top of the complexity of the attacked model because we still have to store up to $n^2$ parameters. Note that the $L_0$ perturbation constraint with limited budget $\Delta$ implies that the solution will be sparse. We build upon this fact and in each epoch, in a survival-of-the-fittest manner, we keep that part of the search space which is "promising" and resample the rest. Despite the differences, we simply call our approach **Projected Randomized Block Coordinate Descent (PR-BCD)** and provide the pseudo code in Algo. 1 (a preliminary version appeared in [19]). On top of the GNN, PR-BCD comes with space complexity of $\Theta(b)$ where $b$ is the block size (number of coordinates) since *everything can be implemented efficiently with sparse operations*. We typically choose $\Delta$ to be a fraction of $m$ and $b > \Delta$, thus, in practice, we have a linear overhead.

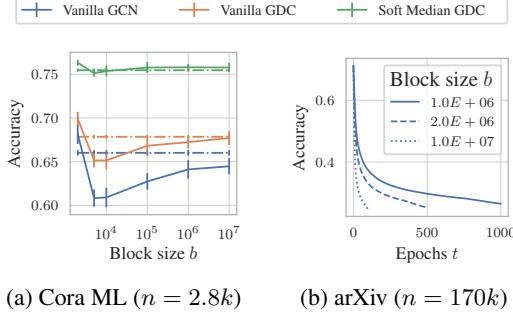

(a) Cora ML ($n = 2.8k$)  (b) arXiv ($n = 170k$)

Figure 3: Influence of block size $b$ on PR-BCD (dashed $L_0$ PGD [43]) with tanh margin loss and $\epsilon = 0.1$. (a) shows adv. accuracy with three-sigma error over five seeds. We resample $E_{\text{res.}} = 50$ epochs and then fine-tune 250. (b) shows adv. accuracy over epochs $t$ with $E \cdot b = \text{const.}$

---

**Algorithm 1** Projected Randomized Block Coordinate Descent (PR-BCD)

1: **Input:** Gr. $(\boldsymbol{A}, \boldsymbol{X})$, lab. $\boldsymbol{y}$, GNN $f_\theta(\cdot)$, loss $\mathcal{L}$
2: **Parameter:** budget $\Delta$, block size $b$, epochs $E$ & $E_{\text{res.}}$, heuristic $h(\dots)$, learning rate $\alpha_t$
3: Draw w/o replacement $\boldsymbol{i}_0 \in \{0, 1, \dots, n^2 - 1\}^b$
4: Initialize zeros for $\boldsymbol{p}_0 \in \mathbb{R}^b$
5: **for** $t \in \{1, 2, \dots, E\}$ **do**
6: $\quad \hat{\boldsymbol{y}} \leftarrow f_\theta(\boldsymbol{A} \oplus \boldsymbol{p}_{t-1}, \boldsymbol{X})$
7: $\quad \boldsymbol{p}_t \leftarrow \boldsymbol{p}_{t-1} + \alpha_t \nabla_{\boldsymbol{p}_{t-1}[\boldsymbol{i}_{t-1}]} \mathcal{L}(\hat{\boldsymbol{y}}, \boldsymbol{y})$
8: $\quad$ Projection $\boldsymbol{p}_t \leftarrow \Pi_{\mathbb{E}[\text{Bernoulli}(\boldsymbol{p}_t)] \leq \Delta}(\boldsymbol{p}_t)$
9: $\quad \boldsymbol{i}_t \leftarrow \boldsymbol{i}_{t-1}$
10: $\quad$ **if** $t \leq E_{\text{res.}}$ **then**
11: $\quad\quad \text{mask}_{\text{res.}} \leftarrow h(\boldsymbol{p}_t)$
12: $\quad\quad \boldsymbol{p}_t[\text{mask}_{\text{res.}}] \leftarrow \boldsymbol{0}$
13: $\quad\quad$ Resample $\boldsymbol{i}_t[\text{mask}_{\text{res.}}]$
14: $\boldsymbol{P} \sim \text{Bernoulli}(\boldsymbol{p}_E)$ s.t. $\sum \boldsymbol{P} \leq \Delta$
15: Return $\boldsymbol{A} \oplus \boldsymbol{P}$

---

**PR-BCD**. For $L_0$-norm PGD we relax the discrete edge perturbations $\boldsymbol{P}$ from $\{0, 1\}^{(n \times n)}$ to $[0, 1]^{(n \times n)}$ as proposed by Xu et al. [43]. Each entry of $\boldsymbol{P}$ denotes the probability for flipping it. In each epoch we only look at a randomly sampled, non-contiguous block of $\boldsymbol{P}$ of size $b$ (line 3, line 10-13) and additionally ignore the diagonal elements (i.e. self-loops). If using an undirected graph, the potential edges are restricted to the upper/lower triangular $n \times n$ matrix. In each epoch $t \in \{1, 2, \dots\}$, $\boldsymbol{p}$ is added to / subtracted from the discrete edge weight (line 6). Note, we overload $\oplus$ s.t. $\boldsymbol{A}_{ij} \oplus p_{ij} = \boldsymbol{A}_{ij} + p_{ij}$ if $\boldsymbol{A}_{ij} = 0$ and $\boldsymbol{A}_{ij} - p_{ij}$ otherwise. We use $\boldsymbol{p}$ and $\boldsymbol{P}$ interchangeably while $\boldsymbol{p}$ only corresponds to the current subset/block of $\boldsymbol{P}_{\boldsymbol{i}_t}$. After each gradient update (line 7), the projection $\Pi_{\mathbb{E}[\text{Bernoulli}(\boldsymbol{p})] \leq \Delta}(\boldsymbol{p})$ adjusts the probability mass such that $\mathbb{E}[\text{Bernoulli}(\boldsymbol{p})] = \sum_{i \in b} p_i \leq \Delta$ and that $\boldsymbol{p} \in [0, 1]$ (line 8). In the end we draw $b$ sample s.t. $\boldsymbol{P} \in \{0, 1\}^{(n \times n)}$ via $\boldsymbol{P} \sim \text{Bernoulli}(\boldsymbol{p})$ (line 14).

The projection $\Pi_{\mathbb{E}[\text{Bernoulli}(\boldsymbol{p})] \leq \Delta}(\boldsymbol{p})$ likely results in many zero elements, but is not guaranteed to be sparse (for details see § C.1). If $\boldsymbol{p}$ has more than 50% non-zero entries, we remove the entries with the lowest probability mass such that 50% of the search space is resampled. Otherwise, we resample all zero entries in $\boldsymbol{p}$. However, one also might apply a more sophisticated heuristic $h(\boldsymbol{p})$ which we leave for future work (see line 11). After $E_{\text{res.}}$ epochs we fine-tune $\boldsymbol{p}$, i.e. we stop resampling and decay the learning rate as in [43]. We also employ early stopping for both stages ($t \leq E_{\text{res.}}$ and $t > E_{\text{res.}}$ with the epoch $t$) such that we take the result of the epoch with highest loss $\mathcal{L}$.

**Block size** $b$**.** With growing $n$ it is unrealistic that each possible entry of the adjacency matrix was part of at least one random search space of (P)R-BCD. As is apparent, with a constant search space size, the number of mutually exclusive chunks of the perturbation matrix grows with $\Theta(n^2)$ and this would imply a quadratic runtime. However, as evident in randomized black-box attacks [37], it is not necessary to test every possible edge to obtain an effective attack. In Fig. 3 (a), we analyze the influence of the block size $b$ on the adversarial accuracy. On small datasets and over a wide range of block sizes $b$, our method performs comparably (or sometimes even better) to its dense equivalent. For larger graphs, we observe that the block size $b$ has a stronger influence on the adversarial accuracy. However, as shown in Fig. 3 (b), one might increase the number of epochs for an improved attack strength. This indicates that PR-BCD successfully identifies the harmful edges to keep.

**GR-BCD.** As an alternative to PR-BCD, we propose Greedy R-BCD (GR-BCD) which greedily flips the entries with the largest gradient in the block so that after $E$ iterations the budget is met. It is even a little bit more scalable as it does *not* require $b > \Delta$ (see § C.2 for details such as the pseudo code).

**Limitations.** We solely propose approximate attacks that do not provide any guarantee on how well they approximate the actual optimization problem and, hence, only provide an *upper* bound on e.g. the adversarial accuracy. We also recommend monitoring the relaxation error. One could use certificates to get the respective lower bound, provided they were scalable enough. Even sparse smoothing [4] might be too slow since we need many forward passes. As our attacks rely on the gradient they also require that the victim model is (approximately) differentiable. Otherwise, the

approximation can become inappropriate. Moreover, we are limited by the scalability of the attacked GNN as we discuss next. For the theoretical complexities of all studied attacks, we refer to § E.

**Scalable GNNs.** Up to now, we implicitly assumed that we have enough memory to obtain the predictions and gradient towards the edges. GNNs that typically process the whole graph "at once", are inherently limited in their scalability. Our PR-BCD attack is even applicable when operating at those limits (see experiments on Products in § 5). To push the limits further, we now consider more scalable GNNs. Some notable scalable GNNs either sample subgraphs [7, 13] or, such as PPRGo [5], simplify the message passing operation. Next, we extend our PR-BCD to a local attack on PPRGo with *constant complexity including the (Soft Median) PPRGo* (we introduce the Soft Median in § 4).

**PPRGo.** To scale to massive graphs effectively, we need to obtain sublinear/constant complexity w.r.t. the number of nodes. This severely restricts the possibilities of how one might approach a global attack and is the reason why we now focus on local attacks (i.e. attacking single node $i$). For an $L$-layer message passing GNN we need to recursively compute the $L$-hop neighborhood to obtain the prediction of a single node. This makes it difficult to obtain a sublinear space complexity (here including the GNN)—especially if one considers arbitrary edge insertions. In contrast, PPRGo [5] leverages the Personalized Page Rank (PPR) matrix $\mathbf{\Pi} = \alpha(\boldsymbol{I} - (1 - \alpha)\boldsymbol{D}^{-1}\boldsymbol{A})^{-1}$ (row normalization) to reduce the number of explicit message passing steps to one (with feat. encoder $f_{\text{enc}}$):

$$\boldsymbol{p} = \text{softmax}\left[\text{AGG}\left\{(\mathbf{\Pi}_{uv}, f_{\text{enc}}(\boldsymbol{x}_u)), \forall\, u \in \mathbb{N}'(v)\right\}\right] \tag{3}$$

**Differentiable PPR Update.** For a local attack on PPRGo, we require a differentiable update of the respective PPR Scores for an edge perturbation on a weighted graph. We achieve this using the Sherman-Morrison formula through a closed-form rank-one update of row $i$ of the PPR matrix:

$$\tilde{\mathbf{\Pi}}_i = \alpha\left(\mathbf{\Pi}'_i - \frac{\mathbf{\Pi}'_{ii}\boldsymbol{v}\mathbf{\Pi}'}{1 + \boldsymbol{v}\mathbf{\Pi}'_{:i}}\right) \tag{4}$$

where $\mathbf{\Pi}' = \alpha^{-1}\mathbf{\Pi}$ and, with degree matrix $\boldsymbol{D}$, $\boldsymbol{v} = (\boldsymbol{D}_{ii} + \sum \boldsymbol{p})^{-1}(\boldsymbol{A}_i + \boldsymbol{p}) - \boldsymbol{D}_{ii}^{-1}\boldsymbol{A}_i$. This suffices to attack incoming edges of a node and since everything is differentiable ($\partial\mathcal{L}'/\partial\boldsymbol{p}$) we do not need a surrogate model (common practice [26, 36, 50]). In § C.3, we give details on the derivation and show how we can leverage the fact that PPRGo uses a *top-$k$-sparsified* PPR matrix to obtain constant complexity $\mathcal{O}(bk)$ (assuming $b \ll n$ and $k \ll n$). With $\Delta < b$, our approach comes with no restriction on how we can insert or remove incoming edges of a specific node. Other approaches such as [14, 26] gain scalability via restricting the set of admissible nodes for edge perturbations.

## 4 Scalable Defense

To complete the robustness picture we now shift focus to defenses. Unfortunately, we are not aware of any defense that scales to graphs significantly larger than PubMed. Thus, we propose a novel, scalable defense based on a robust message-passing aggregation, relying on recent advancements in differentiable sorting [31]. Our *Soft Median* not only comes with the best possible breakdown point of 0.5 but also can have a lower error than its hard equivalent for finite perturbations (see § D.3). Moreover, our Soft Median performs similarly to the recent Soft Medoid [18], but comes with better computational complexity w.r.t. the neighborhood size, lower memory footprint, and enables us to scale to bigger graphs. We can also use this aggregation neatly in the PPRGo architecture resulting in the *first defense that scales to massive graphs with over 111M nodes* (see Eq. 3).

**Background.** We typically have the message passing framework of a GNN:

$$\mathbf{h}_v^{(l)} = \sigma^{(l)}\left[\text{AGG}^{(l)}\left\{\left(\boldsymbol{A}_{vu}, \mathbf{h}_u^{(l-1)}\boldsymbol{W}^{(l)}\right), \forall\, u \in \mathbb{N}'(v)\right\}\right] \tag{5}$$

with neighborhood $\mathbb{N}'(v) = \mathbb{N}(v) \cup v$ including the node itself, the $l$-th layer message passing aggregation $\text{AGG}^{(l)}$, embedding $\mathbf{h}_v^{(l)}$, normalized adjacency matrix $\boldsymbol{A}$, weights $\boldsymbol{W}^{(l)}$, and activation $\sigma^{(l)}$.

**Related Work.** Following Günnemann [20], we classify defenses into three categories: (1) pre-processing [16, 22, 41], (2) training procedure [10, 43, 49], and (3) modifications of the architecture [11, 18, 23, 35, 42, 46–48]. All these previous defenses were not evaluated on graphs substantially larger than PubMed. Note GNNGuard [46] was only evaluated on a subset of arXiv, covering 20% of the nodes and 6% of the edges. Even though our attacks lend themselves well for adversarial

training but we leave it for future work due to the overhead during training. Instead, we build the observation of Geisler et al. [18] that common aggregations (e.g. sum or mean) in Eq. 5 are known to be non-robust. They propose a differentiable robust aggregation for $AGG^{(l)}$ and call it Soft Medoid. It is a continuous relaxation of the Medoid and requires the row/column sum over the distance matrix of the embedding of the nodes in the neighborhood. Hence this operation has a quadratic complexity w.r.t. the neighborhood size and comes with a sizable memory overhead during training and inference.

**Soft Median.** Intuitively, the Soft Median is a weighted mean where the weight for each instance is determined based on the distance to the dimension-wise median $\bar{x}$. This way, instances far from the dimension-wise median are filtered out. We define the Soft Median as

$$\mu_{\text{SoftMedian}}(\boldsymbol{X}) = \text{softmax}\left(-\boldsymbol{c}/T\sqrt{d}\right)^\top \boldsymbol{X} = \boldsymbol{s}^\top \boldsymbol{X} \approx \arg\min_{\boldsymbol{x}'\in\mathbb{X}} \|\bar{\boldsymbol{x}} - \boldsymbol{x}'\|, \qquad (6)$$

with the distances $c_v = \|\bar{\boldsymbol{x}} - \boldsymbol{X}_{v,:}\|$ and number of dimensions $d$. We use $\boldsymbol{X}$ as well as $\mathbb{X}$ interchangeably. For a single dimension, this closely resembles the soft sorting operator as proposed in [31] for the central element and can be understood as a soft version of the median. To apply it to multivariate inputs, we rely on the dimension-wise median which can be computed efficiently for practical choices of $d$. In contrast to the Soft Medoid, we do not require the distances between all input instances which makes the Soft Median much more efficient. Assuming $d$ is sufficiently small, the Soft Median scales linearly with the number of inputs $|\mathbb{N}'(v)|$.

**The temperature**. The temperature parameter $T$ controls the steepness of the weight distribution $\boldsymbol{s}$ between the neighbors. In the extreme case as $T \to 0$ we recover the instance which is closest to the dimension-wise Median (i.e. $\arg\min_{\boldsymbol{x}'\in\mathbb{X}} \|\bar{\boldsymbol{x}} - \boldsymbol{x}'\|$). In the other extreme case $T \to \infty$, the Soft Median is equivalent to the sample mean. We observe a similar empirical behavior as Geisler et al. [18] and we decide on a temperature value in our experiments by grid search.

**Breakdown point.** For any finite $T$, our proposed Soft Median has the best possible breakdown point of 0.5 as we state formally in Theorem 1 (for proof see § D.1). Note that despite the lower complexity compared to Soft Medoid, we maintain the same breakdown point:

**Theorem 1** *Let $\mathbb{X} = \{\mathbf{x}_1, \ldots, \mathbf{x}_n\}$ be a collection of points in $\mathbb{R}^d$ with finite coordinates and temperature $T \in [0, \infty)$. Then the Soft Median location estimator (Eq. 6) has the finite sample breakdown point of $\epsilon^*(\mu_{\text{Soft Median}}, \boldsymbol{X}) = 1/n\lfloor(n+1)/2\rfloor$ (asympt. $\lim_{n\to\infty} \epsilon^*(\mu_{\text{SoftMedian}}, \boldsymbol{X}) = 0.5$).*

We define the *weighted* Soft Median as

$$\mu_{\text{WSM}}(\boldsymbol{X}, \mathbf{a}) = C (\boldsymbol{s} \circ \mathbf{a})^\top \boldsymbol{X} \qquad (7)$$

where $\boldsymbol{s}$ is the softmax weight of Eq. 6 obtained using the weighted dimension-wise Median, $C$ normalizes s.t. $\sum \boldsymbol{s} \circ \mathbf{a} = \sum \mathbf{a}$, $\circ$ is the element-wise multiplication, and $\mathbf{a}$ the edges weights. Similarly to [18], we recover the message passing operation of a GCN [24] for $T \to \infty$.

**Empirical robustness.** The optimal breakdown point only assesses worst-case perturbations. Therefore, in Fig. 4, we analyze the $L_2$ distance in the latent space after the first message passing operation for a clean vs. perturbed graph. Empirically the Soft Median has a 20% lower error than the weighted sum of a GCN (we call

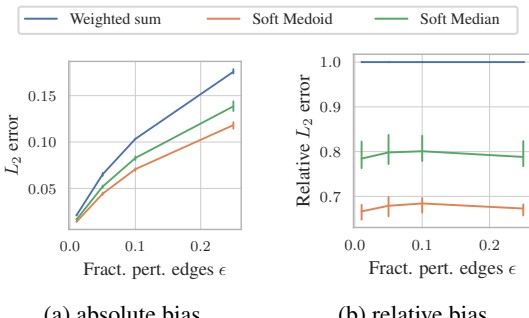

(a) absolute bias         (b) relative bias

Figure 4: Empirical bias $B(\epsilon)$ for the second layer of a GCN with GDC preproc. [25] network under PGD attack with $L_2$ distance. We use $T = 0.2$ and a relative budget of $\epsilon = 0.25$.

it sum since the weights do not sum up to 1). While here the Soft Medoid seems to be more robust, this is not consistent with the adversarial accuracy values in § 5. Interestingly, the Soft Median can outperform its hard equivalent in terms of the finite error as we show in § D.3.

**Limitations.** Our Soft Median has the best possible breakdown point which is a (well-established) indicator for robustness but does not prove adversarial robustness. As for most defenses, ours can provide a false sense of robustness. If possible, use attacks and certification techniques to verify the application-specific efficacy. Similar to the Soft Medoid in [18, 32], we show that the Soft Median can improve the certified robustness in § D.4. Naturally, our Soft Median also comes with higher cost than e.g. a naïve summation despite having the same asymptotic complexity. Nevertheless, the overhead seems to be reasonable as we show in our experiments and in combination with PPRGo one can mitigate the slightly higher memory requirements with smaller batch size.

# 5   Empirical Evaluation

In the following, we present our experiments (consisting of approx. 2,500 runs) to show the efficacy and scalability of our methods on the six graphs detailed in Table 1. **Attacks:** We benchmark **our** GR-BCD and PR-BCD against *PGD* [43], *greedy FGSM* (similar to Dai et al. [14]) as well as *DICE* [37]. **Defenses:** Besides regular/vanilla GNNs we compare **our** Soft Median GDC/PPRGo with *Soft Medoid GDC* [18], *SVD GCN* [16], *RGCN* [48], and *Jaccard GCN* [41]. For the Soft Median, we follow Soft Medoid GDC [18] and diffuse the adjacency matrix

Table 1: Dataset summary. For the dense adjacency matrix we assume 4 bytes per entry. We represent the sparse (COO) matrix via two 8 byte integer pointers and a 4 bytes float value per edge. We highlight configurations above 30 GB.

| Dataset | #Nodes $n$ | Size (dense) | Size (sparse) |
|---|---|---|---|
| Cora ML [2] | 2.8 k | 35.88 MB | 168.32 kB |
| Citeseer [28] | 3.3 k | 43.88 MB | 94.30 kB |
| PubMed [33] | 19.7 k | 1.56 GB | 1.77 MB |
| arXiv [21] | 169.3 k | **114.71 GB** | 23.32 MB |
| Products [21] | 2.4 M | **23.99 TB** | 2.47 GB |
| Papers 100M [21] | 111.1 M | **49.34 PB** | **32.31 GB** |

with PPR/GDC [25] and use PPRGo's efficient implementation to calculate the PPR scores. For the OGB datasets we use the public splits and otherwise sample 20 nodes per class for training/validation. We typically report the average over three random seeds/splits and the 3-sigma error of the mean. The full setup and details about baselines are given in § F.1 For supplementary material including the code and configuration see `https://www.in.tum.de/daml/robustness-of-gnns-at-scale`.

**Time and memory cost.** We want to stress again that most of the baselines barely scale to PubMed using a common 11GB GeForce GTX 1080 Ti (as we do). We only use a 32GB Tesla V100 for the experiments on Products with a full-batch GNN, since a three-layer GCN requires roughly 30 GB already during training. Extrapolating the overhead on PubMed to the largest dataset, Papers 100M, *traditional attacks and defenses would require roughly 1 exabyte ($10^{18}$ bytes) while for ours 11 GB suffice*. Our attacks and defenses are also reasonably fast. On arXiv (170 k nodes), we train for 500 epochs and run the global PR-BCD attack for 500 epochs. The whole training and attacking procedure requires less than 2 minutes. Moreover, one epoch on Papers 100M with the local PR-BCD attack takes less than 10 seconds. See § F.2 for further details and § E for theoretical complexities.

**Surrogate Loss.** We illustrate the losses in Fig. 5, where we clustered the losses in three groups. (1) incentivizing low margins: Cross Entropy CE and margin. (2) focusing on high-confidence nodes: Carlini-Wagner CW, the (neg.) CE of the most-likely, non-target class NCE, and ELU Margin. (3) focusing on nodes close to the decision boundary: MCE **(ours)** and tanh margin **(ours)**. In Fig. 6, we see that the losses of category (3), or equivalently obeying the properties of Definition 1 or Definition 2, are superior to the other losses. For example, with MCE and FGSM the accuracy drops twice as much as with CE. A detailed discussion and mathematical formulation of all losses can be found in § B. Additionally, we report further experiments backing our claims and discuss the losses' properties in more detail. Subsequently, we use MCE for greedy attacks and tanh margin otherwise.

**Robustness w.r.t. global attacks.** In Table 2, we present the experimental results for our proposed global attacks on the small dataset Cora ML since most baselines do not scale much further. Our attacks are as strong as their dense equivalents despite being much more scalable. In Fig. 7, we

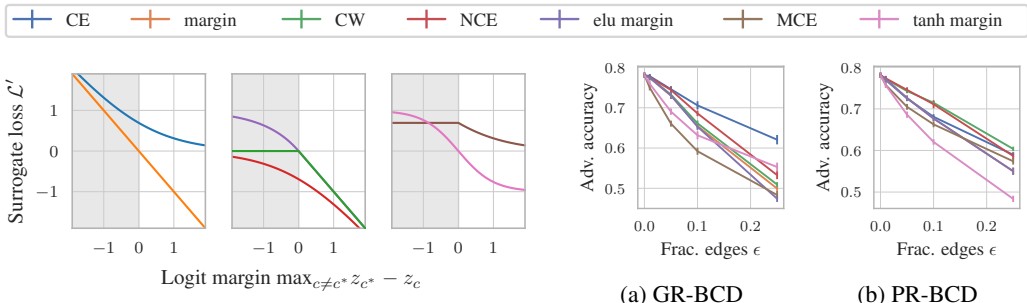

Figure 5: Losses for the binary case. The losses are grouped via their basic properties (see text).

Figure 6: Attacking GCN on Pubmed. The lower the adv. accuracy the better the loss.

Table 2: Comparing attacks (transfer from Vanilla GCN) and defenses. We show the adversarial accuracy for $\epsilon = 0.1$ on Cora ML, and the clean test accuracy (last column). We only highlight the **strongest defense** as the attacks perform similarly. Our approaches are underlined. See § F.3 for more datasets, budgets, and adaptive/direct attacks.

| Attack | FGSM | GR-BCD | PGD | PR-BCD | Acc. |
|---|---|---|---|---|---|
| Soft Median GDC | $0.769 \pm 0.002$ | $0.765 \pm 0.001$ | $0.758 \pm 0.002$ | $0.752 \pm 0.002$ | $0.824 \pm 0.002$ |
| Soft Median PPRGo | $\mathbf{0.778 \pm 0.001}$ | $\mathbf{0.781 \pm 0.002}$ | $\mathbf{0.769 \pm 0.001}$ | $\mathbf{0.770 \pm 0.001}$ | $0.821 \pm 0.001$ |
| Vanilla GCN | $0.641 \pm 0.003$ | $0.622 \pm 0.003$ | $0.662 \pm 0.003$ | $0.645 \pm 0.002$ | $0.827 \pm 0.003$ |
| Vanilla GDC | $0.672 \pm 0.005$ | $0.677 \pm 0.005$ | $0.679 \pm 0.003$ | $0.674 \pm 0.004$ | $\mathbf{0.842 \pm 0.003}$ |
| Vanilla PPRGo | $0.724 \pm 0.003$ | $0.726 \pm 0.002$ | $0.704 \pm 0.001$ | $0.700 \pm 0.002$ | $0.826 \pm 0.002$ |
| Soft Medoid GDC | $0.773 \pm 0.005$ | $0.775 \pm 0.003$ | $0.759 \pm 0.003$ | $0.761 \pm 0.003$ | $0.819 \pm 0.002$ |
| SVD GCN | $0.751 \pm 0.007$ | $0.755 \pm 0.006$ | $0.719 \pm 0.005$ | $0.724 \pm 0.006$ | $0.781 \pm 0.005$ |
| Jaccard GCN | $0.661 \pm 0.002$ | $0.664 \pm 0.001$ | $0.673 \pm 0.002$ | $0.667 \pm 0.003$ | $0.818 \pm 0.003$ |
| RGCN | $0.654 \pm 0.007$ | $0.665 \pm 0.005$ | $0.671 \pm 0.007$ | $0.664 \pm 0.004$ | $0.819 \pm 0.002$ |

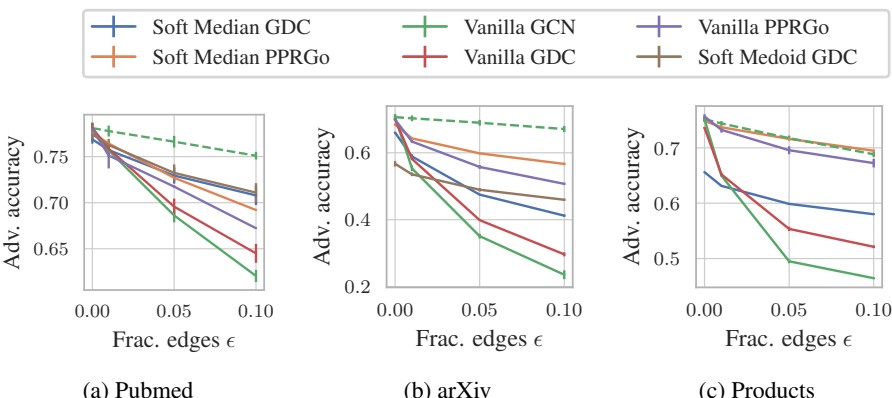

Figure 7: PR-BCD (DICE dashed) on the large datasets (transfer) where the adversarial accuracy denotes the accuracy after attacking with budget $\Delta = \epsilon m$.

compare our PR-BCD attack on all baselines that fit into memory or can be trained within 24 hours on the bigger datasets. On the large products dataset, it suffices to perturb roughly 2% of the edges to push the accuracy below 60%, i.e. reaching the performance of an MLP [21]. We conclude that GNNs on large graphs are indeed not robust (see also § F.5). Our defense Soft Median GDC and Soft Median PPRGo are consistently among the best models tested over all scales. For example with $\epsilon = 0.1$, the accuracy of a Vanilla GCN drops by an absolute 20% while for the Soft Median PPRGo we only lose 5%. To fit our Soft Median GDC on Products into memory, we had to reduce the number of hidden dimensions in comparison to its baselines. However, note that even a Vanilla GCN requires almost the entire memory of the 32 GB GPU. Despite the small sacrifice in clean accuracy, we already outperform most baselines for a budget of $\epsilon > 0.01$. We also faced similar scaling limitations for the Soft Medoid GDC baseline on arXiv. This highlights the lower memory requirements for our Soft Median. In § F.3, we present more exhaustive results and adaptive/direct attacks supporting the robustness of our defense but highlighting the importance of adaptiveness.

**Robustness w.r.t. local attacks.** In Fig. 8, we compare the results of our local PR-BCD with Nettack on Cora ML (undirected). We define the budget $\Delta_i = \epsilon d_i$ and select the nodes for each budget s.t. $\Delta_i \geq 1$. Similarly to Zügner et al. [50], we apply the attack only to the 10 nodes with highest confidence, 10 with lowest, and 20 random nodes (all correctly classified). For more datasets and budgets see § F.4. Our attack seems to be slightly stronger than Nettack on all architectures and budgets. Nettack and PR-BCD use a different strategy to make the original combinatorial optimization problem feasible (see § 1). Nettack uses a linearized surrogate model to select the adversarial edges. Evidently, this leads to a weaker attack compared to relaxing the optimization problem as we proposed with PR-BCD. On the large datasets Products and Papers 100M (directed), we outperform the simple DICE baseline substantially. We compare to DICE since Nettack is not scalable enough. In comparison to the small datasets, the Vanilla GCN/PPRGo are extremely fragile

Table 3: Attack success rate of PR-BCD (ours) and SGA [26] on a Vanilla GCN and Vanilla SGC [40]. The stronger attack is bold. For poisoning we retrain on the perturbed graph of an evasion attack.

| | Attack | | PR-BCD | | | SGA | | |
|---|---|---|---|---|---|---|---|---|
| | **Frac. edges** $\epsilon$, $\Delta_i = \epsilon d_i$ | | 0.25 | 0.50 | 1.00 | 0.25 | 0.50 | 1.00 |
| **Cora ML** | GCN | evasion | **0.38 ± 0.04** | **0.65 ± 0.04** | **0.96 ± 0.02** | 0.36 ± 0.04 | 0.51 ± 0.05 | 0.82 ± 0.03 |
| | | poisoning | 0.46 ± 0.05 | **0.79 ± 0.04** | **0.97 ± 0.02** | **0.47 ± 0.05** | 0.64 ± 0.04 | 0.95 ± 0.02 |
| | SGC | evasion | **0.45 ± 0.05** | **0.57 ± 0.05** | **0.97 ± 0.02** | 0.37 ± 0.04 | 0.43 ± 0.05 | 0.95 ± 0.02 |
| | | poisoning | **0.50 ± 0.05** | **0.66 ± 0.04** | 0.96 ± 0.02 | 0.47 ± 0.05 | 0.62 ± 0.04 | **0.97 ± 0.01** |
| **Citeseer** | GCN | evasion | **0.42 ± 0.05** | **0.66 ± 0.04** | **0.85 ± 0.03** | 0.34 ± 0.04 | 0.50 ± 0.05 | 0.72 ± 0.04 |
| | | poisoning | 0.53 ± 0.05 | **0.78 ± 0.04** | **0.94 ± 0.02** | **0.54 ± 0.05** | 0.76 ± 0.04 | 0.93 ± 0.02 |
| | SGC | evasion | **0.39 ± 0.04** | **0.62 ± 0.04** | **0.91 ± 0.03** | 0.35 ± 0.04 | 0.55 ± 0.05 | 0.85 ± 0.03 |
| | | poisoning | **0.49 ± 0.05** | **0.77 ± 0.04** | 0.92 ± 0.03 | **0.49 ± 0.05** | 0.74 ± 0.04 | **0.97 ± 0.01** |
| **arXiv** | GCN | evasion | **0.92 ± 0.03** | **1.00 ± 0.00** | **1.00 ± 0.00** | 0.58 ± 0.05 | 0.90 ± 0.03 | 0.98 ± 0.01 |
| | | poisoning | 0.82 ± 0.03 | **0.99 ± 0.01** | **1.00 ± 0.00** | 0.52 ± 0.05 | 0.82 ± 0.04 | 0.98 ± 0.01 |
| | SGC | evasion | **0.91 ± 0.03** | **0.97 ± 0.01** | **1.00 ± 0.00** | 0.83 ± 0.04 | 0.94 ± 0.02 | 0.94 ± 0.02 |
| | | poisoning | **0.91 ± 0.03** | **0.97 ± 0.01** | **1.00 ± 0.00** | 0.83 ± 0.04 | 0.94 ± 0.02 | 0.94 ± 0.02 |

and much lower budgets $\Delta_i$ suffice to flip almost every node's prediction. Our proposed defense Soft Median PPRGo on the other hand remains similarly robust as on the small datasets. On Papers 100M with $\Delta_i = 0.25$, the Soft Median PPRGo reduces the attacker's success rate from around 90% to just 30% (90% vs. 1% on Products with $\Delta_i = 0.5$).

In Table 3, we compare to the SGA attack of Li et al. [26] that transfers the attacks from a SGC [40] surrogate. With our PR-BCD we attack the respective model directly. We follow SGA and obtain a poisoning attack by applying the perturbations of an evasion attack to the graph before training. Our PR-BCD clearly dominates SGA–even on SGC. This demonstrates how generally applicable our PR-BCD is without any modifications. We hypothesize that PR-BCD is stronger since, in contrast to SGA, it does not constrain the edge perturbations to be within a subgraph. Moreover, the large gap for a GCN highlights the importance of adaptive attacks (i.e. no surrogate). Also in terms of scalability, we find PR-BCD to be superior, even though SGA is efficient on graphs up to the size of arXiv. However, on products, we observe that for $s = 3$ SGC message passing steps we sometimes require more than 11 Gb and $s = 4$ we typically require more than 32 Gb. However, our PR-BCD with PPRGo scales to graphs 2 magnitudes larger (Papers100M) and requires less than 11 GB (see § F.2).

# 6 Conclusion

We study the adversarial robustness of GNNs at scale. We tackle all three of the identified challenges: (1) we introduce surrogate losses for global attacks that can double the attack strength, (2) we principally scale first-order attacks that optimize over the

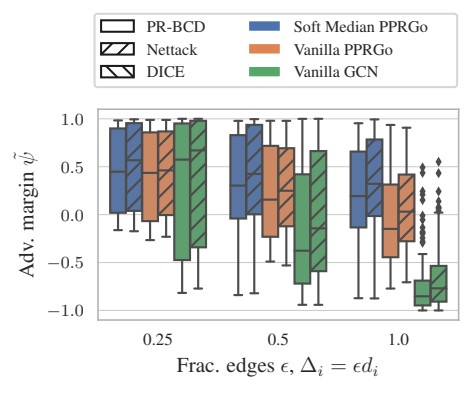

(a) Cora ML

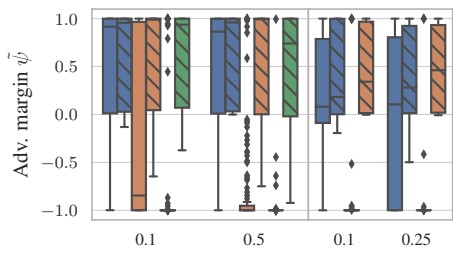

(b) Products (left) & Papers100M (right)

Figure 8: Adversarial classification margins $\tilde{\psi}_i$ of the attacked nodes. In (a), we compare our local PR-BCD attack with Nettack [50] on (undirected) Cora ML. In (b), we show the results on the (directed) large-scale datasets Products (2.5 million nodes) and Papers 100M (111 million nodes), respectively. Our Soft Medoid PPRGo resists the attacks much better than the baselines.

quadratic number of possible edges, and (3) we propose a scalable defense using our novel Soft Median which is differentiable as well as provably robust. We show that our attacks and defenses are practical by scaling to graphs of up to 111 million nodes. In some settings our defense reduces the attack's success rate from around 90 % to 1 %. Most importantly, our work enables the assessment of robustness for massive-scale applications with GNNs.

## Acknowledgments and Disclosure of Funding

This research was supported by the Helmholtz Association under the joint research school "Munich School for Data Science - MUDS".

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
