# A  Notation

In this section, we explicitly summarize the notation. We intended to introduce all terms when needed. However, due to the symbiosis of Graph Neural Networks (GNNs), adversarial robustness, adversarial attacks, adversarial defenses, large-scale optimization, and robust statistics we inevitably need a large number of different symbols. We give the important symbols in Table A.1. Recall that we formulate GNNs as a recursive transformation and aggregation over the features/embedding of the neighboring nodes (with a potentially weighted/normalized adjacency matrix $\boldsymbol{A}$), i.e.

$$\mathbf{h}_v^{(l)} = \sigma^{(l)} \left[ \mathrm{AGG}^{(l)} \left\{ \left( \boldsymbol{A}_{vu}, \mathbf{h}_u^{(l-1)} \boldsymbol{W}^{(l)} \right), \forall\, u \in \mathbb{N}'(v) \right\} \right] \tag{A.1}$$

**Asymptotics.** To describe the growth of a function $f(n)$ and similarly the complexity of algorithms, we use $f(n) = \mathcal{O}(g(n))$ and $f(n) = \Theta(g(n))$ (here $n$ does not denote the number of nodes). Roughly speaking, $f(n) = \mathcal{O}(g(n))$ means that the growth is upper bounded (up to constant factors) and $f(n) = \Theta(g(n))$ means that $f(n)$ grows as fast as $g(n)$. While we do not give a formal definition, we quickly want to recall the well-known facts (that hold under certain conditions which are naturally fulfilled in our applications/analysis):

$$\lim_{n \to \infty} \frac{f(n)}{g(n)} < \infty \Rightarrow f(n) = \mathcal{O}(g(n)) \tag{A.2}$$

$$0 < \lim_{n \to \infty} \frac{f(n)}{g(n)} < \infty \Rightarrow f(n) = \Theta(g(n)) \tag{A.3}$$

Table A.1: Here we list the most important symbols used in this work.

| | |
|---|---|
| $k$ | typically used as the threshold in some top $k$ operation/sparsification |
| $d$ | number of dimension / features (e.g. $\boldsymbol{X}$ is of shape $n \times d$) |
| AGG | some aggregation (e.g. sum, max, Soft Median, ...) |
| $\boldsymbol{uv}^\top$ | two (column) vectors constructing a rank-1 matrix |
| $\mathcal{G} = (\boldsymbol{A}, \boldsymbol{X})$ | the (attributed) graph |
| $n$ | number of nodes |
| $m$ | number of edges |
| $\boldsymbol{A}$ | (clean) adjacency matrix (shape $n \times n$) |
| $\mathbf{a}$ | weights of a row/neighborhood given by the (weighted) adjacency matrix |
| $\boldsymbol{X}$ | features / node attributes as a matrix (shape $n \times d$) |
| x $\mathbb{X}$ | features / node attributes as a set |
| $\mathbb{N}(i)$ | the (direct) neighbors of node $i$ |
| $\boldsymbol{D}$ | the degree matrix of a graph |
| $\boldsymbol{\Pi}$ | Stationary distribution of a random walk with restarts / Personalized Page Rank (PPR) matrix, here $\boldsymbol{\Pi} = \alpha(\boldsymbol{I} - (1-\alpha)\boldsymbol{D}^{-1}\boldsymbol{A})^{-1}$ |
| $f_\theta(\boldsymbol{A}, \boldsymbol{X})$ | Graph Neural Network (GNN) for node classification |
| $\theta$ | (all) model parameters |
| $\boldsymbol{W}$ | weight matrix (contained in $\theta$) |
| $\mathbf{h}$ | embeddings / hidden state |
| $\sigma(\mathbf{h})$ | some (nonlinear) activation function |
| $\mathrm{softmax}(\mathbf{h})$ | the softmax operation/activation |
| $T$ | temperature parameter to control the steepness of the softmax |
| $\boldsymbol{s}$ | the weight vector of a softmax operation |
| $\boldsymbol{p}, \boldsymbol{p}_i$ | probability/confidence scores predicted for an arbitrary node $i$: $\boldsymbol{p} = f_\theta(\boldsymbol{A}, \boldsymbol{X})$ |
| $\boldsymbol{z}, \boldsymbol{z}_i$ | logits / pre-softmax activation predicted for an arbitrary node $i$: $\boldsymbol{z} = f_\theta(\boldsymbol{A}, \boldsymbol{X})$ (overloaded notation) |
| $y, y_i$ | label (ground truth) |
| $\mathcal{L}$ | (target) loss |
| $\mathcal{L}_{0/1}$ | 0/1 loss corresponding to the accuracy |
| $\mathcal{L}'$ | surrogate loss |
| $l$ | typically the layer index, e.g. $\sigma^{(l)}(\mathbf{h}^{(l-1)})$ is the $l$-th layer activation |

| | |
|---|---|
| $L$ | number of layers |
| $\mathbb{C}$ | set of classes |
| $c$ | we typically use $c$ while iterating the classes |
| $c^*$ | denotes the target class / ground truth |
| $\mathbb{V}^+$ | set of correctly classified nodes |
| $\psi$ | classification margin in the confidence space $\psi = \min_{c \neq c^*} p_{c^*} - p_c$ |
| $\tilde{A}$ | (perturbed) adjacency matrix, e.g. during/after an attack |
| $\Delta$ | budget of an attack |
| $\epsilon$ | relative budget usually w.r.t. the number of edges $m$, i.e. $\Delta = \epsilon \cdot m$ |
| $\alpha$ | learning rate |
| $t$ | index of epoch, i.e. $t \in \{0, \ldots, E\}$ |
| $E$ | number of epochs |
| $E_{\text{res.}}$ | number of epochs with resampling of the random block (see Algo. 1) |
| $\Pi(\ldots)$ | A projection in projected gradient descent |
| $P$ | perturbations $\tilde{A} = A \oplus P$ |
| $p$ | in context of PR-BCD, $p$ corresponds to the current subset/block $P_{i_t}$ |
| $i$ | indices representing the current block in GR-BCD/PR-BCD |
| $\oplus$ | exclusive or operation if inputs are binary. If inputs are floats: $A_{ij} \oplus p_{ij} = A_{ij} + p_{ij}$ if $A_{ij} = 0$ and $A_{ij} - p_{ij}$ otherwise |
| $\Pi_{\text{criterion}}(X)$ | project operation w.r.t. some "criterion" |
| $X$ and $\mathbb{X}$ | inputs in the context of the analysis of the Soft Median (matrix and set notation) |
| $\mu(X)$ | some location estimate based on the inputs $X$ |
| $\tilde{X}_\epsilon$ and $\tilde{\mathbb{X}}_\epsilon$ | perturbed feature matrix used in breakdown point analysis (matrix and set notation) |
| $c$ | the cost/distances of the input instances to the dimension-wise median |
| $\circ$ | the element-wise multiplication |
| $C$ | the normalization of the weighted Soft Median |

## B  Surrogate Losses

Hereinafter, we supplement the elaborations and experiments of the main part focusing on the surrogate losses. For the proof of Proposition 1 we refer to § B.4. We give a full definition of the losses in Table B.1. To simplify notation, we define the losses for a single node (except for MCE) and denote the correct class with $c^*$. Note that $\mathbb{V}^+$ is the set of correctly classified nodes, $p$ is the vector of confidence scores, and $z$ is the vector with logits.

Table B.1: Correspondence of global losses and their fulfilled properties corresponding to § 2. "+" means that the property is obeyed and "-" that it is violated. Nevertheless, we add the subjective category "o" to denote if the loss is partially / approximately consistent with the property. NCE is the acronym for non-target class CE, and *elu* stands for Exponential Linear Unit (smooth ReLU relaxation $\text{elu}(z) = \min[\alpha \cdot (\exp(z) - 1), \text{ReLU}(z)]$ with $\alpha = 1$ in all our experiments ).

| Category/Group | | $\downarrow$ Loss \Properties $\rightarrow$ | (I) | (II) | (A) | (B) |
|---|---|---|---|---|---|---|
| (1) | focus on negative margins | $\text{CE} = -z_{c^*} + \log(\sum_{c \in \mathbb{C}} \exp(z_c))$ | - | + | - | o |
| | | $\text{margin} = \max_{c \neq c^*} z_c - z_{c^*}$ | - | - | - | - |
| (2) | focus on high-confidence nodes | $\text{CW} = \min(\max_{c \neq c^*} z_c - z_{c^*}, 0)$ | + | - | + | - |
| | | $\text{NCE} = \max_{c \neq c^*} z_c - \log(\sum_{c' \in \mathbb{C}} \exp(z_{c'}))$ | o | + | + | o |
| | | $\text{elu margin} = -\text{elu}(\max_{c \neq c^*} z_{c^*} - z_c)$ | o | - | + | o |
| (3) | focus on nodes close to decision boundary | $\text{MCE} = \frac{-1}{|\mathbb{V}^+|} \sum_{i \in \mathbb{V}^+} z_{i,c^*} - \log(\sum_{c \in \mathbb{C}} \exp(z_{i,c}))$ | + | + | + | o |
| | | $\text{tanh margin} = -\tanh(\max_{c \neq c^*} z_{c^*} - z_c)$ | o | + | + | + |

Ma et al. [27] propose a black-box attack based on a random-walk importance score. They select 1% of nodes based on some centrality score and then gradually increase the feature perturbation of the selected nodes. They report that their (initial) black-box attack based on this importance score is only effective for low budgets. Since they propose their attack based on an analysis of a white-box

attack, they conclude that this is due to a mismatch between accuracy and CW. First, note that their definition of the CW loss is what we call margin loss: margin $= \max_{c \neq c^*} \mathbf{z}_c - \mathbf{z}_{c^*}$. Instead, we follow Xu et al. [43] CW $= \min(\max_{c \neq c^*} \mathbf{z}_c - \mathbf{z}_{c^*}, 0)$ (see Eq. 6 in [43] with $\kappa = 0$).

In summary, their finding is largely unrelated to our observations for the following reasons: (1) They select a fixed number of nodes (1%) and observe that for severe feature perturbations the loss changes but the accuracy does not. So it seems like that if the perturbation budget of the attacked nodes is large enough then the predictions of the whole receptive field are successfully flipped. Instead, our study is exclusively for structure perturbations. (2) They study how to spread the perturbed nodes over the graph. Instead, we discuss that e.g. with CE most of the budget is spent on nodes that are wrongly classified in the clean graph (Properties I and A). (3) In contrast to Ma et al. [27], we also consider the fact that e.g. the CW loss comes with the risk of unsuccessfully spending all/too much budget on high-confidence nodes (Properties II and B).

## B.1 Learning Dynamics

The necessity of studying the surrogate losses originates from an unexpected behavior of the Cross Entropy (CE) loss and accuracy during an attack. In some cases, the loss increases significantly while the accuracy stays constant or even increases. Similarly, the so-called Carlini-Wagner (CW) loss [6, 43] is very noisy over the epochs $t$ during a global attack (e.g. see Fig. B.1 (e)). Besides the violation of Definition 1 and 2, we hypothesize that the CW is inappropriate due to the effect on the optimization dynamics as we quickly explained in § 2. We now first discuss the learning curves and then come back to the phenomenon.

In Fig. B.1, we show the loss and accuracy during an attack for a large subset of datasets (see Table 1). Here, we study a single-layer GCN on a directed graph since this comes with a 1-to-1 correspondence between modified edges and attacked nodes. Especially for small budgets, the (CE) can increase while the accuracy does not decline. This shows the mismatch between (CE) and accuracy. In Fig. B.1 (a-c), one can see that in the first epochs the accuracy reduces but then recovers almost to the clean accuracy during the attack. This happens despite the monotonic increase of the CE loss. In Fig. B.2, we see that a similar, slightly-weaker behavior also holds for the common case of a two-layer / three-layer GCN on an undirected graph. That the observed effects appear to be weaker can be attributed to (1) the fact that an *undirected* edge always influences both nodes and (2) the diffusion through multiple message-passing steps.

Now we come back to the phenomenon that the node's prediction can oscillate around the decision boundary (as pointed out in § 2). The main reason is the zero gradient if $\psi < 0$ in the MCE (and CW) loss: CW $= \min(\max_{c \neq c^*} \mathbf{z}_c - \mathbf{z}_{c^*}, 0)$. To fully explain the reasons we need to dive into the PGD update and project step in epoch $t$: 5

$$\mathbf{p}_t = \Pi_{\mathbb{E}[\text{Bernoulli}(\mathbf{p}_t)] \leq \Delta} \left[ \mathbf{p}_{t-1} + \alpha_{t-1} \nabla \mathcal{L}(\mathbf{p}_{t-1}, \dots) \right]. \tag{B.1}$$

We can rewrite this expression to

$$\mathbf{p}_t = \Pi_{[0,1]}(\mathbf{p}_{t-1} \underbrace{+\alpha_t \nabla \mathcal{L}(\mathbf{p}_{t-1}, \dots)}_{\text{gradient update}} \underbrace{-\eta_t \mathbf{1}}_{\text{correction}}) \tag{B.2}$$

where $\Pi_{[0,1]}$ clamps the values to the range $[0,1]$ ($\Rightarrow \mathbf{p}_t \in [0,1]^b$) and $\eta_t$ is chosen s.t. $\mathbb{E}[\text{Bernoulli}(\mathbf{p}_t)] \leq \Delta$ (i.e. $\sum \Pi_{[0,1]}(\mathbf{p}_t) \leq \Delta$). There are two competing terms: 1) the gradient update $\alpha_{t-1} \nabla \mathcal{L}(\mathbf{p}_{t-1}, \dots)$ and 2) the correction $\eta_t \mathbf{1}$ (typically lowers all weights in $\mathbf{p}_t$). For reasonable parameter choices, the potential perturbations in $\mathbf{p}_t$ are competing since our budget is limited (i.e. to maximize the loss we would like to flip more edges than budget we have). Then, after some epochs ($t > t_0$), we will have $\eta_t > 0$ and subtract $\eta_t$ from each element in $\mathbf{p}_{t-1}$.

Now if we choose a loss $\mathcal{L}$ (e.g. CW or MCE loss) that has zero gradient, as soon as a node $v$ is misclassified ($\psi_v < 0$), the responsible edge(s) will not benefit from a "gradient update" anymore but $\eta_t > 0$ is still subtracted. So after some iterations node $v$ will be again correctly classified since the required edge flips in $\mathbf{p}_t$ lost weight/strength. This leads to instability.

The symptoms are particularly visible in Figure 9e for the CW loss (after $t_0 = 25$ epochs the accuracy oscillates around 0.7). Moreover, the accuracy for the CW loss (Figure 9 d-f and Figure 10 d-f) are noisier than for the CE or tanh margin losses (other subfigures).

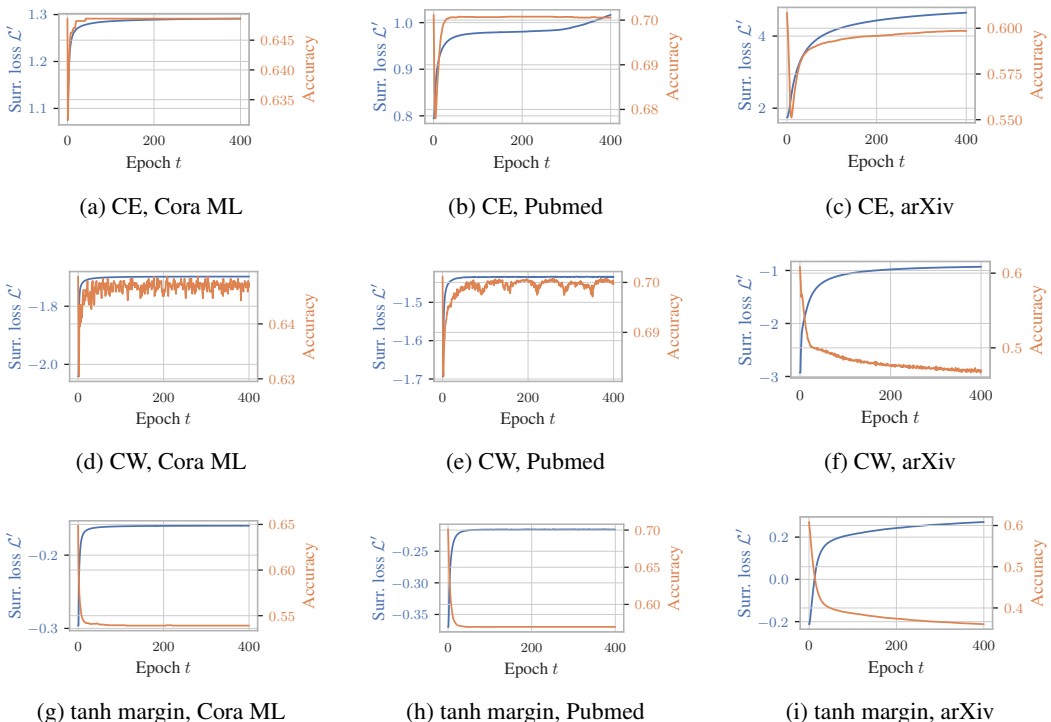

(a) CE, Cora ML        (b) CE, Pubmed        (c) CE, arXiv

(d) CW, Cora ML        (e) CW, Pubmed        (f) CW, arXiv

(g) tanh margin, Cora ML        (h) tanh margin, Pubmed        (i) tanh margin, arXiv

Figure B.1: PGD attack on a single layer GCN with *directed* graph and $\epsilon = 0.01$.

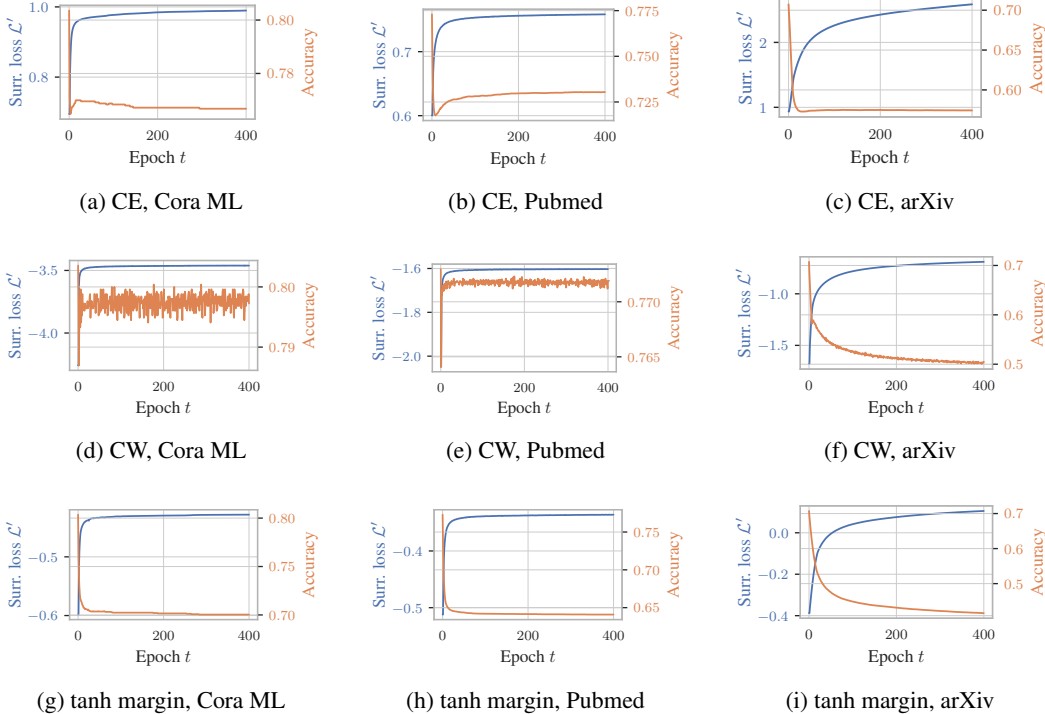

(a) CE, Cora ML        (b) CE, Pubmed        (c) CE, arXiv

(d) CW, Cora ML        (e) CW, Pubmed        (f) CW, arXiv

(g) tanh margin, Cora ML        (h) tanh margin, Pubmed        (i) tanh margin, arXiv

Figure B.2: PGD attack (no fine tuning) on a single layer GCN with *undirected* graph graph and $\epsilon = 0.01$.

## B.2 What Nodes Are Being Attacked?

In Fig. B.3, B.4, B.5 and B.6, we show the distribution for different datasets and budgets. Here we underline that our findings are consistent for these variations in the experiment setup. These plots essentially show the same thing as similarly to Fig. 2 for more configurations. However, instead of bar plots we show density plots since they allow more nuanced conclusions. We distinguish again between a directed and undirected graph. With CE we mostly attack nodes with a negative margin. For example in plot Fig. B.4 (a), the distribution for the attacked nodes is extremely spiky (at ($\psi = -1$)) leading to a failing kernel density estimate. On the contrary, CW attacks correctly classified nodes proportionally to the clean distribution, and *our* tanh margin focuses on nodes close to the decision boundary. We see that our observation holds over a wide range of datasets and budgets. All the stated observations are particularly clear if we consider tiny budgets and a directed graph. In the undirected case, using the CE and tanh margin also target confident nodes. Similarly to before, we attribute this effect to (1) the fact that on an undirected graph an edge always influences both nodes and (2) the diffusion via the recursive message passing. For large budgets and an undirected graph the differences between the losses become less significant. Simultaneously, an increasing budget becomes less realistic for many applications.

In Fig. B.7, which is similar to Fig. 6, we provide larger budgets and more datasets. As long as the budget is sufficiently small, our observations hold: (1) for the greedy FGSM attack the MCE is the strongest loss and (2) for the projected gradient descent attacks (PGD and our PR-BCD) the tanh margin outperforms other losses. With larger budgets, the elu margin seems to be a particularly strong choice. Note that the elu margin diverges for $\psi \to 1$ but, in contrast to the CW, it is smooth and encourages confident misclassifications (see Fig. 5). We hypothesize that for sufficiently large budgets, it makes sense to incentivize first attacking high-confidence nodes instead of nodes close to the decision boundary since those are presumably the most difficult to convert. However, such large budgets (e.g. $\epsilon > 0.25$) are not realistic for most applications. On arXiv, the elu margin is already slightly stronger than the tanh margin for $\epsilon > 0.075$. This is probably largely driven by the fact that we only attack 29% of nodes (i.e. the test set) while the budget is calculated relatively to all edges. Hence, in comparison to Cora ML, Citeseer, or Pubmed, we effectively have a four times larger budget per attacked node.

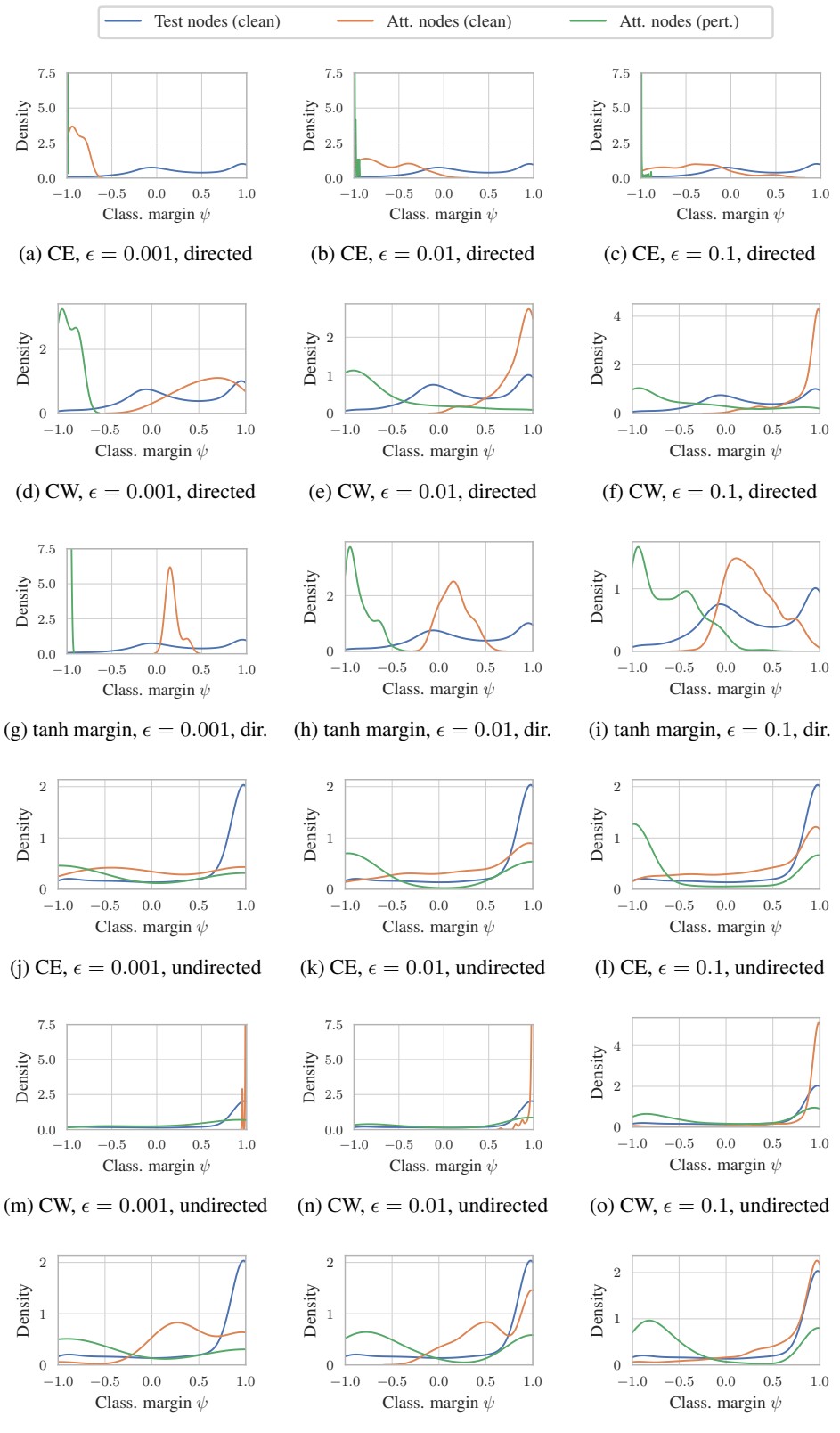

Figure B.3: Distribution of nodes attacked before/after PGD attack on Vanilla GCN on Cora ML.

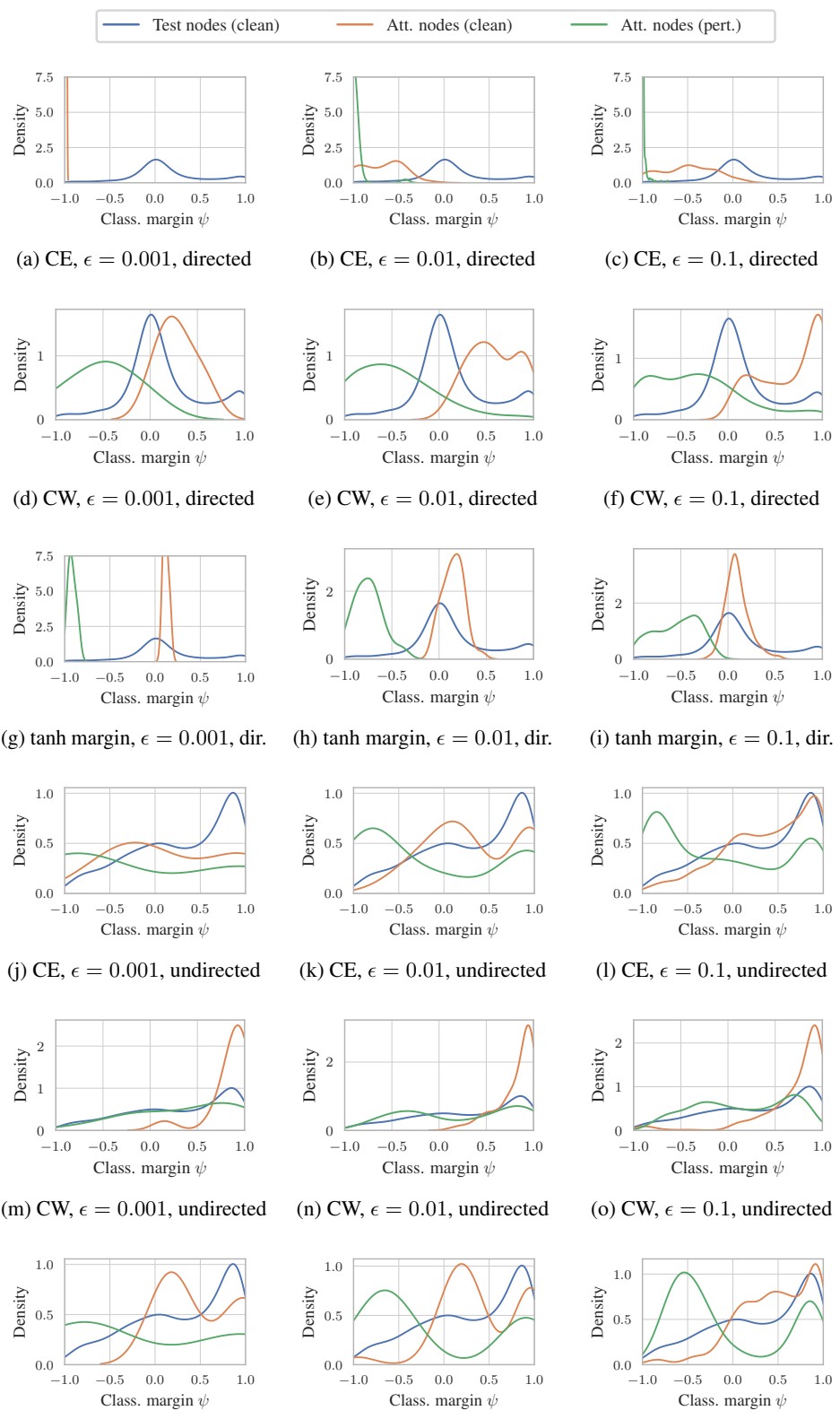

Figure B.4: Distribution of nodes attacked before/after PGD attack on Vanilla GCN on Citeseer.

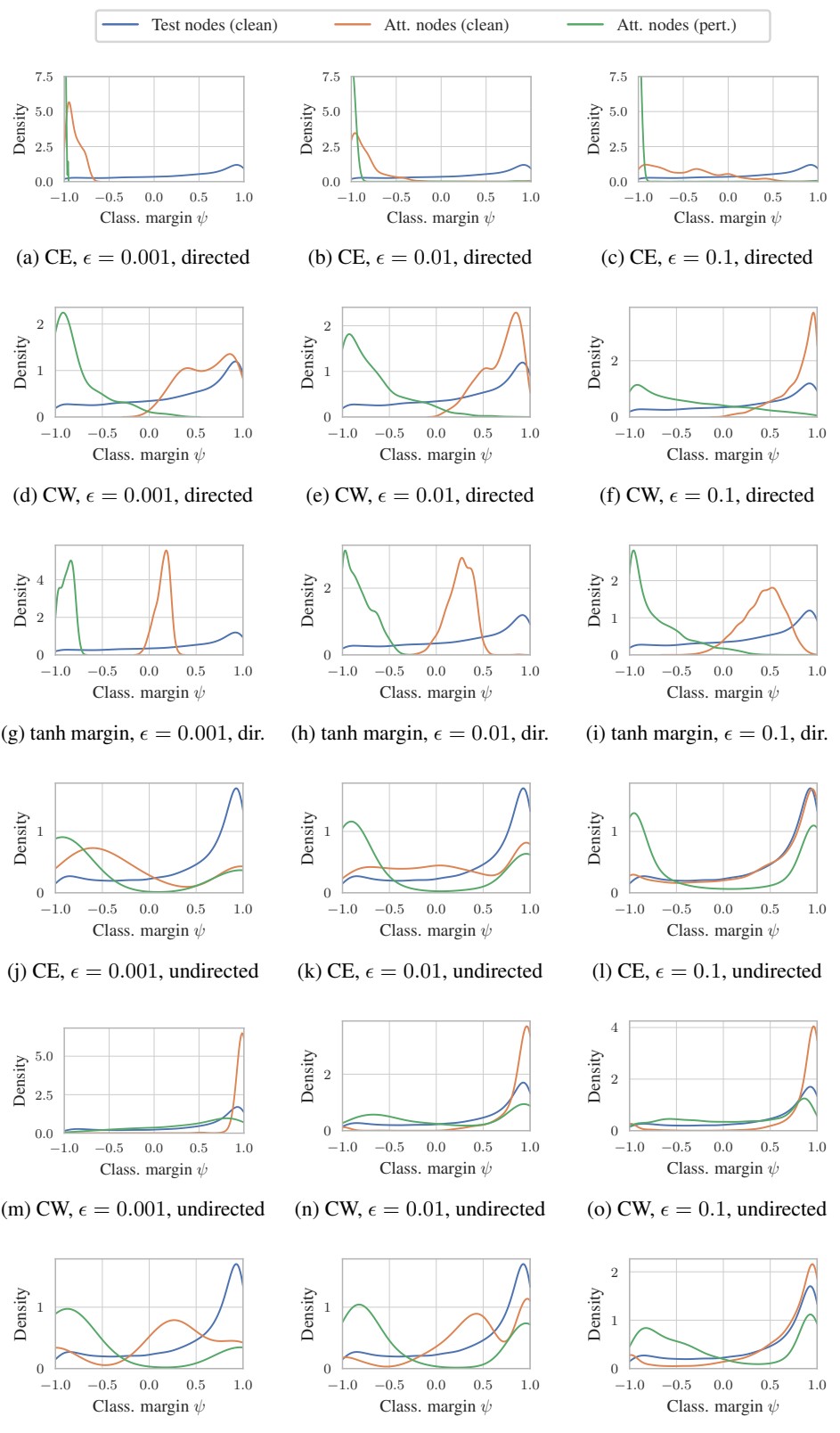

Figure B.5: Distribution of nodes attacked before/after PGD attack on Vanilla GCN on Pubmed.

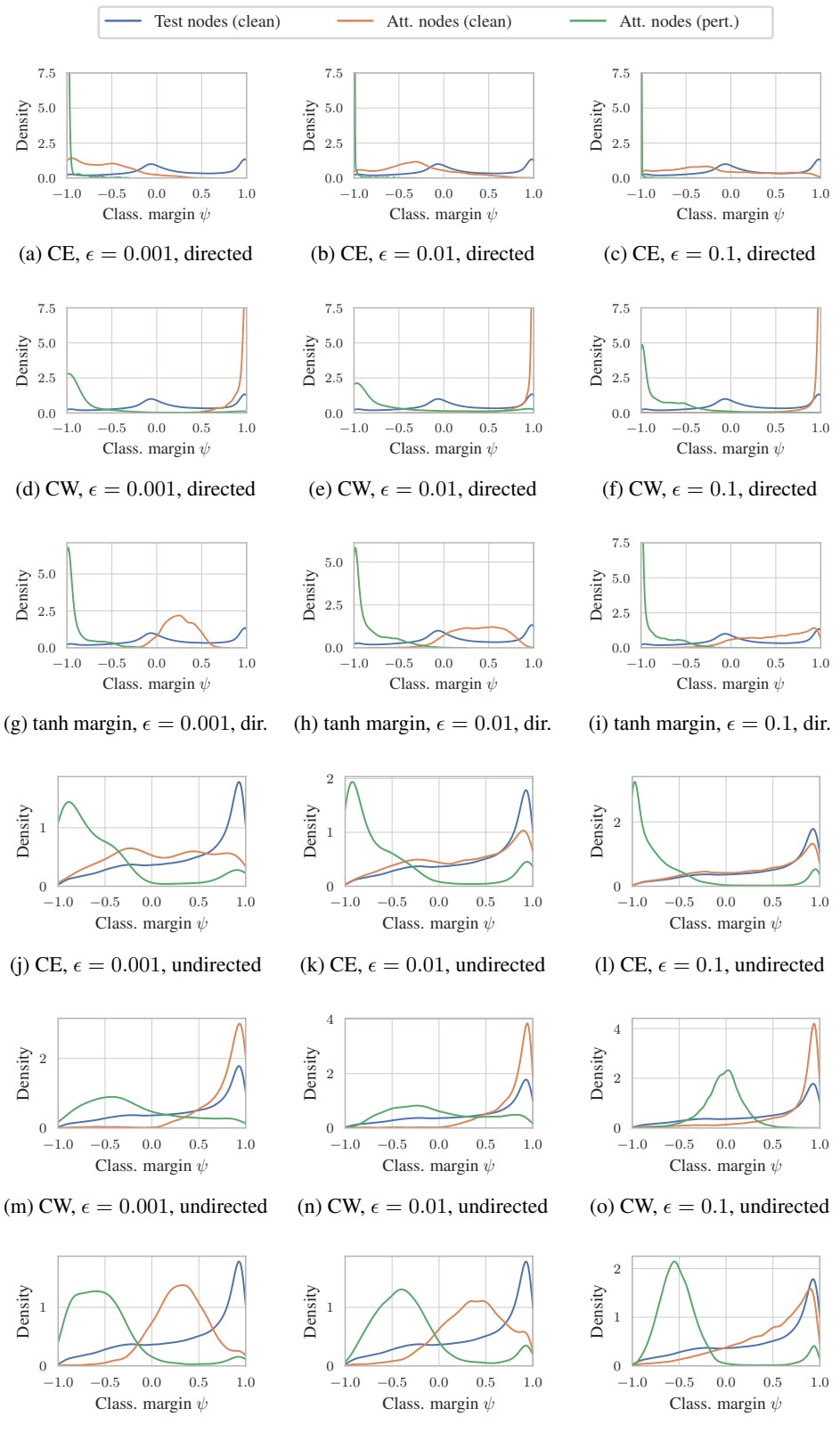

Figure B.6: Distribution of nodes attacked before/after PGD attack on Vanilla GCN on arXiv.

## B.3 Impact of Surrogate Losses on Attack Strength

In Table B.2 and Table B.3, we additionally evaluate how the different losses perform for other models than the Vanilla GCN. Baring a few exceptions, we conclude that our analysis and choice of losses is model agnostic and that our claims and observations hold also for the other architectures.

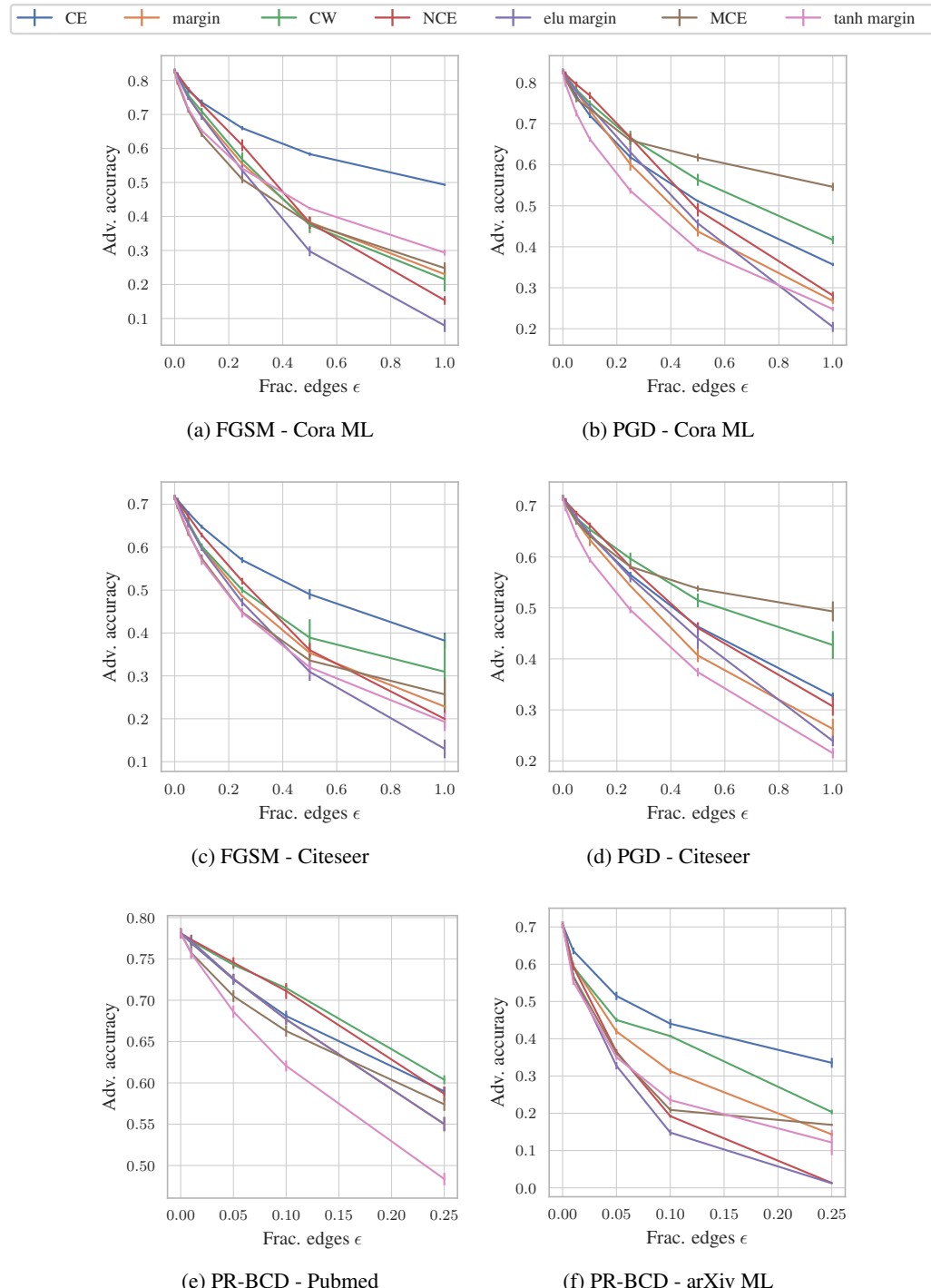

Figure B.7: Comparison of the losses on a Vanilla GCN attacked with a greedy attack and a projected gradient / coordinate descent algorithm. $\epsilon$ denotes the fraction of edges perturbed (relative to the clean graph). The lower the adversarial accuracy the better the loss.

Table B.2: Adversarial accuracy comparing the conventional losses with our losses over the different architectures on Cora ML (transfer attack). $\epsilon$ denotes the fraction of edges perturbed.

| | | Architecture | Soft Median GDC | Soft Median PPRGo | Vanilla GCN | Vanilla GDC | Vanilla PPRGo | Soft Medoid GDC | Jaccard GCN | RGCN |
|---|---|---|---|---|---|---|---|---|---|---|
| **FGSM** | $\epsilon = 0.01$ | CE | 0.813 ± 0.002 | 0.816 ± 0.000 | 0.814 ± 0.004 | 0.826 ± 0.002 | 0.818 ± 0.002 | 0.810 ± 0.003 | 0.806 ± 0.003 | 0.807 ± 0.002 |
| | | margin | 0.820 ± 0.001 | 0.820 ± 0.001 | 0.813 ± 0.003 | 0.825 ± 0.003 | 0.818 ± 0.002 | 0.816 ± 0.002 | 0.804 ± 0.003 | 0.804 ± 0.002 |
| | | CW | 0.820 ± 0.001 | 0.819 ± 0.001 | 0.814 ± 0.003 | 0.826 ± 0.003 | 0.818 ± 0.001 | 0.816 ± 0.002 | 0.805 ± 0.003 | 0.804 ± 0.003 |
| | | NCE | 0.822 ± 0.001 | 0.820 ± 0.001 | 0.818 ± 0.003 | 0.831 ± 0.003 | 0.822 ± 0.002 | 0.818 ± 0.002 | 0.809 ± 0.002 | 0.807 ± 0.003 |
| | | elu margin | 0.821 ± 0.001 | 0.819 ± 0.001 | 0.814 ± 0.003 | 0.826 ± 0.003 | 0.817 ± 0.002 | 0.817 ± 0.002 | 0.804 ± 0.003 | 0.804 ± 0.002 |
| | | MCE | 0.811 ± 0.002 | 0.813 ± 0.001 | **0.795 ± 0.004** | 0.811 ± 0.003 | 0.807 ± 0.001 | 0.808 ± 0.002 | **0.791 ± 0.003** | **0.794 ± 0.000** |
| | | tanh margin | **0.806 ± 0.001** | **0.811 ± 0.001** | 0.801 ± 0.003 | **0.810 ± 0.003** | **0.803 ± 0.002** | **0.807 ± 0.003** | 0.794 ± 0.002 | 0.796 ± 0.001 |
| | $\epsilon = 0.05$ | CE | 0.779 ± 0.001 | 0.789 ± 0.000 | 0.771 ± 0.004 | 0.776 ± 0.001 | 0.781 ± 0.002 | 0.776 ± 0.001 | 0.768 ± 0.003 | 0.764 ± 0.001 |
| | | margin | 0.799 ± 0.002 | 0.803 ± 0.001 | 0.751 ± 0.004 | 0.763 ± 0.004 | 0.774 ± 0.001 | 0.799 ± 0.002 | 0.748 ± 0.003 | 0.745 ± 0.003 |
| | | CW | 0.804 ± 0.002 | 0.804 ± 0.000 | 0.757 ± 0.004 | 0.775 ± 0.005 | 0.779 ± 0.001 | 0.805 ± 0.003 | 0.754 ± 0.003 | 0.752 ± 0.002 |
| | | NCE | 0.811 ± 0.002 | 0.812 ± 0.000 | 0.776 ± 0.002 | 0.794 ± 0.006 | 0.791 ± 0.001 | 0.809 ± 0.002 | 0.770 ± 0.002 | 0.767 ± 0.004 |
| | | elu margin | 0.801 ± 0.001 | 0.802 ± 0.001 | 0.750 ± 0.003 | 0.766 ± 0.005 | 0.773 ± 0.001 | 0.803 ± 0.002 | 0.746 ± 0.002 | 0.745 ± 0.001 |
| | | MCE | 0.784 ± 0.001 | 0.792 ± 0.000 | **0.713 ± 0.003** | 0.738 ± 0.004 | 0.754 ± 0.002 | 0.784 ± 0.003 | 0.722 ± 0.002 | 0.719 ± 0.004 |
| | | tanh margin | **0.767 ± 0.001** | **0.783 ± 0.001** | 0.717 ± 0.002 | **0.726 ± 0.001** | **0.737 ± 0.003** | **0.772 ± 0.002** | **0.720 ± 0.002** | **0.718 ± 0.005** |
| | $\epsilon = 0.1$ | CE | 0.753 ± 0.002 | **0.764 ± 0.001** | 0.736 ± 0.004 | 0.740 ± 0.002 | 0.749 ± 0.003 | 0.751 ± 0.001 | 0.735 ± 0.003 | 0.729 ± 0.002 |
| | | margin | 0.776 ± 0.001 | 0.780 ± 0.001 | 0.696 ± 0.004 | 0.708 ± 0.006 | 0.728 ± 0.001 | 0.780 ± 0.003 | 0.703 ± 0.004 | 0.691 ± 0.003 |
| | | CW | 0.792 ± 0.002 | 0.787 ± 0.002 | 0.709 ± 0.005 | 0.735 ± 0.007 | 0.744 ± 0.001 | 0.792 ± 0.003 | 0.710 ± 0.004 | 0.704 ± 0.003 |
| | | NCE | 0.793 ± 0.002 | 0.792 ± 0.001 | 0.731 ± 0.004 | 0.751 ± 0.007 | 0.760 ± 0.002 | 0.796 ± 0.003 | 0.731 ± 0.003 | 0.727 ± 0.005 |
| | | elu margin | 0.788 ± 0.002 | 0.783 ± 0.002 | 0.693 ± 0.004 | 0.717 ± 0.007 | 0.732 ± 0.002 | 0.791 ± 0.003 | 0.698 ± 0.003 | 0.695 ± 0.002 |
| | | MCE | 0.769 ± 0.002 | 0.778 ± 0.001 | **0.641 ± 0.003** | 0.672 ± 0.005 | 0.724 ± 0.003 | 0.773 ± 0.005 | **0.661 ± 0.002** | **0.654 ± 0.007** |
| | | tanh margin | **0.733 ± 0.001** | 0.765 ± 0.001 | 0.653 ± 0.002 | **0.665 ± 0.000** | **0.690 ± 0.003** | **0.744 ± 0.003** | 0.662 ± 0.003 | 0.660 ± 0.006 |
| | $\epsilon = 0.25$ | CE | 0.687 ± 0.000 | **0.709 ± 0.002** | 0.660 ± 0.003 | 0.665 ± 0.002 | 0.681 ± 0.002 | **0.687 ± 0.002** | 0.664 ± 0.002 | 0.657 ± 0.001 |
| | | margin | 0.729 ± 0.003 | 0.741 ± 0.002 | 0.555 ± 0.008 | 0.580 ± 0.006 | 0.648 ± 0.004 | 0.738 ± 0.005 | 0.586 ± 0.007 | 0.557 ± 0.003 |
| | | CW | 0.769 ± 0.003 | 0.765 ± 0.001 | 0.568 ± 0.010 | 0.625 ± 0.011 | 0.689 ± 0.003 | 0.777 ± 0.004 | 0.598 ± 0.006 | 0.577 ± 0.002 |
| | | NCE | 0.764 ± 0.003 | 0.758 ± 0.002 | 0.609 ± 0.008 | 0.637 ± 0.008 | 0.687 ± 0.002 | 0.771 ± 0.003 | 0.629 ± 0.006 | 0.603 ± 0.008 |
| | | elu margin | 0.756 ± 0.004 | 0.754 ± 0.002 | 0.535 ± 0.009 | 0.586 ± 0.008 | 0.664 ± 0.004 | 0.765 ± 0.003 | 0.576 ± 0.006 | 0.544 ± 0.001 |
| | | MCE | 0.750 ± 0.003 | 0.762 ± 0.001 | **0.509 ± 0.005** | 0.575 ± 0.009 | 0.683 ± 0.003 | 0.762 ± 0.005 | 0.557 ± 0.001 | **0.535 ± 0.015** |
| | | tanh margin | **0.679 ± 0.002** | 0.733 ± 0.002 | 0.541 ± 0.001 | **0.554 ± 0.001** | **0.610 ± 0.002** | 0.690 ± 0.005 | **0.551 ± 0.001** | 0.553 ± 0.007 |
| **PGD** | $\epsilon = 0.01$ | CE | 0.813 ± 0.002 | 0.814 ± 0.001 | 0.815 ± 0.004 | 0.824 ± 0.002 | 0.815 ± 0.001 | **0.810 ± 0.002** | 0.805 ± 0.003 | 0.805 ± 0.002 |
| | | margin | 0.820 ± 0.002 | 0.820 ± 0.001 | 0.816 ± 0.003 | 0.830 ± 0.003 | 0.819 ± 0.002 | 0.816 ± 0.002 | 0.806 ± 0.003 | 0.807 ± 0.002 |
| | | CW | 0.821 ± 0.002 | 0.820 ± 0.001 | 0.818 ± 0.003 | 0.833 ± 0.003 | 0.820 ± 0.001 | 0.817 ± 0.002 | 0.809 ± 0.003 | 0.810 ± 0.002 |
| | | NCE | 0.821 ± 0.001 | 0.820 ± 0.001 | 0.821 ± 0.003 | 0.834 ± 0.004 | 0.821 ± 0.001 | 0.818 ± 0.002 | 0.811 ± 0.003 | 0.812 ± 0.002 |
| | | elu margin | 0.820 ± 0.002 | 0.820 ± 0.001 | 0.816 ± 0.003 | 0.832 ± 0.003 | 0.819 ± 0.001 | 0.818 ± 0.002 | 0.807 ± 0.003 | 0.807 ± 0.002 |
| | | MCE | 0.817 ± 0.001 | 0.815 ± 0.001 | 0.808 ± 0.002 | 0.824 ± 0.004 | 0.813 ± 0.001 | 0.813 ± 0.002 | 0.800 ± 0.002 | 0.803 ± 0.001 |
| | | tanh margin | **0.809 ± 0.001** | **0.811 ± 0.001** | **0.800 ± 0.001** | **0.808 ± 0.002** | **0.800 ± 0.002** | 0.810 ± 0.002 | **0.792 ± 0.003** | **0.792 ± 0.000** |
| | $\epsilon = 0.05$ | CE | **0.775 ± 0.002** | **0.786 ± 0.001** | 0.767 ± 0.003 | 0.773 ± 0.002 | 0.771 ± 0.001 | **0.777 ± 0.002** | 0.761 ± 0.002 | 0.759 ± 0.001 |
| | | margin | 0.804 ± 0.001 | 0.808 ± 0.000 | 0.778 ± 0.003 | 0.790 ± 0.004 | 0.789 ± 0.002 | 0.803 ± 0.002 | 0.773 ± 0.003 | 0.773 ± 0.002 |
| | | CW | 0.810 ± 0.001 | 0.810 ± 0.000 | 0.784 ± 0.003 | 0.795 ± 0.004 | 0.798 ± 0.001 | 0.808 ± 0.002 | 0.779 ± 0.003 | 0.773 ± 0.003 |
| | | NCE | 0.814 ± 0.001 | 0.812 ± 0.001 | 0.796 ± 0.003 | 0.809 ± 0.004 | 0.806 ± 0.001 | 0.810 ± 0.002 | 0.790 ± 0.003 | 0.788 ± 0.003 |
| | | elu margin | 0.808 ± 0.001 | 0.809 ± 0.001 | 0.780 ± 0.003 | 0.797 ± 0.006 | 0.793 ± 0.001 | 0.806 ± 0.003 | 0.777 ± 0.002 | 0.775 ± 0.002 |
| | | MCE | 0.800 ± 0.001 | 0.801 ± 0.002 | 0.760 ± 0.004 | 0.776 ± 0.004 | 0.783 ± 0.002 | 0.801 ± 0.002 | 0.762 ± 0.003 | 0.762 ± 0.000 |
| | | tanh margin | 0.779 ± 0.002 | 0.787 ± 0.001 | **0.726 ± 0.004** | **0.740 ± 0.002** | **0.748 ± 0.002** | 0.782 ± 0.003 | **0.730 ± 0.003** | **0.725 ± 0.005** |
| | $\epsilon = 0.1$ | CE | **0.745 ± 0.002** | **0.762 ± 0.002** | 0.720 ± 0.003 | 0.724 ± 0.002 | 0.733 ± 0.002 | **0.748 ± 0.001** | 0.720 ± 0.003 | 0.719 ± 0.001 |
| | | margin | 0.786 ± 0.002 | 0.789 ± 0.001 | 0.733 ± 0.003 | 0.743 ± 0.006 | 0.753 ± 0.001 | 0.787 ± 0.002 | 0.730 ± 0.003 | 0.724 ± 0.004 |
| | | CW | 0.799 ± 0.001 | 0.799 ± 0.000 | 0.751 ± 0.003 | 0.767 ± 0.006 | 0.776 ± 0.001 | 0.796 ± 0.002 | 0.752 ± 0.003 | 0.745 ± 0.004 |
| | | NCE | 0.804 ± 0.001 | 0.805 ± 0.000 | 0.769 ± 0.004 | 0.786 ± 0.004 | 0.789 ± 0.002 | 0.801 ± 0.001 | 0.765 ± 0.003 | 0.758 ± 0.005 |
| | | elu margin | 0.802 ± 0.001 | 0.797 ± 0.001 | 0.742 ± 0.004 | 0.765 ± 0.006 | 0.767 ± 0.001 | 0.797 ± 0.002 | 0.743 ± 0.003 | 0.732 ± 0.004 |
| | | MCE | 0.792 ± 0.002 | 0.791 ± 0.001 | 0.736 ± 0.003 | 0.751 ± 0.005 | 0.775 ± 0.000 | 0.793 ± 0.002 | 0.737 ± 0.003 | 0.732 ± 0.002 |
| | | tanh margin | 0.758 ± 0.002 | 0.769 ± 0.001 | **0.662 ± 0.003** | **0.679 ± 0.002** | **0.704 ± 0.001** | 0.759 ± 0.003 | **0.673 ± 0.002** | **0.671 ± 0.007** |
| | $\epsilon = 0.25$ | CE | **0.684 ± 0.001** | **0.692 ± 0.001** | 0.618 ± 0.003 | 0.626 ± 0.002 | 0.641 ± 0.003 | **0.688 ± 0.002** | 0.624 ± 0.003 | 0.617 ± 0.001 |
| | | margin | 0.744 ± 0.003 | 0.748 ± 0.003 | 0.601 ± 0.007 | 0.638 ± 0.005 | 0.671 ± 0.003 | 0.752 ± 0.004 | 0.621 ± 0.004 | 0.599 ± 0.004 |
| | | CW | 0.779 ± 0.003 | 0.778 ± 0.002 | 0.666 ± 0.008 | 0.707 ± 0.007 | 0.744 ± 0.002 | 0.778 ± 0.004 | 0.681 ± 0.007 | 0.661 ± 0.006 |
| | | NCE | 0.774 ± 0.002 | 0.773 ± 0.000 | 0.667 ± 0.005 | 0.700 ± 0.007 | 0.731 ± 0.000 | 0.777 ± 0.002 | 0.679 ± 0.004 | 0.659 ± 0.006 |
| | | elu margin | 0.776 ± 0.003 | 0.774 ± 0.001 | 0.631 ± 0.007 | 0.679 ± 0.008 | 0.725 ± 0.002 | 0.775 ± 0.004 | 0.656 ± 0.005 | 0.635 ± 0.004 |
| | | MCE | 0.772 ± 0.003 | 0.777 ± 0.002 | 0.660 ± 0.006 | 0.694 ± 0.007 | 0.752 ± 0.002 | 0.778 ± 0.003 | 0.666 ± 0.005 | 0.664 ± 0.003 |
| | | tanh margin | 0.716 ± 0.004 | 0.739 ± 0.001 | **0.537 ± 0.003** | **0.567 ± 0.005** | **0.632 ± 0.002** | 0.727 ± 0.004 | **0.560 ± 0.001** | **0.547 ± 0.011** |

Table B.3: Adversarial accuracy comparing the conventional losses with our losses over the different architectures on Citeseer (transfer attack). $\epsilon$ denotes the fraction of edges perturbed.

| | | Architecture | Soft Median GDC | Soft Median PPRGo | Vanilla GCN | Vanilla GDC | Vanilla PPRGo | Soft Medoid GDC | Jaccard GCN | RGCN |
|---|---|---|---|---|---|---|---|---|---|---|
| **FGSM** | $\epsilon = 0.01$ | CE | 0.705 ± 0.002 | 0.712 ± 0.006 | 0.710 ± 0.002 | 0.699 ± 0.001 | 0.720 ± 0.005 | 0.705 ± 0.003 | 0.716 ± 0.004 | 0.681 ± 0.005 |
| | | margin | 0.708 ± 0.002 | 0.712 ± 0.006 | 0.704 ± 0.003 | 0.694 ± 0.002 | 0.720 ± 0.006 | 0.707 ± 0.003 | 0.712 ± 0.005 | 0.673 ± 0.005 |
| | | CW | 0.708 ± 0.002 | 0.712 ± 0.006 | 0.705 ± 0.003 | 0.694 ± 0.002 | 0.720 ± 0.006 | 0.707 ± 0.002 | 0.711 ± 0.005 | 0.673 ± 0.005 |
| | | NCE | 0.709 ± 0.002 | 0.714 ± 0.006 | 0.707 ± 0.003 | 0.696 ± 0.002 | 0.722 ± 0.006 | 0.708 ± 0.003 | 0.714 ± 0.005 | 0.675 ± 0.006 |
| | | elu margin | 0.708 ± 0.002 | 0.712 ± 0.006 | 0.704 ± 0.003 | 0.694 ± 0.002 | 0.719 ± 0.006 | 0.706 ± 0.003 | 0.711 ± 0.005 | 0.673 ± 0.005 |
| | | MCE | 0.703 ± 0.003 | 0.712 ± 0.006 | **0.695 ± 0.003** | 0.686 ± 0.001 | 0.715 ± 0.006 | 0.702 ± 0.002 | **0.707 ± 0.004** | **0.672 ± 0.004** |
| | | tanh margin | **0.702 ± 0.002** | **0.710 ± 0.006** | 0.698 ± 0.003 | **0.685 ± 0.000** | **0.710 ± 0.005** | **0.701 ± 0.003** | 0.708 ± 0.005 | 0.672 ± 0.004 |
| | $\epsilon = 0.05$ | CE | 0.688 ± 0.003 | 0.699 ± 0.006 | 0.681 ± 0.002 | 0.664 ± 0.002 | 0.698 ± 0.004 | 0.688 ± 0.003 | 0.693 ± 0.004 | 0.654 ± 0.004 |
| | | margin | 0.701 ± 0.002 | 0.701 ± 0.008 | 0.654 ± 0.004 | 0.639 ± 0.003 | 0.691 ± 0.008 | 0.700 ± 0.003 | 0.672 ± 0.007 | 0.622 ± 0.004 |
| | | CW | 0.702 ± 0.002 | 0.703 ± 0.008 | 0.658 ± 0.004 | 0.646 ± 0.002 | 0.692 ± 0.008 | 0.702 ± 0.004 | 0.677 ± 0.005 | 0.626 ± 0.004 |
| | | NCE | 0.706 ± 0.003 | 0.705 ± 0.008 | 0.673 ± 0.004 | 0.661 ± 0.001 | 0.699 ± 0.007 | 0.706 ± 0.003 | 0.687 ± 0.006 | 0.635 ± 0.005 |
| | | elu margin | 0.703 ± 0.002 | 0.702 ± 0.008 | 0.655 ± 0.004 | 0.642 ± 0.002 | 0.689 ± 0.008 | 0.703 ± 0.003 | 0.674 ± 0.005 | 0.624 ± 0.004 |
| | | MCE | 0.695 ± 0.002 | 0.697 ± 0.006 | **0.633 ± 0.003** | **0.622 ± 0.002** | 0.677 ± 0.008 | 0.694 ± 0.004 | **0.663 ± 0.004** | **0.622 ± 0.003** |
| | | tanh margin | **0.681 ± 0.002** | **0.693 ± 0.006** | 0.636 ± 0.003 | 0.630 ± 0.002 | **0.667 ± 0.005** | **0.685 ± 0.003** | 0.664 ± 0.005 | 0.622 ± 0.003 |
| | $\epsilon = 0.1$ | CE | 0.666 ± 0.003 | 0.684 ± 0.007 | 0.648 ± 0.002 | 0.631 ± 0.003 | 0.669 ± 0.005 | **0.670 ± 0.004** | 0.666 ± 0.003 | 0.621 ± 0.004 |
| | | margin | 0.688 ± 0.003 | 0.692 ± 0.007 | 0.600 ± 0.003 | 0.579 ± 0.004 | 0.648 ± 0.008 | 0.689 ± 0.004 | 0.632 ± 0.006 | 0.574 ± 0.003 |
| | | CW | 0.693 ± 0.003 | 0.694 ± 0.007 | 0.602 ± 0.003 | 0.589 ± 0.004 | 0.660 ± 0.008 | 0.695 ± 0.003 | 0.639 ± 0.005 | 0.581 ± 0.004 |
| | | NCE | 0.699 ± 0.002 | 0.697 ± 0.008 | 0.628 ± 0.003 | 0.621 ± 0.001 | 0.678 ± 0.008 | 0.700 ± 0.003 | 0.656 ± 0.005 | 0.593 ± 0.005 |
| | | elu margin | 0.693 ± 0.002 | 0.693 ± 0.008 | 0.596 ± 0.002 | 0.581 ± 0.004 | 0.655 ± 0.008 | 0.696 ± 0.003 | 0.634 ± 0.004 | 0.575 ± 0.003 |
| | | MCE | 0.676 ± 0.002 | 0.685 ± 0.007 | 0.574 ± 0.004 | **0.562 ± 0.003** | 0.644 ± 0.009 | 0.682 ± 0.003 | 0.622 ± 0.006 | **0.568 ± 0.005** |
| | | tanh margin | **0.663 ± 0.003** | **0.683 ± 0.006** | **0.569 ± 0.005** | 0.568 ± 0.004 | **0.621 ± 0.006** | 0.672 ± 0.004 | **0.609 ± 0.005** | 0.569 ± 0.006 |
| | $\epsilon = 0.25$ | CE | **0.616 ± 0.003** | **0.652 ± 0.007** | 0.570 ± 0.003 | 0.555 ± 0.005 | 0.614 ± 0.006 | **0.624 ± 0.005** | 0.606 ± 0.004 | 0.551 ± 0.003 |
| | | margin | 0.659 ± 0.004 | 0.663 ± 0.010 | 0.486 ± 0.001 | 0.463 ± 0.006 | 0.577 ± 0.008 | 0.668 ± 0.006 | 0.550 ± 0.000 | 0.473 ± 0.006 |
| | | CW | 0.682 ± 0.002 | 0.674 ± 0.009 | 0.500 ± 0.003 | 0.495 ± 0.014 | 0.615 ± 0.007 | 0.686 ± 0.003 | 0.569 ± 0.002 | 0.478 ± 0.003 |
| | | NCE | 0.681 ± 0.002 | 0.676 ± 0.008 | 0.521 ± 0.004 | 0.519 ± 0.012 | 0.613 ± 0.006 | 0.689 ± 0.003 | 0.585 ± 0.004 | 0.487 ± 0.004 |
| | | elu margin | 0.677 ± 0.002 | 0.670 ± 0.008 | 0.471 ± 0.004 | 0.460 ± 0.012 | 0.595 ± 0.008 | 0.685 ± 0.004 | 0.553 ± 0.002 | **0.455 ± 0.005** |
| | | MCE | 0.658 ± 0.002 | 0.671 ± 0.009 | 0.448 ± 0.003 | **0.439 ± 0.005** | 0.605 ± 0.009 | 0.675 ± 0.004 | 0.545 ± 0.003 | 0.461 ± 0.008 |
| | | tanh margin | 0.631 ± 0.002 | 0.666 ± 0.007 | **0.447 ± 0.005** | 0.456 ± 0.006 | **0.548 ± 0.005** | 0.649 ± 0.003 | **0.511 ± 0.002** | 0.462 ± 0.012 |
| **PGD** | $\epsilon = 0.01$ | CE | 0.705 ± 0.002 | 0.712 ± 0.006 | 0.710 ± 0.003 | 0.699 ± 0.001 | 0.720 ± 0.005 | **0.703 ± 0.002** | 0.714 ± 0.005 | 0.680 ± 0.005 |
| | | margin | 0.707 ± 0.002 | 0.712 ± 0.006 | 0.706 ± 0.004 | 0.694 ± 0.002 | 0.719 ± 0.006 | 0.707 ± 0.003 | 0.714 ± 0.006 | 0.675 ± 0.005 |
| | | CW | 0.708 ± 0.002 | 0.712 ± 0.006 | 0.708 ± 0.003 | 0.697 ± 0.001 | 0.721 ± 0.006 | 0.707 ± 0.003 | 0.715 ± 0.005 | 0.677 ± 0.005 |
| | | NCE | 0.708 ± 0.002 | 0.714 ± 0.006 | 0.709 ± 0.003 | 0.700 ± 0.002 | 0.723 ± 0.006 | 0.708 ± 0.003 | 0.716 ± 0.005 | 0.679 ± 0.006 |
| | | elu margin | 0.708 ± 0.002 | 0.713 ± 0.006 | 0.707 ± 0.003 | 0.696 ± 0.002 | 0.720 ± 0.006 | 0.707 ± 0.003 | 0.714 ± 0.005 | 0.677 ± 0.006 |
| | | MCE | 0.706 ± 0.002 | 0.712 ± 0.006 | 0.704 ± 0.003 | 0.694 ± 0.000 | 0.720 ± 0.006 | 0.706 ± 0.003 | 0.713 ± 0.004 | 0.675 ± 0.005 |
| | | tanh margin | **0.703 ± 0.002** | **0.711 ± 0.006** | **0.696 ± 0.003** | **0.685 ± 0.000** | **0.712 ± 0.006** | 0.703 ± 0.003 | **0.706 ± 0.005** | **0.670 ± 0.005** |
| | $\epsilon = 0.05$ | CE | 0.689 ± 0.002 | **0.697 ± 0.007** | 0.677 ± 0.002 | 0.661 ± 0.002 | 0.695 ± 0.005 | 0.689 ± 0.005 | 0.688 ± 0.004 | 0.647 ± 0.003 |
| | | margin | 0.702 ± 0.002 | 0.702 ± 0.007 | 0.670 ± 0.003 | 0.654 ± 0.002 | 0.693 ± 0.007 | 0.701 ± 0.003 | 0.688 ± 0.004 | 0.642 ± 0.004 |
| | | CW | 0.704 ± 0.002 | 0.707 ± 0.006 | 0.676 ± 0.002 | 0.661 ± 0.001 | 0.707 ± 0.007 | 0.702 ± 0.002 | 0.693 ± 0.004 | 0.646 ± 0.005 |
| | | NCE | 0.706 ± 0.002 | 0.708 ± 0.007 | 0.686 ± 0.002 | 0.676 ± 0.001 | 0.708 ± 0.005 | 0.705 ± 0.003 | 0.700 ± 0.004 | 0.654 ± 0.005 |
| | | elu margin | 0.704 ± 0.002 | 0.703 ± 0.007 | 0.678 ± 0.001 | 0.660 ± 0.002 | 0.698 ± 0.005 | 0.705 ± 0.002 | 0.693 ± 0.004 | 0.649 ± 0.005 |
| | | MCE | 0.696 ± 0.001 | 0.702 ± 0.006 | 0.669 ± 0.002 | 0.645 ± 0.003 | 0.700 ± 0.006 | 0.698 ± 0.003 | 0.688 ± 0.004 | 0.639 ± 0.006 |
| | | tanh margin | **0.686 ± 0.002** | 0.700 ± 0.005 | **0.643 ± 0.002** | **0.627 ± 0.001** | **0.675 ± 0.004** | **0.688 ± 0.002** | **0.666 ± 0.004** | **0.629 ± 0.003** |
| | $\epsilon = 0.1$ | CE | **0.663 ± 0.001** | **0.681 ± 0.006** | 0.643 ± 0.003 | 0.622 ± 0.001 | 0.664 ± 0.005 | **0.665 ± 0.004** | 0.662 ± 0.003 | 0.621 ± 0.002 |
| | | margin | 0.691 ± 0.002 | 0.696 ± 0.008 | 0.634 ± 0.006 | 0.615 ± 0.002 | 0.668 ± 0.008 | 0.690 ± 0.004 | 0.661 ± 0.006 | 0.606 ± 0.004 |
| | | CW | 0.697 ± 0.001 | 0.701 ± 0.007 | 0.654 ± 0.002 | 0.635 ± 0.005 | 0.691 ± 0.005 | 0.698 ± 0.003 | 0.676 ± 0.005 | 0.620 ± 0.005 |
| | | NCE | 0.697 ± 0.002 | 0.702 ± 0.008 | 0.663 ± 0.002 | 0.647 ± 0.004 | 0.691 ± 0.007 | 0.700 ± 0.002 | 0.681 ± 0.005 | 0.624 ± 0.006 |
| | | elu margin | 0.700 ± 0.002 | 0.699 ± 0.008 | 0.646 ± 0.002 | 0.626 ± 0.003 | 0.684 ± 0.006 | 0.699 ± 0.003 | 0.670 ± 0.004 | 0.612 ± 0.006 |
| | | MCE | 0.687 ± 0.001 | 0.691 ± 0.007 | 0.641 ± 0.002 | 0.605 ± 0.006 | 0.684 ± 0.006 | 0.690 ± 0.003 | 0.666 ± 0.005 | 0.614 ± 0.005 |
| | | tanh margin | 0.675 ± 0.002 | 0.692 ± 0.007 | **0.594 ± 0.002** | **0.581 ± 0.003** | **0.649 ± 0.003** | 0.677 ± 0.003 | **0.630 ± 0.003** | **0.589 ± 0.004** |
| | $\epsilon = 0.25$ | CE | **0.617 ± 0.005** | **0.653 ± 0.008** | 0.565 ± 0.003 | 0.544 ± 0.001 | 0.605 ± 0.008 | **0.627 ± 0.006** | 0.594 ± 0.004 | 0.550 ± 0.002 |
| | | margin | 0.660 ± 0.004 | 0.671 ± 0.009 | 0.543 ± 0.000 | 0.512 ± 0.004 | 0.610 ± 0.006 | 0.670 ± 0.005 | 0.593 ± 0.004 | 0.522 ± 0.005 |
| | | CW | 0.682 ± 0.002 | 0.685 ± 0.009 | 0.597 ± 0.006 | 0.575 ± 0.010 | 0.670 ± 0.006 | 0.687 ± 0.004 | 0.636 ± 0.005 | 0.557 ± 0.005 |
| | | NCE | 0.683 ± 0.001 | 0.681 ± 0.008 | 0.581 ± 0.002 | 0.571 ± 0.007 | 0.651 ± 0.006 | 0.688 ± 0.003 | 0.623 ± 0.002 | 0.541 ± 0.007 |
| | | elu margin | 0.681 ± 0.002 | 0.681 ± 0.009 | 0.560 ± 0.005 | 0.541 ± 0.010 | 0.650 ± 0.007 | 0.687 ± 0.003 | 0.612 ± 0.003 | 0.537 ± 0.005 |
| | | MCE | 0.670 ± 0.004 | 0.681 ± 0.008 | 0.581 ± 0.002 | 0.548 ± 0.007 | 0.665 ± 0.007 | 0.677 ± 0.006 | 0.624 ± 0.004 | 0.552 ± 0.004 |
| | | tanh margin | 0.649 ± 0.002 | 0.671 ± 0.006 | **0.496 ± 0.003** | **0.486 ± 0.002** | **0.590 ± 0.007** | 0.658 ± 0.004 | **0.553 ± 0.004** | **0.497 ± 0.006** |

## B.4 Proof of Proposition 1

**Proposition 1** *Let $\mathcal{L}'$ be the surrogate for the 0/1 loss $\mathcal{L}_{0/1}$ used to attack a node classification algorithm $f_\theta(\boldsymbol{A}, \boldsymbol{X})$ with a global budget $\Delta$. Suppose we greedily attack nodes in order of $\partial\mathcal{L}'/\partial z_{c^*}(\psi_0) \leq \partial\mathcal{L}'/\partial z_{c^*}(\psi_1) \leq \cdots \leq \partial\mathcal{L}'/\partial z_{c^*}(\psi_l)$ until the budget is exhausted $\Delta < \sum_{i=0}^{l+1} \Delta_i$. Under Assumptions 1 & 2, we then obtain the global optimum of $\max_{\tilde{\boldsymbol{A}} \text{ s.t. } \|\tilde{\boldsymbol{A}} - \boldsymbol{A}\|_0 < \Delta} \mathcal{L}_{0/1}(f_\theta(\tilde{\boldsymbol{A}}, \boldsymbol{X}))$ if $\mathcal{L}'$ has the properties* **(I)** $\partial\mathcal{L}'/\partial z_{c^*}|_{\psi<0} = 0$ *and* **(II)** $\partial\mathcal{L}'/\partial z_{c^*}|_{\psi_0} < \partial\mathcal{L}'/\partial z_{c^*}|_{\psi_1}$ *for any $0 < \psi_0 < \psi_1$.*

We can easily see that the greedy algorithm obeying the attack order $0 \leq \psi_0 \leq \psi_1 \leq \cdots \leq \psi_u \leq \ldots$ obtains the optimal solution by an exchange argument. Since $g(\psi_i)$ is strictly increasing and does not change the order, we can simply omit it. Let us suppose we are given the optimal plan $\rho^*$ and the greedy solution has the plan $\rho$. Suppose $\rho^*$ would contain one or more nodes for that $w > u$ instead of $q \leq u$. We know that $\psi_w \geq \psi_q$ and hence $\Delta_w \geq \Delta_q$. Thus, replacing $w$ by $q$ would either lead to the an equally good or even better solution (contradiction!). Hence, the greedy plan $\rho$ is at least as good as the optimal plan $\rho^*$. The solution is unique except for ties s.t. $0 \leq \psi_i = \psi_j \leq \psi_u$.

Consequently, a surrogate loss $\mathcal{L}'$ that leads to the order above will yield the global optimum as well. The order is preserved if (compare with Definition 1):

(I) $\partial\mathcal{L}'/\partial z_{c*}|_{\psi<0} = 0$

(II) $\partial\mathcal{L}'/\partial z_{c*}|_{\psi_0} < \partial\mathcal{L}'/\partial z_{c*}|_{\psi_1}$ for any $0 < \psi_0 < \psi_1$ (i.e. $\partial\mathcal{L}'/\partial z_{c*}$ is strictly concave for positive inputs)

From property (II) follows that $\partial\mathcal{L}'/\partial z_{c*}$ is minimal for $\psi \to 0^+$. $\square$

### B.5 Alternative Version of Proposition 1

Here we state an alternative version of Proposition 1 if we relax Assumption 2 s.t. it only needs to hold in expectation.

**Assumption 2** The *expected* budget required to change the prediction of node $i$ increases with the margin: $\mathbb{E}[\Delta_i|\psi_i] = g(|\psi_i|)$ for some increasing function $g(|\psi_i|) \geq 1$.

**Proposition 1** Let $\mathcal{L}'$ be the surrogate for the 0/1 loss $\mathcal{L}_{0/1}$ used to attack a node classification algorithm $f_\theta(\mathbf{A}, \mathbf{X})$ with a global budget $\Delta$. Additionally to Assumptions 1 and 2, suppose the adversary perturbs the chosen node until it is misclassified. We then obtain the global optimum of

$$\max_{\tilde{\mathbf{A}} \text{ s.t. } \|\tilde{\mathbf{A}}-\mathbf{A}\|_0 \leq \Delta} \mathbb{E}[\mathcal{L}_{0/1}(f_\theta(\tilde{\mathbf{A}}, \mathbf{X}))]$$

through greedily attacking the nodes in order $\frac{\partial\mathcal{L}'}{\partial\mathbf{z}_{c*}}(\psi_0) \leq \frac{\partial\mathcal{L}'}{\partial\mathbf{z}_{c*}}(\psi_1) \leq \cdots \leq \frac{\partial\mathcal{L}'}{\partial\mathbf{z}_{c*}}(\psi_l)$ until the budget is exhausted $\Delta \leq \sum_{i=0}^{l+1} \Delta_i$, if $\mathcal{L}'$ has the properties **(I)** $\frac{\partial\mathcal{L}'}{\partial\mathbf{z}_{c*}}|_{\psi<0} = 0$ and **(II)** $\frac{\partial\mathcal{L}'}{\partial\mathbf{z}_{c*}}|_{\psi_0} < \frac{\partial\mathcal{L}'}{\partial\mathbf{z}_{c*}}|_{\psi_1}$ for any $0 \leq \psi_0 < \psi_1$.

Assumption 2 only needs to hold for a small fraction of nodes with low $\psi_i$. For the empirical distribution of a two-layer GCN on Cora ML, $\mathbb{E}[\Delta_i|\psi_i] = 1$ and $\text{Var}[\Delta_i|\psi_i] = 0$ for the 22.9% nodes with lowest margin $\psi_i$. Hence,

$$\max_{\tilde{\mathbf{A}} \text{ s.t. } \|\tilde{\mathbf{A}}-\mathbf{A}\|_0 \leq \Delta} \mathbb{E}[\mathcal{L}_{0/1}(f_\theta(\tilde{\mathbf{A}}, \mathbf{X}))] \approx \max_{\tilde{\mathbf{A}} \text{ s.t. } \|\tilde{\mathbf{A}}-\mathbf{A}\|_0 \leq \Delta} \mathcal{L}_{0/1}(f_\theta(\tilde{\mathbf{A}}, \mathbf{X}))$$

for small $\Delta$.

## C Scalable Attacks

We start with some general remarks on $L_0$ Projected Gradient Descent in § C.1. Then we give more details on our attacks PR-BCD and GR-BCD (§ C.2). In § C.3, we conclude this section with the derivation and complexity of the update of PPR scores (required for attacking PPRGo).

### C.1 $L_0$ Projected Gradient Descent

For $L_0$ Projected Gradient Descent ($L_0$-PGD) we largely follow Xu et al. [43]. In fact, theirs is a special case of our PR-BCD (with the exceptions detailed below). To recover the ($L_0$-PGD), one solely needs to select all possible indices in line 3 in Algo. 1 and drop lines 10-14.

As discussed in § 3, we aim to solve:

$$\max_{\mathbf{P} \text{ s.t. } \mathbf{P} \in \{0,1\}^{n\times n}, \sum \mathbf{P} \leq \Delta} \mathcal{L}(f_\theta(\mathbf{A} \oplus \mathbf{P}, \mathbf{X})). \tag{C.1}$$

where we explicitly model the perturbations $\mathbf{P} \in \{0,1\}^{n\times n}$ ($\mathbf{P}_{ij} = 1$ denotes an edge flip). For the sake of optimizing $\mathbf{P}$ with first-order/gradient methods we relax it from $\{0,1\}^{(n\times n)}$ to $[0,1]^{(n\times n)}$. In words, during the attack we allow a *weighted adjacency matrix* where the weights of $\mathbf{P}$ at the same time represent the probability to flip this edge in the last step of the attack. The sampling $\mathbf{P} \sim \text{Bernoulli}(\mathbf{p}_t)$ s.t. $\sum \mathbf{P} \leq \Delta$ is required to obtain a binary perturbed adjacency matrix in the end: $\tilde{\mathbf{A}} \in \{0,1\}^{(n\times n)}$. Note that we overload $\oplus$ (besides its binary XOR meaning) s.t. $\mathbf{A}_{ij} \oplus p_{ij} = \mathbf{A}_{ij} + p_{ij}$ if $\mathbf{A}_{ij} = 0$ and $\mathbf{A}_{ij} - p_{ij}$ otherwise.

**Projection.** Recall that after each gradient update, the projection $\Pi_{\mathbb{E}[\text{Bernoulli}(\mathbf{p})]\leq\Delta}(\mathbf{p})$ adjusts the probability mass such that $\mathbb{E}[\text{Bernoulli}(\mathbf{p})] = \sum_{i\in b} p_i \leq \Delta$ and that $\mathbf{p} \in [0,1]$. Specifically, the

projection operation

$$\Pi_{\mathbb{E}[\text{Bernoulli}(\boldsymbol{p})] \leq \Delta}(\boldsymbol{p}) = \begin{cases} \Pi_{[0,1]}(\boldsymbol{p}) & \text{if } \mathbf{1}^\top \Pi_{[0,1]}(\boldsymbol{p}) \leq \Delta \\ \Pi_{[0,1]}(\boldsymbol{p} - \lambda\mathbf{1}) \text{ s.t. } \mathbf{1}^\top \Pi_{[0,1]}(\boldsymbol{p} - \lambda\mathbf{1}) = \Delta & \text{otherwise} \end{cases} \quad \text{(C.2)}$$

where $\Pi_{[0,1]}(\boldsymbol{p})$ is simply clamping the values to the interval $[0,1]$ and $\lambda$ originates from the Lagrange formulation of the constrained optimization problem. $\lambda$ can be efficiently calculated with the bisection method in $\log_2\left[{}^{\max(\boldsymbol{p})-\min(\boldsymbol{p}-1)}/\xi\right]$ steps with the admissible error $\xi$. In contrast to Xu et al. [43], we additionally limit the number of steps to account for numerical instabilities on very large graphs.

**Sampling solution.** To retrieve a discrete and valid perturbed adjacency matrix in the last step, we sample $\boldsymbol{P} \sim \text{Bernoulli}(\boldsymbol{p}_t)$ s.t. $\sum \boldsymbol{P} \leq \Delta$. Xu et al. [43] propose to sample for 20 times and reject all samples that violate the constraint. To eliminate the case that no solution was found and for improved attack strength (at the cost of a potential bias), we take the top-$\Delta$ values of $\boldsymbol{p}$ instead of sampling in the first iteration of this "rejection sampling" procedure. In case of ties, we take the preceding sample.

**Learning rate.** To obtain a constant learning rate regardless of the budget, we scale a "base" learning rate (hyperparameter) by the budget. When using the PR-BCD attack (which we discuss next), we use the block size $b/n^2$ as an additional scaling factor and then apply the square root.

## C.2 Projected and Greedy Randomized Block Coordinate Descent

We first give some implementation details on our Projected Randomized Block Coordinate Descent (PR-BCD). Then we formally introduce Greedy Randomized Block Coordinate Descent (GR-BCD).

**Sampling w/o replacement.** As it turns out, even sampling w/o replacement $\boldsymbol{i}_0 \in \{0, 1, \dots, n^2-1\}^b$, which we need to determine the current block, is not easily parallelizable if one just has $\mathcal{O}(b)$ memory and, hence, rather slow on modern GPUs. For this reason, we simply sample with replacement and afterward drop the duplicates. This comes at the cost of not having a block with exactly $b$ elements. Especially on large graphs, the difference is rather small, although, collisions do exist. For a proper analysis, we refer to well-studied problems such as the Birthday Paradox or hash sets/tables.

**Representing zeros.** We require that all elements in $\boldsymbol{p}$ have a small, negligible non-zero value, i.e. must not be exactly zero. This affects the initialization and the projection procedure. We require it for two reasons: (1) we can easily "detect" edge removals ($p_i$ must be subtracted instead of added) and (2) some sparse operations implicitly remove edges of zero weight.

**GR-BCD.** The biggest pitfall while aiming for maximum scalability is that we do not desire a runtime of $\mathbf{O}(m)$. Instead, we solely want to iterate a constant number of steps (epochs) $E$. We simply achieve this through defining a schedule $\Delta_t$ for $t \in \{1, 2, \dots, E\}$ where $\sum_{t=1}^{E} \Delta_t = \Delta$. In our experiments, we distribute the budget evenly and leave more complicated alternatives for future work. For the pseudo code of GR-BCD see Algorithm 2.

**Advantages and limitations.** Our GR-BCD shares many commonalities with PR-BCD but does not require a learning rate $\alpha_t$, heuristic $h(\dots)$, and $E_{\text{res.}}$ since we resample in each epoch. Another advantage is that we do not require $b > \Delta$, which makes it more scalable than PR-BCD. However, since PR-BCD is scalable itself, in our experiments, we always kept the same block sizes for both attacks for improved comparability of results. GR-BCD's biggest drawback is its much slower learning dynamics. That is if an edge is flipped it is rarely flipped back. This is particularly important if one does not attack a GCN / designs an adaptive attack (see § F.3).

## C.3 Derivation and Complexity of Personalized Page Rank Update

In the following, we discuss how we can attack a single node on PPRGo using PR-BCD (i.e. a local attack). Since PPRGo avoids a recursive message passing scheme, relying on the PPR scores, we need an efficient, differentiable procedure to update the PPR scores given the perturbation of the adjacency matrix. We further limit the perturbations to the incoming edges. Perturbing adjacent edges is the most effective attack [50]. To update the PPR scores of a directed graph for a node in $\boldsymbol{\Pi}$, we use the Sherman-Morrison formula

$$(\boldsymbol{B} + \boldsymbol{u}\boldsymbol{v}^\top)^{-1} = \boldsymbol{B}^{-1} - \frac{\boldsymbol{B}^{-1}\boldsymbol{u}\boldsymbol{v}^\top\boldsymbol{B}^{-1}}{1 + \boldsymbol{v}^\top\boldsymbol{B}^{-1}\boldsymbol{u}} \quad \text{(C.3)}$$

---

**Algorithm 2** Greedy Randomized Block Coordinate Descent (GR-BCD)

---
1: **Input:** Gr. $(\boldsymbol{A}, \boldsymbol{X})$, lab. $\boldsymbol{y}$, GNN $f_\theta(\cdot)$, loss $\mathcal{L}$
2: **Parameter:** block size $b$, schedule $\Delta_t$ for $t \in \{1, 2, \ldots, E\}$
3: Draw w/o replacement $\boldsymbol{i}_0 \in \{0, 1, \ldots, n^2 - 1\}^b$
4: Initialize zeros for $\boldsymbol{p}_0 \in \mathbb{R}^b$
5: initialize $\hat{\boldsymbol{A}} \leftarrow \boldsymbol{A}$
6: **for** $t \in \{1, 2, \ldots, E\}$ **do**
7: $\quad \hat{\boldsymbol{y}} \leftarrow f_\theta(\hat{\boldsymbol{A}} \oplus \boldsymbol{p}_{t-1}, \boldsymbol{X})$
8: $\quad$ Flip arg top-$\Delta_t(\nabla_{\boldsymbol{i}_{t-1}} \mathcal{L}(\hat{\boldsymbol{y}}, \boldsymbol{y}))$ edges in $\hat{\boldsymbol{A}}$
9: $\quad$ mask$_{\text{res.}} \leftarrow h(\boldsymbol{p}_t)$
10: $\quad \boldsymbol{p}_t[\text{mask}_{\text{res.}}] \leftarrow \boldsymbol{0}$
11: $\quad$ Resample $\boldsymbol{i}_t[\text{mask}_{\text{res.}}]$
12: Return $\hat{\boldsymbol{A}}$

---

for rank one update $\boldsymbol{u}\boldsymbol{v}^\top$ of the inverse of an invertible matrix $\boldsymbol{B} \in \mathbb{R}^{n \times n}$. The rank one $\boldsymbol{u}\boldsymbol{v}^\top$ update in general has shape $[n \times n]$ and therefore comes with space complexity $\mathcal{O}(n^2)$ and the update via the Sherman-Morrison formula has $\mathcal{O}(n^3)$. Since we use row normalization with PPRGo, we can attack the PPR scores via updating a single row $\tilde{\boldsymbol{\Pi}}_i$ of the adjacency matrix $\boldsymbol{A}$ (including normalization) and obtain the gradient for the $b$ potentially non-zero entries in $\boldsymbol{p}$.

We can write the closed-form local PPR update as:

$$\tilde{\boldsymbol{\Pi}}_i = \alpha \left[ \boldsymbol{I} - (1 - \alpha) \boldsymbol{D}^{-1} \boldsymbol{A} + \boldsymbol{u}\boldsymbol{v}^\top \right]_i^{-1} = \alpha \left( \boldsymbol{\Pi}'_i - \frac{\boldsymbol{\Pi}'_{ii} \boldsymbol{v} \boldsymbol{\Pi}'}{1 + \boldsymbol{v}\boldsymbol{\Pi}'_{:i}} \right) \tag{C.4}$$

where $\boldsymbol{\Pi}' = (\boldsymbol{I} - (1 - \alpha)\boldsymbol{D}^{-1}\boldsymbol{A})^{-1} = \alpha^{-1}\boldsymbol{\Pi}$ and we choose $u_j = 0 \, \forall j \neq i$ and $u_i = 1$. For PPR we need e.g. a row stochastic matrix and hence need to normalize the adjacency matrix, also accounting for the prospective update. This implies that through an alteration of $b$ entries in the $i$-th row of the *unnormalized* adjacency matrix, we need to adjust every entry of this row to obtain the *normalized* adjacency matrix. We can simply achieve this through adding the normalized row $(\boldsymbol{D}_{ii} + \sum \boldsymbol{p})^{-1}(\boldsymbol{A}_i + \boldsymbol{p})$ after the alteration and subtract its original entries $\boldsymbol{D}_{ii}^{-1}\boldsymbol{A}_i$. Putting this together, the rank one update of the $i$-th row results in $\boldsymbol{v} = (\boldsymbol{D}_{ii} + \sum \boldsymbol{p})^{-1}(\boldsymbol{A}_i + \boldsymbol{p}) - \boldsymbol{D}_{ii}^{-1}\boldsymbol{A}_i$ where $\boldsymbol{p}$ is a sparse vector with at most $b$ non-zero elements.

With dense matrices, this would leave us with a complexity of $\mathcal{O}(bn)$ due to the vector-matrix product $\boldsymbol{v}\boldsymbol{\Pi}'$. We follow Bojchevski et al. [5] and use the top-k-sparsified PPR $\boldsymbol{\Pi}^{(k)}$ instead of $\boldsymbol{\Pi}$ with at most $k$ entries per row. Since $\boldsymbol{v}$ has at most $b$ non-zero entries, most columns in the slice $\boldsymbol{\Pi}^{(k)}_{\boldsymbol{v} \neq 0}$ only contain zero elements. Thus, we can equivalently write $\boldsymbol{v}\boldsymbol{\Pi}'$ as a dense vector matrix product of shapes $[1, b]$ and $[b, r]$, where r is the number of non-zero columns in the rows $\boldsymbol{\Pi}'_{\boldsymbol{i}_t}$. Recall that $\boldsymbol{i}_t$ are the indices of epoch $t$ and that $b \geq |\boldsymbol{i}_t|$. If we assume randomly distributed ones, the probability of a non-zero entry is $k/n$. Hence we can model $P(\sum \boldsymbol{\Pi}^{(k)}_{\boldsymbol{v} \neq 0, j}) = \text{Bin}(b, k/n)$ for column $j$ and analogously

$$\begin{aligned}
\mathbb{E}[r] &= n \cdot P\left( \sum \boldsymbol{\Pi}^{(k)}_{\boldsymbol{v} \neq 0, j} > 0 \right) \\
&= n \left[ 1 - P\left( \sum \boldsymbol{\Pi}^{(k)}_{\boldsymbol{v} \neq 0, j} = 0 \right) \right] \\
&= n \left[ 1 - \left( 1 - \frac{k}{n} \right)^b \right] \\
&= \frac{n^b - (n - k)^b}{n^{b-1}} .
\end{aligned} \tag{C.5}$$

For an appropriate choice of $k \ll n$, the expected number of non-zero rows is $\mathbb{E}[r] = \mathcal{O}(bk)$. The stronger asymptotic relation $\mathbb{E}[r] = \Theta(bk)$ (and more strict alternatives) holds, but we omit this discussion for simplicity. Instead, we refer to Fig. C.1 for an illustration. In summary, the complexity of $\boldsymbol{v}\boldsymbol{\Pi}'$ and our local attack on PPRGo is $\mathcal{O}(bk)$. Please note that in contrast to the global PR-BCD attack, this *includes* the (Soft Median) PPRGo, and therefore is much more scalable. In practice, we observed slightly lower values for $r$ than predicted by the relation above. We hypothesize that this is

due to the fact that many rows contain less than $k$ non-zero elements (depending on the approximation of the PPR scores) but that our assumption of randomly distributed non-zero elements holds.

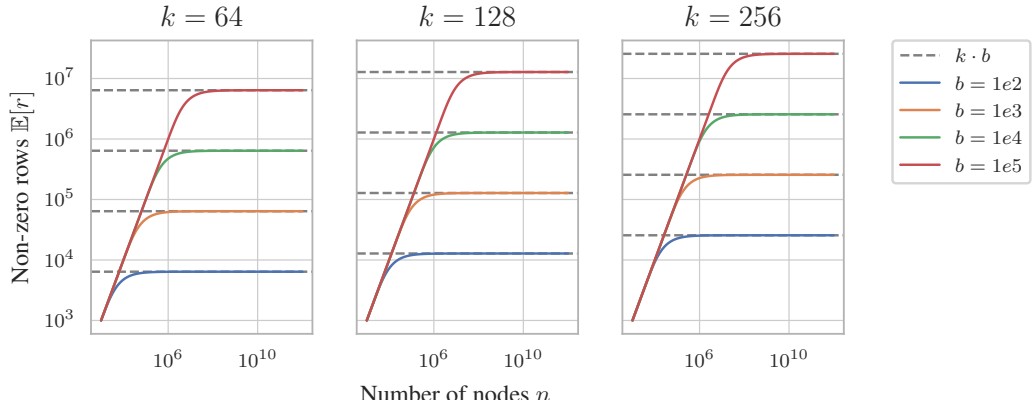

Figure C.1: $\mathbb{E}[r] = \frac{n^b - (n-k)^b}{n^{b-1}}$ for different (practical) values of $k$ and $b$.

To obtain a local attack we simply need to change lines 3 and 13 of Algo. 1 to sample only indices $i_0 \in \{0, 1, \ldots, n-1\}$. Further, we either keep line 6 if we attack e.g. GCN [24] or update $\tilde{\mathbf{\Pi}}_i$ as described in Eq. 4. We simply use a margin loss in logit space since we only have a local budget. After the attack and before applying the victim model the last time, we recalculate the PPR score for the target node based on the perturbed graph structure. The difference between the margin in the best epoch and after recalculating the PPR scores is usually negligible and shows that the approximation holds.

## D   Scalable Defense

We first formally define the breakdown point. Then in § D.1, we give the proof of Theorem 1 and in § D.2 we extend the discussion to the weighted Soft Median.

**Breakdown point.** Many metrics have been proposed that capture the robustness of a point estimate / aggregation with different flavors. One of the most widely used properties is the breakdown point. The (finite-sample) breakdown point captures the minimal fraction $\epsilon = m/n$ so that the result of the location estimator $\mu(\mathbf{X})$ can be arbitrarily placed [15]:

$$\epsilon^*(t, \mathbf{X}) = \min_{1 \leq m \leq n} \left\{ \frac{m}{n} : \sup_{\tilde{\mathbf{X}}_\epsilon} \|\mu(\mathbf{X}) - \mu(\tilde{\mathbf{X}}_\epsilon)\| = \infty \right\} \tag{D.1}$$

In this section we use $m$ and $n$ differently than in the rest of the paper: $m$ denotes the number of perturbed inputs and $n$ the number of inputs of the aggregation $\mu(\mathbf{X})$ (or number of rows in $\mathbf{X}$).

### D.1   Proof of Theorem 1

**Theorem 1** *Let* $\mathbb{X} = \{\mathbf{x}_1, \ldots, \mathbf{x}_n\}$ *be a collection of points in* $\mathbb{R}^d$ *with finite coordinates and temperature* $T \in [0, \infty)$. *Then the Soft Median location estimator (Eq. 6) has the finite sample breakdown point of* $\epsilon^*(\mu_{Soft\ Median}, \mathbf{X}) = 1/n \lfloor (n+1)/2 \rfloor$ *(asympt.* $\lim_{n \to \infty} \epsilon^*(\mu_{SoftMedian}, \mathbf{X}) = 0.5$).

Let $\tilde{\mathbb{X}}_\epsilon$ be decomposable such that $\tilde{\mathbb{X}}_\epsilon = \tilde{\mathbb{X}}_\epsilon^{(c)} \cup \tilde{\mathbb{X}}_\epsilon^{(p)}$ and $\tilde{\mathbb{X}}_\epsilon^{(c)} \cap \tilde{\mathbb{X}}_\epsilon^{(p)} = \emptyset$. Here $\tilde{\mathbb{X}}_\epsilon^{(c)}$ denotes the clean and $\tilde{\mathbb{X}}_\epsilon^{(p)}$ the perturbed inputs. We now have to find the minimal fraction of outliers $\epsilon$ for which $\sup_{\tilde{\mathbf{X}}_\epsilon} \|\mu_{SoftMedian}(\tilde{\mathbf{X}}_\epsilon)\| < \infty$ does not hold anymore. According to Eq. D.1, if we now want to arbitrarily perturb the Soft Median, we must $\|\tilde{\mathbf{x}}_v\| \to \infty$, $\exists v \in \tilde{\mathbb{X}}_\epsilon^{(p)}$. Next we analyze the influence of this instance on Eq. 6 (w.l.o.g. we omit the factor $\sqrt{D}$):

$$\hat{\mathbf{s}}_v \tilde{\mathbf{x}}_v = \frac{\exp\left\{-\frac{1}{T}\|\bar{\mathbf{x}} - \tilde{\mathbf{x}}_v\|\right\} \tilde{\mathbf{x}}_v}{\sum_{i \in \tilde{\mathbb{X}}_\epsilon^{(c)}} \exp\left\{-\frac{1}{T}\|\bar{\mathbf{x}} - \mathbf{x}_i\|\right\} + \sum_{j \in \tilde{\mathbb{X}}_\epsilon^{(p)}} \exp\left\{-\frac{1}{T}\|\bar{\mathbf{x}} - \mathbf{x}_j\|\right\}}$$

Instead of $\lim_{\|\tilde{\boldsymbol{x}}_v\|\to\infty}\hat{\boldsymbol{s}}_v\boldsymbol{x}_v$, we can equivalently derive the limit for the numerator and the denominator independently, as long as the denominator does not approach 0 which is easy to show (the denominator is $> 0$ and $\leq |\tilde{\mathbb{X}}_\epsilon|$). With $\lim_{\|\tilde{\boldsymbol{x}}_v\|\to\infty}$ we denote the fact that the statements holds regardless how we achieve that the norm approaches infinity:

$$\lim_{\|\tilde{\boldsymbol{x}}_v\|\to\infty}\left\|\exp\left\{-\frac{1}{T}\|\bar{\boldsymbol{x}}-\tilde{\boldsymbol{x}}_v\|\right\}\tilde{\boldsymbol{x}}_v\right\| = \begin{cases} 0, \text{ if } \lim_{\|\tilde{\boldsymbol{x}}_v\|\to\infty}\|\bar{\boldsymbol{x}}-\tilde{\boldsymbol{x}}_v\| = 0 \\ \infty, \text{ otherwise} \end{cases}$$

Please note that $\lim_{x\to\infty}xe^{-x/a}=0$ for $a\in[0,\infty)$. As long as $\epsilon < 0.5$, we know that for each dimension the perturbed dimension-wise median must be still within the range of the clean points. Or in other words, the perturbed median lays within the smallest possible hypercube around the original clean data $\mathbb{X}$. As long as $\epsilon < 0.5$ we have that $\lim_{\|\tilde{\boldsymbol{x}}_v\|\to\infty}\|\bar{\boldsymbol{x}}-\tilde{\boldsymbol{x}}_v\| = 0$. Consequently, $\|\mu(\boldsymbol{X})-\mu(\tilde{\boldsymbol{X}}_\epsilon)\| = \infty$ can only be true if $m/n \geq 0.5$ for $T\in[0,\infty)$. $\square$

### D.2 Weighted Soft Median

It is easy to show that our proof also holds in the weighted case. Recall that we denote the weights of the neighbors with $\mathbf{a}$. We extract from the weighted/normalized adjacency matrix (compare with Eq. A.1). Given that a computer program represents numbers with limited precision, we do not provide an elaborate proof for weights in $\mathbb{R}$. Instead, we can convert a weighted problem into an unweighted, if we can find the greatest common divisor $\gcd(\mathbf{a}) = \gcd([\mathbf{a}_1 \quad \dots \quad \mathbf{a}_n])$. Once we found a gcd, we can use it to determine the factor of replications for each instance s.t. the relations to not change. For more details we refer to [18].

### D.3 Empirical Error

Similar to the finding in [18], we observe our Soft Median comes with a lower error if facing perturbed inputs (see Fig. D.1 which reproduce and complement Fig. 2 in [18]). Here we plot the error $\|t(\boldsymbol{X})-t(\tilde{\boldsymbol{X}})\|_2$, for 50 samples from a centered ($t_{\text{SoftMedian}}(\boldsymbol{X}) = 0$) bivariate normal distribution. The adversary is a point mass perturbation on the first axis over increasing fraction of outlier $\epsilon$.

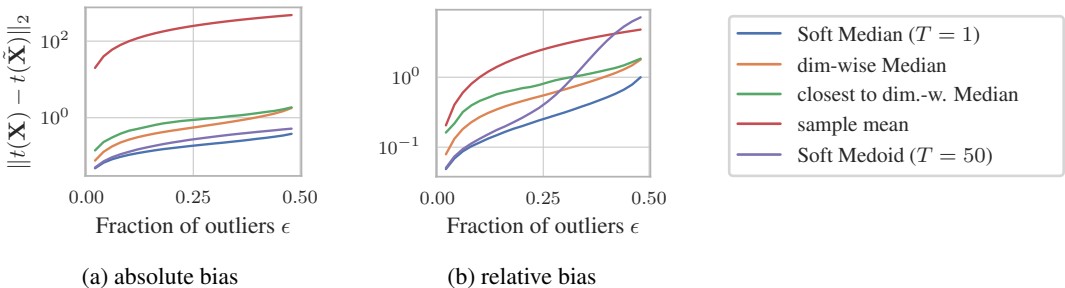

    (a) absolute bias           (b) relative bias

Figure D.1: Empirical error for a point mass perturbation. We reproduce Figure 2 in [18] and add our Soft Median.

### D.4 Improving Provable Robustness

Similarly to the Soft Medoid in [18, 32], in Table D.1, we show that the Soft Median can improve the certified robustness. Here we apply randomized smoothing [4] and obtain a significantly greater provable adversarial robustness. In the subsequent table, we show the "Accumulated certificates" obtained by randomized smoothing (same setup as in Table 2 in [18]). Even though our defense does not come with an adversarial robustness guarantee, we show that it can lead to increased *provable* robustness.

Table D.1: Certified robustness with randomized smoothing [4] following the setup of [18].

|          | Accumulated certificate | Add & Del. | Add. | Del. | Accuracy |
|----------|------------------------|-----------|------|------|----------|
| **Cora ML** | Soft Median GDC | **5.7** | **0.66** | **4.9** | **0.833** |
|          | Vanilla GCN | 1.84 | 0.21 | 4.41 | 0.823 |
|          | Soft Medoid GDC | 5.5 | 0.64 | 4.78 | 0.814 |
| **Citeseer** | Soft Median GDC | **4.43** | **0.57** | 4.31 | **0.728** |
|          | Vanilla GCN | 1.24 | 0.11 | 3.88 | 0.710 |
|          | Soft Medoid GDC | 3.64 | 0.49 | **4.33** | 0.705 |

# E  Theoretical Complexities

In the following, we summarize the theoretical complexities approaches we use in our experiments. We assume that the number of features and hidden neurons is negligible in comparison to e.g. the number of nodes. $k$ denotes the GDC/PPRGo hyperparameter for top-$k$-sparsification of the PPR matrix, $n$ is the number of nodes, and $m$ is the number of edges. We try to keep the overview simple and e.g. only list the most important hyperparameters. If a model preprocesses the adjacency matrix, we report the time complexities for preprocessing and GNN separately. For the attacks, we report the *additional* complexity (i.e. GNN excluded). We also use $b$ for the block size, $\Delta$ as the budget, and $E$ for the number of epochs. We chose $b = \mathcal{O}(m)$ for global attacks. In Table E.1 and Table E.2, we list the theoretical complexities of the used models and global attacks respectively.

Table E.1: Theoretical complexities of the models and defenses.

| Architecture | Memory Complexity | Time Complexity (all nodes) |
|--------------|-------------------|------------------------------|
| Soft Median GDC | $\mathcal{O}(k \cdot n)$ | $\mathcal{O}(n) + \mathcal{O}(k \cdot n)$ |
| Soft Median PPRGo | $\mathcal{O}(k)$ | $\mathcal{O}(n) + \mathcal{O}(k \cdot n)$ |
| Vanilla GCN | $\mathcal{O}(m)$ | $\mathcal{O}(m)$ |
| Vanilla GDC | $\mathcal{O}(k \cdot n)$ | $\mathcal{O}(n) + \mathcal{O}(k \cdot n)$ |
| Vanilla PPRGo | $\mathcal{O}(k)$ | $\mathcal{O}(n) + \mathcal{O}(k \cdot n)$ |
| Soft Medoid GDC | $\mathcal{O}(k^2 \cdot n)$ | $\mathcal{O}(n) + \mathcal{O}(k^2 \cdot n)$ |
| SVD GCN | $\mathcal{O}(n^2)$ | $\mathcal{O}(m) + \mathcal{O}(n^2)$ |
| Jaccard GCN | $\mathcal{O}(m)$ | $\mathcal{O}(n^2) + \mathcal{O}(m)$ |
| RGCN | $\mathcal{O}(m)$ | $\mathcal{O}(m)$ |

Table E.2: Theoretical complexities of the global attacks.

| Global Attack | Memory Complexity | Time Complexity | Details |
|---------------|-------------------|-----------------|---------|
| PR-BCD | $\mathcal{O}(b)$ | $\mathcal{O}(E \cdot b \log(b))$ | $b \geq \Delta$ |
| GR-BCD | $\mathcal{O}(b)$ | $\mathcal{O}(E \cdot b)$ | $b \geq \frac{\Delta}{E}$ |
| FGSM | $\mathcal{O}(n^2)$ | $\mathcal{O}(\Delta \cdot n^2)$ | |
| PGD | $\mathcal{O}(n^2)$ | $\mathcal{O}(E \cdot n^2)$ | |
| DICE | $\mathcal{O}(\Delta)$ | $\mathcal{O}(\Delta)$ | |

For the local complexities in Table E.3 we distinguish between the complexities *with and without* GNN.

We now continue with a discussion of SGA's complexity. The space complexity of SGA is largely driven by two factors: (a) edge deletions are considered for the entire receptive field of the attacked node, and (b) the edge insertions are considered to the top $\Delta$ nodes of the second most likely class.

To determine the top $\Delta$ for (b) one needs to obtain the gradient w.r.t. to the edge connecting to the nodes of the second most likely class. For some graphs this requires to check $\mathcal{O}(n)$ nodes. Even though this can be done iteratively or in batches.

Table E.3: Theoretical complexities of the local attacks.

| Local Attack Complexity of GNN | Memory Complexity | | Time Complexity | |
|---|---|---|---|---|
| | excluded | included | excluded | included |
| PR-BCD | $\mathcal{O}(b)$ | $\mathcal{O}(m)$ | $\mathcal{O}(E \cdot b \log(b))$ | $\mathcal{O}(E \cdot m)$ |
| PR-BCD (with PPRGo) | - | $\mathcal{O}(b \cdot k)$ | - | $\mathcal{O}(E \cdot b \log(b))$ |
| Nettack | $\mathcal{O}(\Delta)$ | $\mathcal{O}(m)$ | $\mathcal{O}(\Delta \cdot n)$ | $\mathcal{O}(E \cdot m)$ |
| SGA (with SGC) | - | $\mathcal{O}(m)$ | - | $\mathcal{O}(E \cdot m)$ |
| DICE | $\mathcal{O}(\Delta)$ | $\mathcal{O}(m)$ | $\mathcal{O}(\Delta)$ | $\mathcal{O}(E \cdot m)$ |
| DICE (with PPRGo) | - | $\mathcal{O}(\Delta)$ | - | $\mathcal{O}(E \cdot \Delta)$ |

It is even more challenging to obtain the gradient towards the edges for (a). This is simply due to the recursive nature of GNNs. For example, in a graph that approximately follows a power-law distribution, there will be some nodes with very high degree. Thus, for a moderate number of message passing steps and with high-degree nodes in the neighborhood, this subgraph might span a large fraction of nodes. Hence, in the worst case, the number of edges we need to consider scales with $\mathcal{O}(m)$. Note this is the same limitation we describe in § 3 when we say that "we are limited by the scalability of the attacked GNN". Hence, we also experience this if attacking a GCN locally on datasets larger than Products. This was our main motivation to also consider PPRGo.

# F  Empirical Evaluation

In § F.1 we start with a more thorough description of the experiment setup and in § F.2 we give insights into the time and memory cost. We conclude with additional experiments using our global attack and local attack in § F.3 and § F.4, respectively. While our local PR-BCD attack is adaptive since it does not rely on a surrogate model, our global attacks are not adaptive and transfer the attack from a Vanilla GCN (common practice presumably because many defenses/baselines are not differentiable, e.g. see [16, 18, 41, 48]). Nevertheless, we also experiment with direct, adaptive attacks on our defense Soft Median GCN in § F.3 or more specifically Table F.4.

## F.1  Setup

**Datasets.** We use the common Cora ML [2] and Citeseer [28] for a comprehensive comparison of the state-of-the-art attacks and defenses (most baselines do not scale further). For large scale experiments, we use PubMed [33] as well as arXiv, Products and Papers 100M of the recent Open Graph Benchmark [21]. Since most approaches rely on graphs of similar size as PubMed for their "large-scale" experiments, we scale the global attack by more than 100 times (number of nodes), or by factor 15,000 if counting the possible adjacency matrix entries (see Table F.1). We scale our local attack to Papers 100M which has 111 million nodes, outscaling previous local attacks by a factor of 500. For a detailed overview see Table F.1.

Table F.1: Statistics of the used datasets (extension of Table 1). For the dense adjacency matrix we assume that each elements is represented by 4 bytes. In the sparse case we use two 8 byte integer pointers and a 4 bytes float value. For Cora ML, Citeseer and PubMed we extract the largest connected component.

| Dataset | License | #Features $d$ | #Nodes $n$ | #Edges $e$ | #Possible edges $n^2$ | Average degree $e/n$ | Size (dense) | Size (sparse) |
|---|---|---|---|---|---|---|---|---|
| Cora ML [2] | N/A | 2,879 | 2,810 | 15,962 | 7.896E+06 | 5.68 | 31.58 MB | 319.24 kB |
| Citeseer [28] | N/A | 3,703 | 2,110 | 7,336 | 4.452E+06 | 3.48 | 17.81 MB | 146.72 kB |
| PubMed [33] | N/A | 500 | 19,717 | 88,648 | 3.888E+08 | 4.50 | 1.56 GB | 1.77 MB |
| arXiv [21] | ODC-BY | 128 | 169,343 | 1,166,243 | 2.868E+10 | 6.89 | 114.71 GB | 23.32 MB |
| Products [21] | Amazon | 100 | 2,449,029 | 123,718,280 | 5.998E+12 | 50.52 | 23.99 TB | 2.47 GB |
| Papers 100M [21] | ODC-BY | 128 | 111,059,956 | 1,615,685,872 | 1.233E+16 | 14.55 | 49.34 PB | 32.31 GB |

**Attacks.** We compare our global attacks PR-BCD and GR-BCD (§ 3) with PGD [43], and greedy FGSM (similar to Dai et al. [14]) attacks. The greedy FGSM-like attack is the dense equivalent of our GR-BCD attack with the exception of flipping one edge at a time. Regardless of the scale of the

datasets, we also compare to the global DICE [37] attack. DICE is a greedy, randomized black-box attack that flips one randomly determined entry in the adjacency matrix at a time. An edge is deleted if both nodes share the same label and an edge is added if the labels of the nodes differ. We ensure that a single node does not become disconnected. Moreover, we use 60% of the budget to add new edges and otherwise remove edges. We compare our local PR-BCD (§ 3) with Nettack [50]. Nettack perturbs the adjacency matrix greedily exploiting the properties of the linearized GCN surrogate.

**Defenses.** We compare our Soft Median architectures with state-of-the-art defenses [16, 18, 41, 48]. Following [18], we use the GDC/PPR preprocessing [25] in combination with our Soft Median. The SVD GCN [16] uses a (dense) low-rank approximation (here rank 50) of the adjacency matrix to filter adversarial perturbations. RGCN [48] models the neighborhood aggregation via Gaussian distribution to filter outliers, and Jaccard GCN [41] filters edges based on attribute dissimilarity (here threshold 0.01). For the Soft Medoid GDC, we use the temperature $T = 0.5$ as it is a good compromise between accuracy and robustness (except for arXiv where we choose $T = 5.0$). For more details about the Soft Medoid GDC see § 4.

**Checkpointing.** Empirically, almost 30 GB are required to train a three-layer GCN on Products (our largest dataset for global attacks) using sparse matrices. However, obtaining the gradient, e.g. towards the perturbation vector/matrix, requires extra memory. We notice that most operations in modern GNNs only depend on the neighborhood size (i.e. a row in the adjacency matrix). As proposed by Chen et al. [12], the gradient is obtainable with sublinear memory cost via checkpointing. The idea is to discard some intermediate results in the forward phase (that would be cached) and recompute them in the backward phase. Specifically, we chunk some operations (e.g. matrix multiplication) within the message passing step to successfully scale to larger graphs. This allows us to attack a three-layer GCN on Products with full GPU acceleration.

**Hyperparameters.** For full details we refer to our code and configuration. We run every experiment for three random seeds (unless otherwise stated) except for the largest dataset Papers100M.

*Models:* We typically train for at most 3000 epochs with early stopping and patience of 300. We use a learning rate of 0.01 and a weight decay of 0.001 for all models (except PPRGo and SGC). On all datasets, we use a restart probability of 0.15 (except arXiv 0.1) for GDC and sparsify the adjacency by selecting the top 64 edges in each row. For the remaining configuration, we closely follow the setup of Geisler et al. [18] on Cora, Citeseer, and Pubmed. On arXiv as well as Products we follow Hu et al. [21] but still train for 3000 epochs with early-stopping patience of 300 and use three layers / message passing steps. We only deviate from the standard configuration for the PPRGo models as they are more sensitive to the hyperparameter choice. We use checkpointing for the Soft Medoid and Soft Median GDC on arXiv and for all models on Products. For the Soft Medoid and Soft Median GDC, we lower the number of layers and hidden dimensions such that they fit in the GPU memory. We determine the optimal temperature/parameters of our Soft Median GDC/PPRGo through a rudimentary grid search. On the small datasets, we typically end up with either $T = 0.2$ or $T = 0.5$. The larger the dataset, the larger the best temperature becomes (this effect is even stronger in combination with PPRGo). For the largest dataset Papers100M, we plot and analyze the influence of the temperature in Fig. F.3.

*Attacks:* For the global attacks GR-BCD and PR-BCD, we run the attack for 500 epochs (100 epochs fine-tuning with PR-BCD). We choose a block size $b$ of 1,000,000, 2,500,000, 10,000,000 for Cora ML/Citeseer, Pubmed and arXiv/Products, respectively. For our local PR-BCD, we also attack for 500 epochs on Cora ML and Citeseer but observe that for Products and Papers100M 30 epochs are sufficient. Respectively, we choose a block size $b$ of 10,000, 10,000, 20,000 and 2,500. We select the learning rates such that the budget requirement is met.

## F.2   Time and Memory Cost

We present the time and memory cost for our global PR-BCD in Table F.2 and for our local PR-BCD in Table F.3[2]. In all cases, our attack is reasonably fast and comes with a low memory footprint. We refer to Fig. 1 for a comparison with the dense PGD attack. Note that a global PGD attack on arXiv would require around 1 TB (see Fig. 1) while our PR-BCD only requires around 4 GB while still

---

[2]Note that the numbers presented here are slightly more pessimistic than in the main part since here we report the runtime/memory consumption including preprocessing and postprocessing steps. We will make this consistent in a future version of this paper.

Table F.2: Time cost and memory cost of our *global* PR-BCD attack on a Vanilla GCN.

| Dataset | Block size $b$ | Max GPU memory \ GB | Duration of epoch \ s |
|---------|----------------|---------------------|------------------------|
| Cora ML | 1,000,000 | 0.34 | 0.12 |
| PubMed | 2,500,000 | 0.89 | 0.32 |
| arXiv | 10,000,000 | 4.29 | 1.55 |

Table F.3: Time cost and memory cost of our *local* PR-BCD attack on PPRGo.

| Dataset | Architecture (victim) | Block size $b$ | Max GPU memory \ GB | Duration of epoch \ s |
|---------|----------------------|----------------|---------------------|------------------------|
| Cora ML | Soft Median PPRGo | 10,000 | 0.08 | 0.49 |
| | Vanilla PPRGo | 10,000 | 0.12 | 0.52 |
| Citeseer | Soft Median PPRGo | 10,000 | 0.08 | 0.39 |
| | Vanilla PPRGo | 10,000 | 0.14 | 0.47 |
| Products | Soft Median PPRGo | 20,000 | 5.58 | 4.29 |
| | Vanilla PPRGo | 20,000 | 6.07 | 3.26 |
| Papers 100M | Soft Median PPRGo | 2,500 | 2.85 | 14.66 |
| | Vanilla PPRGo | 2,500 | 2.81 | 14.82 |

being effective (see Table F.2). In Table F.3 we see that with a proper choice of the block size $b$ once can attack a massive dataset on a reasonably sized GPU. Moreover, our Soft Medoid PPRGo and the Vanilla PPRGo come with a comparable runtime and memory consumption.

## F.3  Global Attacks

We compare the robustness of the models over different attack budgets in Fig. F.1 (similarly to Fig. 7). In Table F.5, we give a detailed overview of how each model's accuracy declines for all benchmarked attacks. More importantly, in Table F.4, we provide the results for an adaptive attack.

Table F.4: Adversarial accuracy using *direct, adaptive* attacks and our Soft Median GDC with the vanilla baselines. We show the adversarial accuracy and the clean test accuracy (last column). We highlight the **strongest defense** in bold as the attacks perform similarly. Our approaches are underlined.

| Dataset | Attack | GR-BCD | | | | | |
|---------|--------|--------|--------|--------|--------|--------|--------|
| | Frac. edges $\epsilon$ | 0.01 | 0.05 | 0.10 | 0.25 | 0.50 | 1.00 |
| Cora ML | Soft Median GDC | $0.807 \pm 0.002$ | $\mathbf{0.773 \pm 0.002}$ | $\mathbf{0.749 \pm 0.001}$ | $\mathbf{0.692 \pm 0.004}$ | $\mathbf{0.648 \pm 0.002}$ | $\mathbf{0.603 \pm 0.001}$ |
| | Vanilla GCN | $0.789 \pm 0.003$ | $0.699 \pm 0.003$ | $0.619 \pm 0.004$ | $0.475 \pm 0.004$ | $0.333 \pm 0.003$ | $0.148 \pm 0.005$ |
| | Vanilla GDC | $\mathbf{0.808 \pm 0.002}$ | $0.749 \pm 0.003$ | $0.703 \pm 0.003$ | $0.623 \pm 0.005$ | $0.513 \pm 0.005$ | $0.396 \pm 0.007$ |
| Citeseer | Soft Median GDC | $\mathbf{0.705 \pm 0.001}$ | $\mathbf{0.687 \pm 0.002}$ | $\mathbf{0.664 \pm 0.001}$ | $\mathbf{0.626 \pm 0.003}$ | $\mathbf{0.582 \pm 0.001}$ | $\mathbf{0.536 \pm 0.004}$ |
| | Vanilla GCN | $0.689 \pm 0.002$ | $0.618 \pm 0.001$ | $0.554 \pm 0.001$ | $0.410 \pm 0.003$ | $0.265 \pm 0.002$ | $0.105 \pm 0.008$ |
| | Vanilla GDC | $0.679 \pm 0.001$ | $0.626 \pm 0.002$ | $0.588 \pm 0.006$ | $0.504 \pm 0.003$ | $0.421 \pm 0.003$ | $0.309 \pm 0.007$ |

| | Attack | PR-BCD | | | | | |
|---------|--------|--------|--------|--------|--------|--------|--------|
| | Frac. edges $\epsilon$ | 0.01 | 0.05 | 0.10 | 0.25 | 0.50 | 1.00 |
| Cora ML | Soft Median GDC | $0.796 \pm 0.002$ | $\mathbf{0.735 \pm 0.002}$ | $\mathbf{0.690 \pm 0.002}$ | $\mathbf{0.615 \pm 0.003}$ | $\mathbf{0.564 \pm 0.005}$ | $\mathbf{0.523 \pm 0.005}$ |
| | Vanilla GCN | $0.792 \pm 0.003$ | $0.704 \pm 0.004$ | $0.635 \pm 0.005$ | $0.478 \pm 0.003$ | $0.309 \pm 0.005$ | $0.141 \pm 0.005$ |
| | Vanilla GDC | $\mathbf{0.799 \pm 0.002}$ | $0.711 \pm 0.003$ | $0.645 \pm 0.005$ | $0.532 \pm 0.006$ | $0.457 \pm 0.005$ | $0.400 \pm 0.006$ |
| Citeseer | Soft Median GDC | $\mathbf{0.692 \pm 0.002}$ | $\mathbf{0.650 \pm 0.002}$ | $\mathbf{0.615 \pm 0.003}$ | $\mathbf{0.548 \pm 0.005}$ | $\mathbf{0.494 \pm 0.006}$ | $\mathbf{0.446 \pm 0.008}$ |
| | Vanilla GCN | $0.689 \pm 0.002$ | $0.621 \pm 0.003$ | $0.560 \pm 0.004$ | $0.429 \pm 0.007$ | $0.282 \pm 0.012$ | $0.127 \pm 0.009$ |
| | Vanilla GDC | $0.670 \pm 0.001$ | $0.591 \pm 0.004$ | $0.515 \pm 0.002$ | $0.374 \pm 0.003$ | $0.264 \pm 0.007$ | $0.194 \pm 0.006$ |

**Adaptive, global attack.** In Table F.5, we compare our Soft Median GDC to a Vanilla GCN (and Vanilla GDC as ablation). We make two major observations: (1) our Soft Median GDC outperforms the baselines by a large margin over all budgets $\epsilon > 0.01$; (2) while in the previous experiments the greedy attacks (FGSM or GR-BCD) seemed to perform on par (sometimes even stronger) with PGD as well as PR-BCD, this does not hold if we attack Vanilla GDC or our Soft Median GDC. This implies that for a Vanilla GCN the gradient on the clean edges is an excellent indicator for the

effectiveness of an edge flip. Moreover and with no surprise, transfer attacks can provide a false impression of robustness. We want to emphasize that for GNNs most defenses were not studied using an adaptive attack (e.g. see [16, 18, 41, 48]). We now compare the performance with an MLP that achieves around 60% on Cora ML [34]. With GR-BCD we hit 60% adversarial accuracy for $\epsilon = 1$ but for the PR-BCD we already drop below it somewhere in $[0.25, 0.5]$. Note that the Vanilla GCN already drops below the MLP performance $\epsilon \approx 0.1$. For large budgets (e.g. $\epsilon = 1.0$), our Soft Median GDC has a four to five times higher adversarial accuracy than a Vanilla GCN.

Unfortunately, we are not aware of an efficient PPR implementation (large fraction of nodes at once) that allows us to backpropagate through it. Moreover, our Soft Median would lose a lot of its robustness, if we removed the PPR diffusion (GDC). Hence, we are limited to a calculation of the PPR scores with the matrix inverse. This is not scalable (runtime complexity $\mathbf{O}(n^3)$) and the inverse of a sparse matrix is not sparse in general (space complexity $\mathbf{O}(n^2)$). For this reason, we can only use an adaptive, global attack on Soft Median GDC on sufficiently small datasets. For adaptive attacks at scale, we refer to the local attacks in § 5 and § F.4.

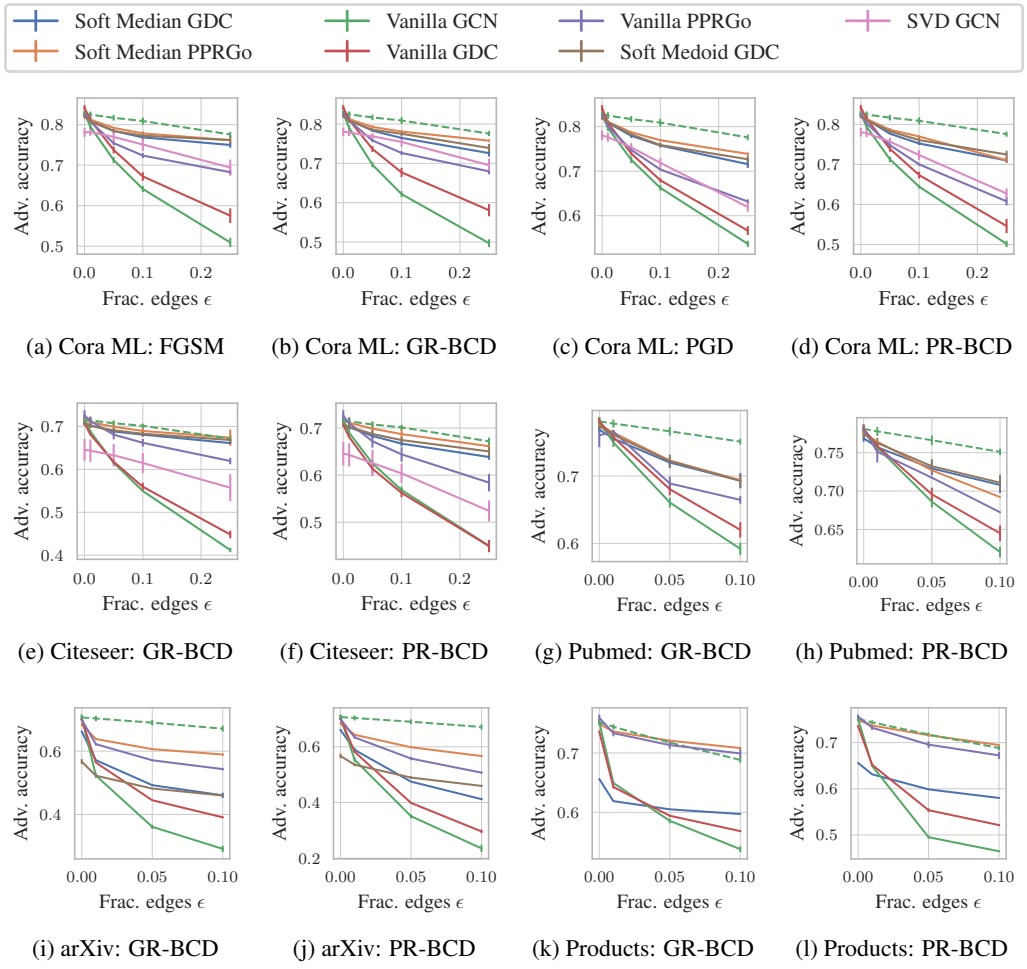

Figure F.1: Adversarial accuracy for the budget $\epsilon$ as a fraction of edges. The lower the adversarial accuracy, the stronger the attack or weaker the defense. The higher the adversarial accuracy, the stronger the defense or weaker the attack. Here we provide more detail than in Fig. 7.

Table F.5: Comparing attacks (transfer from Vanilla GCN) and defenses. We show the adversarial accuracy and the clean test accuracy (last column). We highlight the **strongest defense** in bold as the attacks perform similarly. A more *nuanced highlighting* is given for the strongest attack for each architecture and budget. Our approaches are underlined.

| Dataset | Attack / Frac. edges ε | DICE 0.05 | DICE 0.1 | FGSM 0.05 | FGSM 0.1 | GR-BCD 0.05 | GR-BCD 0.1 | PGD 0.05 | PGD 0.1 | PR-BCD 0.05 | PR-BCD 0.1 | Acc. |
|---|---|---|---|---|---|---|---|---|---|---|---|---|
| Cora ML | _Soft Median GDC_ | 0.816 ± 0.002 | 0.813 ± 0.002 | 0.784 ± 0.001 | 0.769 ± 0.002 | 0.783 ± 0.001 | 0.765 ± 0.001 | 0.779 ± 0.002 | 0.758 ± 0.002 | 0.777 ± 0.000 | 0.752 ± 0.002 | 0.824 ± 0.002 |
| Cora ML | _Soft Median PPRGo_ | 0.814 ± 0.002 | 0.804 ± 0.001 | **0.792 ± 0.000** | **0.778 ± 0.001** | **0.793 ± 0.002** | **0.781 ± 0.002** | **0.787 ± 0.001** | **0.769 ± 0.001** | **0.787 ± 0.002** | **0.770 ± 0.001** | 0.821 ± 0.001 |
| Cora ML | Vanilla GCN | 0.817 ± 0.003 | 0.809 ± 0.004 | 0.713 ± 0.003 | 0.641 ± 0.003 | 0.697 ± 0.003 | 0.622 ± 0.003 | 0.726 ± 0.004 | 0.662 ± 0.004 | 0.713 ± 0.003 | 0.645 ± 0.002 | 0.827 ± 0.003 |
| Cora ML | Vanilla GDC | **0.830 ± 0.003** | **0.819 ± 0.002** | 0.738 ± 0.004 | 0.672 ± 0.005 | 0.737 ± 0.003 | 0.677 ± 0.005 | 0.740 ± 0.002 | 0.679 ± 0.003 | 0.739 ± 0.003 | 0.674 ± 0.004 | **0.842 ± 0.003** |
| Cora ML | Vanilla PPRGo | 0.816 ± 0.002 | 0.807 ± 0.001 | 0.754 ± 0.002 | 0.724 ± 0.003 | 0.758 ± 0.002 | 0.726 ± 0.002 | 0.748 ± 0.002 | 0.704 ± 0.001 | 0.748 ± 0.002 | 0.700 ± 0.002 | 0.826 ± 0.002 |
| Cora ML | Vanilla GAT | 0.763 ± 0.002 | 0.725 ± 0.003 | 0.741 ± 0.001 | 0.688 ± 0.002 | 0.743 ± 0.000 | 0.699 ± 0.001 | 0.731 ± 0.002 | 0.683 ± 0.003 | 0.738 ± 0.001 | 0.677 ± 0.002 | 0.806 ± 0.001 |
| Cora ML | Soft Medoid GDC | 0.814 ± 0.002 | 0.809 ± 0.002 | 0.784 ± 0.003 | 0.773 ± 0.005 | 0.786 ± 0.002 | 0.775 ± 0.003 | 0.782 ± 0.003 | 0.759 ± 0.003 | 0.783 ± 0.001 | 0.761 ± 0.003 | 0.819 ± 0.002 |
| Cora ML | SVD GCN | 0.766 ± 0.005 | 0.752 ± 0.003 | 0.770 ± 0.006 | 0.751 ± 0.007 | 0.769 ± 0.004 | 0.755 ± 0.006 | 0.753 ± 0.004 | 0.719 ± 0.005 | 0.757 ± 0.004 | 0.724 ± 0.006 | 0.781 ± 0.005 |
| Cora ML | Jaccard GCN | 0.809 ± 0.003 | 0.803 ± 0.003 | 0.722 ± 0.002 | 0.661 ± 0.002 | 0.719 ± 0.001 | 0.664 ± 0.001 | 0.730 ± 0.003 | 0.673 ± 0.002 | 0.725 ± 0.001 | 0.667 ± 0.003 | 0.818 ± 0.003 |
| Cora ML | RGCN | 0.808 ± 0.002 | 0.796 ± 0.003 | 0.719 ± 0.004 | 0.654 ± 0.004 | 0.725 ± 0.002 | 0.665 ± 0.005 | 0.725 ± 0.005 | 0.671 ± 0.007 | 0.724 ± 0.003 | 0.664 ± 0.004 | 0.819 ± 0.002 |
| Citeseer | _Soft Median GDC_ | 0.706 ± 0.001 | 0.699 ± 0.001 | 0.695 ± 0.002 | 0.676 ± 0.002 | 0.688 ± 0.002 | 0.681 ± 0.002 | 0.686 ± 0.002 | 0.675 ± 0.002 | 0.683 ± 0.002 | 0.667 ± 0.003 | 0.708 ± 0.002 |
| Citeseer | _Soft Median PPRGo_ | 0.709 ± 0.006 | 0.700 ± 0.006 | **0.697 ± 0.006** | **0.685 ± 0.007** | **0.699 ± 0.006** | **0.690 ± 0.006** | **0.700 ± 0.005** | **0.692 ± 0.007** | **0.699 ± 0.007** | **0.687 ± 0.006** | 0.716 ± 0.006 |
| Citeseer | Vanilla GCN | 0.708 ± 0.003 | 0.702 ± 0.002 | 0.633 ± 0.003 | 0.574 ± 0.004 | 0.616 ± 0.001 | 0.550 ± 0.001 | 0.643 ± 0.002 | 0.594 ± 0.002 | 0.625 ± 0.004 | 0.568 ± 0.004 | 0.716 ± 0.003 |
| Citeseer | Vanilla GDC | 0.694 ± 0.001 | 0.687 ± 0.002 | 0.622 ± 0.002 | 0.562 ± 0.003 | 0.618 ± 0.003 | 0.560 ± 0.004 | 0.627 ± 0.001 | 0.581 ± 0.003 | 0.614 ± 0.005 | 0.562 ± 0.004 | 0.707 ± 0.001 |
| Citeseer | Vanilla PPRGo | **0.719 ± 0.005** | **0.708 ± 0.006** | 0.677 ± 0.008 | 0.644 ± 0.009 | 0.681 ± 0.005 | 0.662 ± 0.004 | 0.675 ± 0.004 | 0.649 ± 0.004 | 0.672 ± 0.006 | 0.644 ± 0.007 | **0.726 ± 0.006** |
| Citeseer | Vanilla GAT | 0.582 ± 0.006 | 0.544 ± 0.004 | 0.571 ± 0.011 | 0.520 ± 0.011 | 0.577 ± 0.008 | 0.527 ± 0.004 | 0.574 ± 0.004 | 0.522 ± 0.005 | 0.584 ± 0.012 | 0.524 ± 0.006 | 0.647 ± 0.012 |
| Citeseer | Soft Medoid GDC | 0.704 ± 0.004 | 0.701 ± 0.002 | 0.694 ± 0.004 | 0.682 ± 0.003 | 0.691 ± 0.003 | 0.683 ± 0.002 | 0.688 ± 0.002 | 0.677 ± 0.003 | 0.688 ± 0.004 | 0.675 ± 0.003 | 0.708 ± 0.003 |
| Citeseer | SVD GCN | 0.635 ± 0.011 | 0.623 ± 0.012 | 0.632 ± 0.012 | 0.617 ± 0.012 | 0.633 ± 0.012 | 0.615 ± 0.011 | 0.630 ± 0.010 | 0.599 ± 0.013 | 0.626 ± 0.013 | 0.604 ± 0.009 | 0.646 ± 0.012 |
| Citeseer | Jaccard GCN | 0.716 ± 0.005 | 0.708 ± 0.004 | 0.663 ± 0.004 | 0.622 ± 0.006 | 0.654 ± 0.004 | 0.616 ± 0.003 | 0.666 ± 0.004 | 0.630 ± 0.003 | 0.650 ± 0.005 | 0.609 ± 0.005 | 0.721 ± 0.005 |
| Citeseer | RGCN | 0.676 ± 0.006 | 0.663 ± 0.006 | 0.622 ± 0.003 | 0.568 ± 0.005 | 0.624 ± 0.005 | 0.584 ± 0.004 | 0.629 ± 0.003 | 0.589 ± 0.004 | 0.628 ± 0.004 | 0.583 ± 0.006 | 0.686 ± 0.005 |
| PubMed | _Soft Median GDC_ | 0.761 ± 0.002 | 0.752 ± 0.003 | - | - | 0.727 ± 0.004 | 0.693 ± 0.005 | - | - | 0.730 ± 0.005 | 0.708 ± 0.005 | 0.769 ± 0.002 |
| PubMed | _Soft Median PPRGo_ | 0.764 ± 0.001 | 0.752 ± 0.002 | - | - | **0.723 ± 0.000** | **0.694 ± 0.001** | - | - | 0.727 ± 0.000 | 0.692 ± 0.000 | 0.776 ± 0.002 |
| PubMed | Vanilla GCN | 0.766 ± 0.003 | 0.751 ± 0.002 | - | - | 0.661 ± 0.003 | 0.592 ± 0.004 | - | - | 0.686 ± 0.004 | 0.620 ± 0.003 | **0.781 ± 0.003** |
| PubMed | Vanilla GDC | 0.766 ± 0.003 | 0.748 ± 0.002 | - | - | 0.680 ± 0.004 | 0.620 ± 0.005 | - | - | 0.696 ± 0.004 | 0.645 ± 0.005 | 0.781 ± 0.002 |
| PubMed | Vanilla PPRGo | 0.717 ± 0.001 | 0.721 ± 0.007 | - | - | 0.714 ± 0.001 | 0.673 ± 0.002 | - | - | 0.704 ± 0.007 | 0.658 ± 0.004 | 0.765 ± 0.008 |
| PubMed | Soft Medoid GDC | **0.766 ± 0.003** | **0.756 ± 0.003** | - | - | 0.722 ± 0.004 | 0.693 ± 0.005 | - | - | **0.732 ± 0.004** | **0.711 ± 0.005** | 0.774 ± 0.003 |
| arXiv | _Soft Median GDC_ | 0.645 ± 0.002 | 0.629 ± 0.002 | - | - | 0.504 ± 0.003 | 0.462 ± 0.001 | - | - | 0.479 ± 0.002 | 0.420 ± 0.005 | 0.666 ± 0.002 |
| arXiv | _Soft Median PPRGo_ | 0.669 ± 0.001 | 0.654 ± 0.001 | - | - | **0.606 ± 0.001** | **0.589 ± 0.002** | - | - | **0.598 ± 0.001** | **0.567 ± 0.002** | 0.684 ± 0.001 |
| arXiv | Vanilla GCN | **0.690 ± 0.004** | **0.671 ± 0.004** | - | - | 0.361 ± 0.003 | 0.292 ± 0.005 | - | - | 0.351 ± 0.003 | 0.235 ± 0.006 | **0.706 ± 0.004** |
| arXiv | Vanilla GDC | 0.672 ± 0.001 | 0.648 ± 0.001 | - | - | 0.446 ± 0.001 | 0.390 ± 0.001 | - | - | 0.399 ± 0.001 | 0.297 ± 0.003 | 0.701 ± 0.001 |
| arXiv | Vanilla PPRGo | 0.680 ± 0.002 | 0.662 ± 0.002 | - | - | 0.571 ± 0.001 | 0.543 ± 0.002 | - | - | 0.558 ± 0.003 | 0.507 ± 0.002 | 0.699 ± 0.002 |
| arXiv | Soft Medoid GDC | 0.554 ± 0.004 | 0.543 ± 0.003 | - | - | 0.482 ± 0.001 | 0.460 ± 0.001 | - | - | 0.490 ± 0.002 | 0.460 ± 0.002 | 0.567 ± 0.004 |
| Products | _Soft Median GDC_ | 0.637 ± 0.000 | 0.624 ± 0.000 | - | - | 0.605 ± 0.000 | 0.597 ± 0.000 | - | - | 0.599 ± 0.001 | 0.580 ± 0.001 | 0.656 ± 0.000 |
| Products | _Soft Median PPRGo_ | 0.725 ± 0.001 | **0.712 ± 0.001** | - | - | **0.721 ± 0.001** | **0.708 ± 0.001** | - | - | **0.716 ± 0.001** | **0.695 ± 0.001** | 0.749 ± 0.001 |
| Products | Vanilla GCN | 0.717 ± 0.002 | 0.688 ± 0.002 | - | - | 0.586 ± 0.002 | 0.538 ± 0.002 | - | - | 0.495 ± 0.002 | 0.465 ± 0.001 | 0.751 ± 0.002 |
| Products | Vanilla GDC | 0.693 ± 0.000 | 0.661 ± 0.000 | - | - | 0.594 ± 0.001 | 0.568 ± 0.001 | - | - | 0.554 ± 0.002 | 0.521 ± 0.001 | 0.736 ± 0.001 |
| Products | Vanilla PPRGo | **0.727 ± 0.002** | 0.711 ± 0.002 | - | - | 0.713 ± 0.003 | 0.699 ± 0.004 | - | - | 0.696 ± 0.003 | 0.672 ± 0.004 | **0.757 ± 0.002** |

## F.4 Local Attacks

We present additional results for the local attacks. In Fig. F.2 we complement Fig. 8 with the additional dataset Citeseer and more budgets on Products as well as Papers100M. For the experiments on the undirected Cora ML and Citeseer, we treat the graph as if it was directed during the attack and symmetrize it afterward. Despite this approximation, our attack remains strong.

Table F.6: Attack success rates for the local attacks. Our approaches are underlined. For the attack a higher value is better and for the defence a lower value is better. We highlight the **strongest defense** in bold.

| Dataset | Attack | PR-BCD | | | |
|---|---|---|---|---|---|
| | Frac. edges $\epsilon$, $\Delta_i = \epsilon d_i$ | 0.10 | 0.25 | 0.50 | 1.00 |
| Cora ML | Soft Median PPRGo | **0.125 ± 0.003** | **0.208 ± 0.003** | **0.333 ± 0.004** | **0.417 ± 0.004** |
| | Vanilla PPRGo | 0.133 ± 0.003 | 0.317 ± 0.004 | 0.425 ± 0.004 | 0.583 ± 0.004 |
| | Vanilla GCN | 0.292 ± 0.004 | 0.375 ± 0.004 | 0.642 ± 0.004 | 0.958 ± 0.002 |
| Citeseer | Soft Median PPRGo | **0.058 ± 0.002** | **0.242 ± 0.004** | **0.358 ± 0.004** | **0.517 ± 0.004** |
| | Vanilla PPRGo | **0.058 ± 0.002** | 0.333 ± 0.004 | 0.492 ± 0.004 | 0.600 ± 0.004 |
| | Vanilla GCN | 0.125 ± 0.003 | 0.417 ± 0.004 | 0.658 ± 0.004 | 0.850 ± 0.003 |
| Products | Soft Median PPRGo | **0.108 ± 0.003** | **0.083 ± 0.002** | **0.142 ± 0.003** | **0.250 ± 0.004** |
| | Vanilla PPRGo | 0.683 ± 0.004 | 0.858 ± 0.003 | 0.925 ± 0.002 | 0.950 ± 0.002 |
| | Vanilla GCN | 0.883 ± 0.003 | 0.950 ± 0.002 | 0.992 ± 0.001 | 0.992 ± 0.001 |
| Papers 100M | Soft Median PPRGo | **0.250 ± 0.011** | **0.325 ± 0.012** | **0.275 ± 0.011** | **0.300 ± 0.011** |
| | Vanilla PPRGo | 0.900 ± 0.007 | 0.925 ± 0.007 | 0.875 ± 0.008 | 0.975 ± 0.004 |

| Dataset | Attack | Nettack | | | |
|---|---|---|---|---|---|
| | Frac. edges $\epsilon$, $\Delta_i = \epsilon d_i$ | 0.10 | 0.25 | 0.50 | 1.00 |
| Cora ML | Soft Median PPRGo | **0.058 ± 0.002** | **0.125 ± 0.003** | **0.217 ± 0.003** | **0.317 ± 0.004** |
| | Vanilla PPRGo | 0.092 ± 0.002 | 0.258 ± 0.004 | 0.358 ± 0.004 | 0.475 ± 0.004 |
| | Vanilla GCN | 0.208 ± 0.003 | 0.367 ± 0.004 | 0.533 ± 0.004 | 0.917 ± 0.002 |
| Citeseer | Soft Median PPRGo | **0.042 ± 0.002** | **0.150 ± 0.003** | **0.308 ± 0.004** | **0.458 ± 0.004** |
| | Vanilla PPRGo | 0.050 ± 0.002 | 0.300 ± 0.004 | 0.433 ± 0.004 | 0.608 ± 0.004 |
| | Vanilla GCN | 0.133 ± 0.003 | 0.425 ± 0.004 | 0.667 ± 0.004 | 0.858 ± 0.003 |

| Dataset | Attack | DICE | | | |
|---|---|---|---|---|---|
| | Frac. edges $\epsilon$, $\Delta_i = \epsilon d_i$ | 0.10 | 0.25 | 0.50 | 1.00 |
| Products | Soft Median PPRGo | **0.017 ± 0.001** | **0.017 ± 0.001** | **0.008 ± 0.001** | **0.100 ± 0.003** |
| | Vanilla PPRGo | 0.108 ± 0.003 | 0.167 ± 0.003 | 0.250 ± 0.004 | 0.308 ± 0.004 |
| | Vanilla GCN | 0.183 ± 0.003 | 0.233 ± 0.004 | 0.267 ± 0.004 | 0.467 ± 0.004 |
| Papers 100M | Soft Median PPRGo | 0.150 ± 0.009 | 0.125 ± 0.008 | **0.125 ± 0.008** | **0.100 ± 0.008** |
| | Vanilla PPRGo | **0.075 ± 0.007** | **0.100 ± 0.008** | 0.200 ± 0.010 | 0.350 ± 0.012 |

In Table F.6, we give an alternative metric to assess our attack PR-BCD and defense Soft Median PPRGo. Here we compare the *attack success rate*. A "success" stands for a change of prediction in the course of the attack. On the upside, the attack success rate is less susceptible to the distribution of the clean margin (especially if comparing different models). On the downside, the attack success rate is not as fine-grained as comparing the margins. We see that our PR-BCD is stronger in most cases (i.e. the attack success rate is higher). Moreover, our Soft Median PPRGo beats all baselines for every budget except two cases for small budgets with the random, weak DICE attack.

**The temperature hyperparameter $T$.** Last, we study how the temperature affects the robustness of the Soft Median PPRGo model (see Fig. F.3). Similarly to [18], we observe that for large values it performs comparably to the vanilla model. If we lower the temperature the robustness increases at the cost of a slightly lower clean accuracy. If we set the temperature too low, the accuracy still declines but also does the robustness. This shows the trade-off between clean accuracy and robustness. Hence, one needs to carefully choose a good temperature for the application at hand. A good strategy is to (1) tune a vanilla model to meet the desired predictive performance and (2) successively decay the temperature (starting high) until the robustness is decreasing or until the drop in predictive performance exceeds the application-specific threshold.

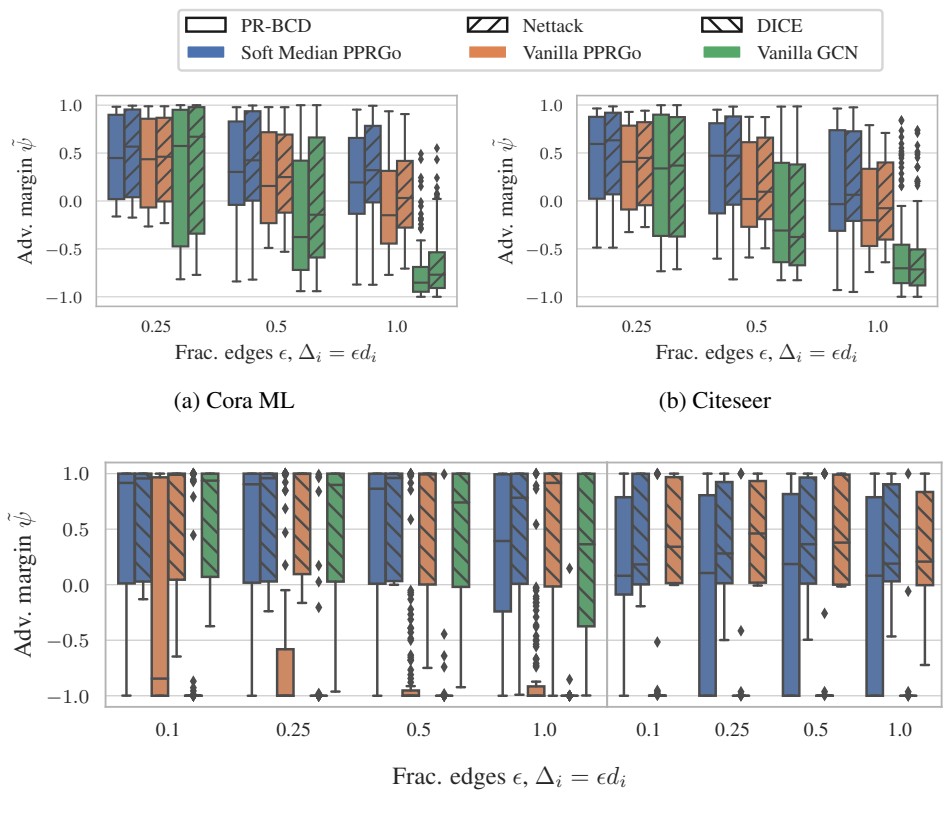

(a) Cora ML

(b) Citeseer

(c) Products (left) & Papers100M (right)

Figure F.2: Adversarial classification margins $\tilde{\psi}_i$ of the attacked nodes. This figure extends Fig. 8 with an additional dataset and more budgets. In (a) and (b), we compare our local PR-BCD attack with Nettack [50] on (undirected) Cora ML and Citeseer. In (c), we show the results on the (directed) large-scale datasets Products (2.5 million nodes) and Papers 100M (111 million nodes), respectively. Our Soft Medoid PPRGo resists the attacks much better than the baselines.

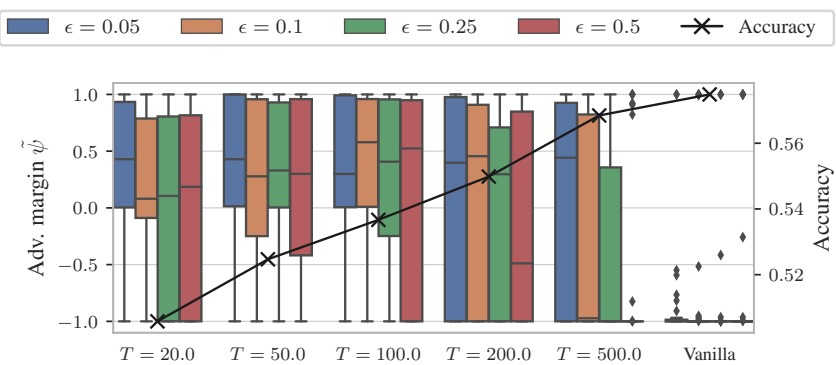

Figure F.3: Adversarial classification margins $\tilde{\psi}_i$ (left y axis) and clean accuracy (right y axis) over the temperature and for different budgets on Papers100M (111 million nodes). We see that the clean accuracy increases if the temperature increases. The robustness decreases for very low and large temperatures. There is a sweet spot for maximum robustness in between.

## F.5 Relationship of Graph Size and GNNs Robustness

We now analyze the results of PR-BCD w.r.t. the graph size. We start comparing the global attacks with a budget of $\epsilon = 0.1$ for the Vanilla GCN model. We observe a relative drop in the adversarial

accuracy by 22% on Cora, 67 % on arXiv, and 38% on Products. On the larger graphs, the degradation of the accuracy is much larger which indicates a relationship between the adversarial robustness and the size of the graph. This relationship seems to persist for architectures other than GCN as well. Of course, this comparison neglects the characteristics of the dataset itself (e.g. in contrast to Cora ML, arXiv's test set is smaller and it contains continuous features). However, the trend that large graphs are more fragile becomes more radical if we consider local attacks. For local attacks on large graphs, we observe that even small budgets suffice to fool almost all nodes. On small graphs, PPRGo already seems to be quite an effective defense in comparison to a Vanilla GCN (see Fig. 8). In contrast, without our Soft Median and particularly on the large graphs, it is easy to flip the prediction of basically every node (also compare with Table F.6). We leave a detailed and rigorous study of this relationship for future work.