# OpenReview forum: "Robustness of Graph Neural Networks at Scale"
_NeurIPS.cc/2021/Conference — NeurIPS 2021 Poster_

### Official Review · Reviewer_9yU5 · 2021-07-02

**Rating:** 7
**Confidence:** 3

**Summary:**

This paper studies the robustness of graph neural networks on a massive scale compared to the existing robustness work that focuses on smaller graphs, in terms of three perspectives: surrogate loss, attack, and defense.
- The firstly proposed surrogate loss can be used to enhance the strength of adversarial attacks.
- The authors propose the scalable attack method for large graphs, which is implemented with sparse operations for reducing space complexity by considering a subset of entire adjacency elements when optimizing networks.
- Lastly, the authors propose the scalable yet robust function for aggregating neighboring information on graph neural networks, which is adopted from the previous Soft Medoid method for the scalable purpose.

**Limitations And Societal Impact:**

The authors clearly outline the limitations and potential societal impact of their work, and I am completely satisfied with them.

**Main Review:**

### Strengths
- Compared to existing robustness studies that mostly deal with smaller graphs, this paper provides valuable insights on robustness of large-scale graphs. Interestingly, the proposed attack methods can attack entire 2 million nodes at once. Also, the defense method can reduce the strength of attacks on large and even small graphs, while requiring minimal complexities. I believe this large-scale robustness framework is a good contribution to the graph neural network community, and can bring interesting follow-up work on a real-world scale.
- This paper is well written and well polished, and especially most of the claims are supported with empirical results. Also, a theoretical definition of the global optimum of surrogate loss, and the proposed loss function under this theoretical ground are technically strong.
- While this paper focuses on large-scale graphs, the proposed surrogate loss can be further used for general purposes, since it shows good performances on even small graphs.
- The authors provide a well-structured source code.

### Weaknesses
- While the authors claim that the proposed Masked Cross Entropy and tanh Margin losses improve the strength of attacks more on larger graph cases, the empirical results do not support this. First of all, the results of surrogate loss in Figure 6 are obtained by the smallest graph in the paper. Also, additional experiments in the supplementary file (Figure 15) show that existing losses, such as NC and elu Margin, are more effective compared to the proposed losses on some datasets (CoraML and Citeseer with FGSM and arXiv ML with PR-GCN).
- Regarding defense, while the proposed Soft Median coupled with PPRGo clearly shows good robustness performances, the Soft Median coupled with GDC is generally weak compared to the baseline, Soft Medoid coupled with GDC. I agree with effective scalability of the authors' method, but it would be better to clearly discuss the result and reason of degenerating performances of authors' one.

### Related work
- I am curious about why the authors do not experimentally compare their method against the existing large-scale adversarial attack method [1]. While Li et al. [1] only consider local attacks, it could be directly compared with the authors' local adversarial attack method in the Robustness w.r.t. local attacks paragraph of Section 5.

### Clarity
- I am confused about the relationship between density, perturbation, and margin in section 2 associated with figure 2. It would be better to have few explicit sentences for explaining the intrinsic meaning of the figure.
- In figure 3 (b), the block size is most small for 1.0E+07. Is it correct?
- It is hard to understand equation 6. There are no details on $\pi$ function, encoder, and combination of PPRGo. It would be better to more formally explain it in the supplementary file, if there is a space issue.

### Typo
- In figure 8, the caption indicates (d), (e), and (f), while the actual figure only includes two sub-figures: (a) and (b).

[1] Li et al. Adversarial attack on large scale graph, arXiv 2020.


**Time Spent Reviewing:**

7 hours

---

> ### Author Response · Authors · 2021-08-10
> **Author Response to Reviewer 9yU5**
>
> We thank you for the positive feedback and now discuss the points made. If any further questions arise please let us know.
>
> ## W) Weaknesses
>
> ### W1) Surrogate Losses
>
> #### W1.1) Relationship between scale and surrogate losses
>
> We identified the counter-intuitive behavior of CE and CW while running our attacks on large graphs with tiny budgets. This motivated us to study this topic in the first place. Moreover, the absolute difference is much more significant on our large graphs. For example, with a budget of $\epsilon=0.1$ on GCN on arXiv (Figure 15f) the accuracy degrades from 72\% to 23\% if we use the tanh margin loss. With CE loss it only drops to 44\% (*21\% difference*). On the small Cora ML dataset, we observe a drop from 82\% to 66\% with the tanh margin loss (everything else kept equal). With CE we only observe 75\% (*9\% difference*).
>
> Motivated by these observations, our analysis is intended for a small budget on large graphs ($\frac{\Delta}{n} \to 0$). As long as we do not scale $\Delta$ accordingly with $n$, the issue with previous losses becomes more and more significant the larger the graph is. On the other hand, you are also right that this behavior is also apparent on "small" graphs.
>
> #### W1.2) Figure 15: tanh margin / MCE not always the best
>
> In Figure 15, one needs to be more cautious before drawing conclusions. Please note that for the small datasets we also plot much higher budgets (up to $\epsilon = 1.0$), while our losses were designed for small budgets. Our main conclusion from this figure is that indeed and without any exception our losses perform the best for small budgets. ArXiv is also an exception in the sense that only 29\% of the nodes are test/attacked nodes (typically 90-99\%) while the budget is set relative to the absolute number of edges. If we normalize the budget by the fraction of test/attacked nodes, a budget of $\epsilon=0.25$ on Cora ML relates to a budget of roughly $\epsilon=0.07$ on arXiv. For large budgets on Cora ML and Citeseer, we also see that especially the e.g. elu margin is a very strong/competitive loss. For more details and a similar argument see also lines 584-596.
>
> To address the weaknesses W1), we plan to replace the dataset used in Figure 6 with arXiv or products. Additionally, we will refine the budget $\epsilon$ to be relative to the number of test nodes (specifically Figure 15).
>
> ### 2) Soft Medoid GDC is not stronger than Soft Median GDC
>
> We are not sure to which Table/Figure you are referring. We assume it must be Table 2 or Figure 7 (please correct us if this is not the case).
>
> In Table 2, the differences are not big enough to justify a major discussion. If we compare the results for PR-BCD (largest gap), the Soft Medoid GDC achieves an accuracy after being attacked of $0.761 \pm 0.003$ and the Soft Median GDC $0.752 \pm 0.002$. On the contrary, the clean accuracy increases from $0.819 \pm 0.002$ to $0.824 \pm 0.002$. We argue that one needs to have exactly the same clean accuracy to compare two models with such small differences.
>
> In Figure 7a (PubMed) the difference is not significant (please compare the error bars). On arXiv (Figure 7b), the Soft Median GDC significantly outperforms the Soft Medoid GDC. On Products (Figure 7c), we were not able to fit the Soft Medoid GDC into memory at all. Also for the Soft Median GDC, we had to make compromises (note that we have 32GB memory and already the Vanilla GCN requires roughly 30GB). We also discuss this in lines 346-352.
>
>
> ## R) Related work
>
> ### R1) Comparison with Li et al.'s [1] local attacks
>
> Unfortunately, they only study poisoning / train time attacks while we exclusively study evasion / test time attacks (see lines 36-37). Hence, the approaches are not directly comparable. Moreover, when we ran the experiments, the code was not referenced in the arXiv-version of the paper (the authors updated it right before the submission deadline).
>
> However, comparing their results with ours allows the conclusion that their attack is expected to perform slightly worse. Particularly, their results in Figure 7b suggest that they only slightly outperform Nettack (a rare exception of attack that is used both for poisoning and evasion). On the contrary, from Figure 8b it follows that our local PR-BCD has a clear edge over Nettack (perhaps due to adaptiveness and the arguments in lines 360-367). From a comparison of our attacks, we see that they rely on a surrogate model (i.e. their attack is not adaptive in the sense of [A]) and they only allow their attack to perturb a subgraph.
>
> [A] - Florian Tramèr, Nicholas Carlini, and Wieland Brendel. 2020. On Adaptive Attacks to Adversarial Example Defenses. In NeurIPS 2020. 1633-1645.
>
>
> ## C) Clarity
>
> ### C1) Explanation of Figure 2
>
> The orange line “Att. nodes (clean)” is most important in this figure and it reflects the distribution of classification margins of attacked nodes before they were attacked. With the CE loss, a very large fraction of the budget is spent on nodes that are misclassified from the very beginning (i.e. margin $\psi$ is negative)---this is bad and leads to weak attacks. In contrast, the tanh margin has a very clear focus on correctly classified nodes. Moreover, note that the CE tends to push "Att. nodes (pert.)", the margin after the attack, close to $\psi \approx -1$. This makes also sense since the gradient is very high for values close to -1 and even becomes infinite for exactly -1. For the tanh margin, we do not observe such behavior. The "Test nodes (clean)" are identical in both figures and are intended to be a reference. We will make this more clear in a revision of this paper.
>
>
> ### C2) In figure 3 (b), the block size is most small for 1.0E+07. Is it correct?
>
> The block sizes in Figure 3b are correct. Please note that each of 10 million edges (i.e. a block size of $b=1.0\mathrm{E+}07$) requires two 8 Byte indices and one 4 Byte float according to which are then duplicated to represent an undirected graph (unfortunately PyTorch’s sparse matrices are currently not more efficient). So already the input size is increased by 400MB. Mostly due to caching for backpropagation, the memory requirements increase further (approximately 10-fold). For example in Table 8, we see that the attack on arXiv with block size $b=1.0\mathrm{E+}07$ requires 4.29 GB of GPU memory. Please let us know if this does not fully address your question.
>
> ### C3) PPRGo: What is $\pi(v)_u$ in Eqn (6)?
>
> $\pi(v)_u = \Pi_{v,u}$ denotes the personalized page rank scores. We indeed missed making this notation clear and will revise it. We will also explain the function, encoder, and aggregation of PPRGo properly in the appendix.

---

> > ### Comment · Reviewer_9yU5 · 2021-08-17
> > **Thank you for your response**
> >
> > I really appreciate your comments, which mostly address my concerns on the related works and clarity as well as the strengths of attacks and defenses. I comment each of them below:
> >
> > * For the attack with the proposed Masked Cross Entropy and tanh Margin losses, as you mentioned, it is more reasonable to replace the datasets considered in Figure 6 with larger datasets. Also, I think that your intention -- an analysis of attacks with a small budget on large graphs -- should be mentioned for this experiment. When I first saw and interpreted the results in Figure 15, I understood that existing losses, such as NC and elu Margin, make more significant performance drops against the proposed losses, while this is my misinterpretation though.
> >
> > * For the comparisons between Soft Medoid GDC and Soft Median GDC, I referred to Table 2. There are no overlaps of the averages with considering variances between the baseline (Soft Medoid GDC) and the proposed model (Soft Median GDC), which seems quite significant to me. More specifically, in Line 349, the authors describe that the Soft Mediod GDC has a limitation on scaling, however after comparing the Soft Mediod GDC with the author's method (Soft Median GDC) in Table 2, I notice that there is indeed a trade-off to remark between performance and scaling. Contrary to this observation, I feel that the proposed method is the best among the others even in the performance, based on the descriptions in Paragraph: Robustness w.r.t. global attacks.
> >
> > * For the related work, I now understand that the focus of attack is clearly different, however, if applicable, it might be beneficial to compare the architecture on the related work [1] while excluding the concept of the setting (poisoning or evasion). This means, if the proposed method outperforms the architecture of the related large-scale attack, then the strength of the proposed method would be much more significant.
> >
> > * For clarity, thank you for your detailed explanations.

---

> > > ### Author Response · Authors · 2021-09-02
> > > **Additional experiments comparing our PR-BCD with Li et al.'s SGA**
> > >
> > > We apologize for the delayed response but we took your advice seriously and we now compare ours to Li et al. [21] (you referenced them with [1]). We also extended our scope to local poisoning attacks.
> > >
> > > Following Li et al. [21] and Nettack [42], we use the (local) attacks to run an evasion attack and then use the perturbed adjacency matrix for retraining (i.e. a poisoning attack). In the tables below, we now report the results on Cora ML and Citeseer and will add larger datasets in a revision of the paper (with the relative budget $\epsilon$ to the node degree).
> > >
> > > Our PRBCD seems to be substantially stronger than SGA. We hypothesize that adaptive/direct attacks (surrogate and victim match) are typically stronger than transfer attacks. Moreover, we even outperform SGA on an SGC model (recall that SGA is designed for SGC). This demonstrates how generally applicable our PRBCD is (without modifications).
> > >
> > > Scaling SGA [21] to graphs of ogbn-papers100M's size will fail for two reasons. (1) SGA requires the [`k`-hop subgraph around the attacked node $t$](https://github.com/EdisonLeeeee/GraphGallery/blob/master/graphgallery/attack/targeted/pytorch/sga.py#L141), denoted as $(V_s, E_s)$ in [21]. $(V_s, E_s)$ might represent a large fraction of the graph if having high-degree neighbors and, in the worst case, covers the entire graph. Additionally, (2) for potential edge additions, SGA requires obtaining the gradient of all edges that belong to the second most likely class. If $N$ is large and $C$ is small this becomes very expensive (batching might help). For example on ogbn-produtcs, we use SGA to attack a random node of medium degree (180). To obtain the gradient to the edges of the `k`-hop subgraph (line 8 in Algorithm 1 in [21]) we require 3.6 GB for a SGC with `k=2` and 9.8 GB for `k=3` (`k=3` is the default on this dataset). With `k=4` more than 32 GB are required (too large for most GPUs). If we attack PPRGo with our local PRBCD, we do not have restrictions similar to (1) and (2) (see §3 and §A.3.3). Hence, our LocalPRBCD is much more scalable. In our experiments, we require less than 3 GB to attack ogbn-papers100M (two orders of magnitude larger than ogbn-produtcs).
> > >
> > > ### PR-BCD (ours) on Cora ML
> > >
> > > |    | Dataset   | Surrogate   | Victim   | Attack Type   | $\epsilon=0.1$   | $\epsilon=0.25$   | $\epsilon=0.5$   | $\epsilon=1.0$   |
> > > |---:|:----------------------------|:--------------------|:-----------------|:----------------------|:------------------|:-------------------|:------------------|:------------------|
> > > |  0 | Cora ML                     | Vanilla GCN         | Vanilla GCN      | evasion               | 0.283 $\pm$ 0.041 | 0.375 $\pm$ 0.044  | 0.650 $\pm$ 0.044 | 0.958 $\pm$ 0.018 |
> > > |  1 | Cora ML                     | Vanilla GCN         | Vanilla GCN      | poisoning             | 0.333 $\pm$ 0.043 | 0.458 $\pm$ 0.046  | 0.792 $\pm$ 0.037 | 0.967 $\pm$ 0.016 |
> > > |  2 | Cora ML                     | Vanilla SGC         | Vanilla SGC      | evasion               | 0.292 $\pm$ 0.042 | 0.450 $\pm$ 0.046  | 0.567 $\pm$ 0.045 | 0.967 $\pm$ 0.016 |
> > > |  3 | Cora ML                     | Vanilla SGC         | Vanilla SGC      | poisoning             | 0.300 $\pm$ 0.042 | 0.500 $\pm$ 0.046  | 0.658 $\pm$ 0.043 | 0.958 $\pm$ 0.018 |
> > >
> > > ### SGA [21] on Cora ML
> > >
> > > |    | Dataset   | Surrogate   | Victim   | Attack Type   | $\epsilon=0.1$   | $\epsilon=0.25$   | $\epsilon=0.5$   | $\epsilon=1.0$   |
> > > |---:|:----------------------------|:--------------------|:-----------------|:----------------------|:------------------|:------------------|:------------------|:------------------|
> > > |  0 | Cora ML                     | Vanilla SGC         | Vanilla GCN      | evasion               | 0.192 $\pm$ 0.036 | 0.358 $\pm$ 0.044 | 0.508 $\pm$ 0.046 | 0.825 $\pm$ 0.035 |
> > > |  1 | Cora ML                     | Vanilla SGC         | Vanilla GCN      | poisoning             | 0.275 $\pm$ 0.041 | 0.467 $\pm$ 0.046 | 0.642 $\pm$ 0.044 | 0.950 $\pm$ 0.020 |
> > > |  2 | Cora ML                     | Vanilla SGC         | Vanilla SGC      | evasion               | 0.233 $\pm$ 0.039 | 0.367 $\pm$ 0.044 | 0.433 $\pm$ 0.045 | 0.950 $\pm$ 0.020 |
> > > |  3 | Cora ML                     | Vanilla SGC         | Vanilla SGC      | poisoning             | 0.275 $\pm$ 0.041 | 0.475 $\pm$ 0.046 | 0.617 $\pm$ 0.045 | 0.975 $\pm$ 0.014 |
> > >
> > >
> > > ### PR-BCD (ours) on Citeseer
> > >
> > > |    | Dataset   | Surrogate   | Victim   | Attack Type   | $\epsilon=0.1$   | $\epsilon=0.25$   | $\epsilon=0.5$   | $\epsilon=1.0$   |
> > > |---:|:----------------------------|:--------------------|:-----------------|:----------------------|:------------------|:-------------------|:------------------|:------------------|
> > > |  0 | Citeseer                    | Vanilla GCN         | Vanilla GCN      | evasion               | 0.125 $\pm$ 0.030 | 0.417 $\pm$ 0.045  | 0.658 $\pm$ 0.043 | 0.850 $\pm$ 0.033 |
> > > |  1 | Citeseer                    | Vanilla GCN         | Vanilla GCN      | poisoning             | 0.217 $\pm$ 0.038 | 0.525 $\pm$ 0.046  | 0.775 $\pm$ 0.038 | 0.942 $\pm$ 0.021 |
> > > |  2 | Citeseer                    | Vanilla SGC         | Vanilla SGC      | evasion               | 0.175 $\pm$ 0.035 | 0.392 $\pm$ 0.045  | 0.625 $\pm$ 0.044 | 0.908 $\pm$ 0.026 |
> > > |  3 | Citeseer                    | Vanilla SGC         | Vanilla SGC      | poisoning             | 0.208 $\pm$ 0.037 | 0.492 $\pm$ 0.046  | 0.767 $\pm$ 0.039 | 0.917 $\pm$ 0.025 |
> > >
> > > ### SGA [21] on Citeseer
> > >
> > > |    | Dataset   | Surrogate   | Victim   | Attack Type   | $\epsilon=0.1$   | $\epsilon=0.25$   | $\epsilon=0.5$   | $\epsilon=1.0$   |
> > > |---:|:----------------------------|:--------------------|:-----------------|:----------------------|:------------------|:------------------|:------------------|:------------------|
> > > |  0 | Citeseer                    | Vanilla SGC         | Vanilla GCN      | evasion               | 0.058 $\pm$ 0.021 | 0.342 $\pm$ 0.043 | 0.500 $\pm$ 0.046 | 0.725 $\pm$ 0.041 |
> > > |  1 | Citeseer                    | Vanilla SGC         | Vanilla GCN      | poisoning             | 0.275 $\pm$ 0.041 | 0.542 $\pm$ 0.046 | 0.758 $\pm$ 0.039 | 0.925 $\pm$ 0.024 |
> > > |  2 | Citeseer                    | Vanilla SGC         | Vanilla SGC      | evasion               | 0.142 $\pm$ 0.032 | 0.350 $\pm$ 0.044 | 0.550 $\pm$ 0.046 | 0.850 $\pm$ 0.033 |
> > > |  3 | Citeseer                    | Vanilla SGC         | Vanilla SGC      | poisoning             | 0.233 $\pm$ 0.039 | 0.492 $\pm$ 0.046 | 0.742 $\pm$ 0.040 | 0.975 $\pm$ 0.014 |
> > >
> > > The results differ slightly from Li et al.'s paper because we follow the evaluation protocol of Nettack (see lines 356-359). In Nettack's setup, e.g. correctly classified nodes of high confidence are overrepresented. We carefully verified the correctness of our SGA implementation and will open source the code as well as configuration.

---

> > > > ### Comment · Reviewer_9yU5 · 2021-09-03
> > > > **I really appreciate your additional experimental results**
> > > >
> > > > Thank you very much for your efforts on experimental comparisons against the relevant work. I strongly believe that the additional results make the paper much more solid. I think this is the last comment that I sincerely hope the authors carefully reflect all the missing or unclear parts pointed out by reviewers in their next revision. Best of luck.

---

### Official Review · Reviewer_1HMw · 2021-07-07

**Rating:** 6
**Confidence:** 4

**Summary:**

The paper studies adversarial attacks against graph neural networks as well as adversarial robustness at scale.
The authors first propose two first-order optimization attacks, i.e., PR-BCD and GR-BCD. The authors then propose an empirical scalable defense based on soft median aggregation. Both attacks and defenses are evaluated on benchmark datasets and show their effectiveness.

Strengths
+The studied problem is important
+Propose a novel attack loss function
+Evaluation is extensive

Weaknesses
-The paper studies white-box attacks
-Algorithm details are unclear
-The proposed defense is empirical

**Limitations And Societal Impact:**

The authors addressed the limitations of their work

**Main Review:**

—-Threat model

The paper studies white-box attacks, which is not practical from my perspective.

—Algorithm details

From Figure 2, both tanh margin and cross entropy spend a lot of budgets on misclassified nodes. What’s the takeaway of Figure 2?

Is Assumption 1 satisfied when performing global attacks, as they aim to reduce the performance of all nodes and could largely perturb the graph (given enough budget) which involves a substantial number of nodes?

What’s the key difference between the proposed PR-BCD attack in Algorithm 1 and the optimization-based attack in [36]?

PR-BCD needs to sort the probability mass before removing entries with the lowest probability mass. Sorting a huge vector is time-consuming. How could this sorting be scalable to huge graphs?

How is the PPR matrix calculated in huge graphs, as it needs to compute the inverse of a huge matrix?

What’s $\pi(v)_u$ in Eqn (6)?

My main concern is that the proposed soft median based defense is empirical and does not have theoretical guarantees. As the paper mainly focuses “robustness” of GNNs, I doubt that empirical defenses provide a false sense of robustness.



—Evaluation

The term “perturbed accuracy” makes me confused.

No global attack results on Papers 100M.


—Other comments:

In conclusion,  … our novel Soft Median which is differentiable as well as provably robust. => This is inaccurate.




**Time Spent Reviewing:**

6

---

> ### Author Response · Authors · 2021-08-10
> **Author Response to Reviewer 1HMw part 2**
>
> ## 3) Explanation of Figure 2
>
> The orange line “Att. nodes (clean)” is most important in this figure and it reflects the distribution of classification margins of attacked nodes before they were attacked. With the CE loss, a very large fraction of the budget is spent on nodes that are misclassified from the very beginning (i.e. margin $\psi$ is negative) - this is bad and leads to weak attacks. In contrast, the tanh margin has a very clear focus on correctly classified nodes. Moreover, note that the CE tends to push "Att. nodes (pert.)", the margin after the attack, close to $\psi \approx -1$. This makes also sense since the gradient is very high for values close to -1 and even becomes infinite for exactly -1. For the tanh margin, we do not observe such behavior. The "Test nodes (clean)" are identical in both figures and are intended to be a reference. We will make this more clear in a revision of this paper.
>
>
> ## 4) Does Assumption 1 hold for large budgets?
>
> Our Assumption 1 relies on the fact that we are looking at *small budgets* (we say "Particularly on large graphs with small budgets, $\frac{\Delta}{n} \to 0$, [...]"). For large budgets, this assumption is likely false. However, since we study adversarial robustness, we are especially interested in small perturbations and hence emphasize small budgets.
>
>
> ## 5) Key difference between PR-BCD (Algorithm 1) and PGD [36]?
>
> The main difference between ours (PR-BCD) and $L_0$-PGD / the topology attack of [36] is that  $L_0$-PGD requires to hold all $\Theta(n^2)$ elements of the adjacency matrix in main memory. [36]’s  $L_0$-PGD comes with this restriction since each entry in the (perturbed) adjacency matrix is a parameter they explicitly optimize over. With [36], right after the very first gradient update, the perturbed adjacency matrix will not be sparse anymore.
>
> We overcome this limitation leveraging recent advances in large-scale optimization, namely Randomized Block Coordinate Descent [24, 25, 32]. In each step, we optimize over a random subset of parameters (which we call “block”, does not need to be contiguous). We additionally propose a strategy to maintain a sparse set of perturbations $\mathbf{P}$ throughout the attack in a survival-of-the-fittest manner (i.e. we keep the highest weights). Our PR-BCD specifically relies on the continuous relaxation proposed in [36] (lines 8 and 16 in Algorithm 1). We describe the difference/commonalities at the beginning of the “PR-BCD” paragraph at §3. We will add a side-by-side comparison of [36]’s pseudo code and our Algorithm 1 in a revision of this paper.
>
> In conclusion, [36] comes with prohibitively high cost on large graphs (see Figure 1, Table 1, “Time and memory cost” paragraph in §5, and §A.5.2). Note that none of the referenced global attacks relying on optimization (e.g. [7, 33, 41, 42]) have a memory complexity better than ours.
>
>
> ## 6) Is sorting the bottleneck?
>
> You are right that sorting can become expensive on extremely large graphs. Each epoch of our global PR-BCD attack runs in $\mathcal{O}(b \log(b))$ time (with block size $b$, additionally to the attacked GNN). However, the (arg-) sort of a float32 vector with 250 million entries requires roughly 800 ms (GeForce GTX 1080 Ti, roughly 9 GB used). On a 32 GB V100 we can (arg-) sort such a vector with 1 billion entries in 2 seconds. In our experiments, we only sort smaller vectors $\mathcal{O}(b)$. We report the memory and time cost in Tables 8 and 9. For instance, we sort 10 million entries on arXiv while an entire attack epoch requires 1.5 seconds. To use a GNN on datasets 1-2 magnitudes larger than Papers 100M, it certainly makes sense to go for a distributed setup anyways, to overcome memory limitations.
>
>
> ## 7) PPR matrix complexity
>
> We present the matrix inverse PPR formula because we use this formulation to derive an efficient (constant time) and differentiable update procedure. On large graphs, we calculate the PPR scores with Andersen’s algorithm [F] (following PPRGo). The runtime of Andersen's algorithm is linear in the number of nodes and also depends on the desired error and teleport probability. We acknowledge the importance for our work and will incorporate this feedback.
>
> [F] - Reid Andersen, Fan Chung, and Kevin Lang. 2006. Local graph partitioning using pagerank vectors. In FOCS. 475–486.
>
>
> ## 8) PPRGo: What is $\pi(v)_u$ in Eqn (6)?
> $\pi(v)\_u = \Pi\_{v,u}$ denotes the personalized page rank scores. We indeed missed making this notation clear and will revise it. We will also explain the function, encoder, and aggregation of PPRGo properly in the appendix.
>
>
> ## 9) The term “perturbed accuracy”
>
> With perturbed accuracy, we denote the accuracy after perturbing/attacking the graph. We tried to avoid metrics such as success rates etc. because we do not study adversarial robustness from the attacker's perspective. We rather want to assess how stable the target metric (i.e. accuracy) is under small perturbations of the input. We will improve the introduction of this term in a revision or switch to the alternative "accuracy after attack".
>
>
> ## 10) Global attack on Papers 100M.
>
> Our attack is a huge leap forward in terms of scalability as we scale our global attacks to a graph (Products) with 120 times more nodes and 1400 times more edges than PubMed (the currently widely used “large scale dataset” for global attacks/defenses). To put this in perspective, we scale our *global* attacks to graphs ten times larger than the previous work about an adversarial attack on large scale graphs [21] scales their *local* attack. We achieve this without any constraints on the receptive field etc. and hence have a maximally strong threat model.
>
> Scaling to even larger datasets comes with additional challenges and is out of scope of this work. For example if we consider Papers 100M, common GPUs do not provide enough memory to obtain the prediction of all nodes and then backpropagate towards the existing nodes’ features (i.e. also full-batch training on one GPU is impossible).
>
>
> ## 11) Other comments
>
> While the statement in line 387 is true in its literal sense (provably optimal breakdown point of 0.5), we agree that it could be misleading if taken out of context. Hence we propose to change the formulation to "we propose a scalable defense using our novel Soft Median, which is differentiable and has the best possible breakdown point of 0.5".

---

> > ### Comment · Reviewer_1HMw · 2021-08-20
> > **Robustness of Graph Neural Networks at Scale**
> >
> > Thanks for addressing my comments!

---

> ### Author Response · Authors · 2021-08-10
> **Author Response to Reviewer 1HMw part 1**
>
> We thank you for acknowledging the importance of the studied problem and the extensive feedback focusing on our scalable attack/defense. We first discuss your major concerns with our work (robustness of the defense and practicality of white-box attacks) and then the remaining points. We hope to provide interesting as well as convincing insights and are happy to follow up on any further points.
>
> ## 1) Soft Median and empirical defenses
>
> **We evaluate using adaptive attacks.**
> In general, we agree that assessing the adversarial robustness of an empirical defense with attacks, can potentially provide a wrong sense of robustness (see lines 209-211 and 295-296). For this reason, we follow the advice in [A] as much as possible and evaluate our defense with adaptive attacks (Figure 8, Table 10 and §A.5.4). These attacks are adaptive since we differentiate through the Soft Median as well as the personalized page rank calculation (i.e. the attack can also attack these components specifically). Even though it is best practice to evaluate an empirical defense on strong adaptive attacks [A], none of the following empirical defenses for GNNs follow this principle [13, 14, 16, 17, 29, 33, 34, 38, 40] (e.g. they often attack a *surrogate model* and then transfer the perturbed adjacency matrix to their defense).
>
> **Our empirical defense can improve certified robustness.**
> We state in lines 297-298 that “Similarly to the Soft Medoid we expect that our Soft Median can also improve the certified robustness [14, 27]”. This means that such a defense can be used e.g. in combination with randomized smoothing [2] to obtain a significantly greater provable adversarial robustness. In the subsequent table, we show the "Accumulated certificates" obtained by randomized smoothing (same setup as in Table 2 in [14]). Even though our defense does not come with a robustness guarantee, we show that it can lead to increased *provable* robustness.
>
> | | Architecture ↓ \ Accum. certificate → | Add & Del.    | Add. | Del. |Accuracy  |
> |----------|------------------------|---------------------|------|------|----------|
> | Cora ML  | Soft Median GDC (ours) | **5.7**          |**0.66**|**4.9**|**0.833** |
> | Cora ML  | Vanilla GCN            | 1.84                | 0.21 | 4.41 | 0.823    |
> | Cora ML  | Soft Medoid GDC        | 5.5                 | 0.64 | 4.78 | 0.814    |
> | Citeseer | Soft Median GDC (ours) | **4.43**          |**0.57**| 4.31 |**0.728** |
> | Citeseer | Vanilla GCN            | 1.24                | 0.11 | 3.88 | 0.710    |
> | Citeseer | Soft Medoid GDC        | 3.64                | 0.49 |**4.33**| 0.705  |
>
> **There is no approach for certifiable robustness at scale.**
> We argue that similar to [11, 42], our work breaks new ground by being the first to study adversarial robustness of GNNs at scale. For the same reason, we cannot rely on existing certification techniques to show the efficacy of our defense at scale. Unfortunately, even randomized smoothing [2] (experiment above) on massive graphs is very expensive.
>
> **Empirical defenses can increase the robustness**
> In case you suspect that *all* empirical defenses provide a false notion of robustness, we would like to mention that adversarial training is a heuristic defense that has been consistently shown to work in practice (especially in the image domain). Even though strong, adaptive attacks rendered many empirical defenses ineffective [A], this does not imply that all empirical defenses are ineffective w.r.t. increasing empirical and provable robustness. As a counterexample, we want to mention adversarial training again or the experiment above (randomized smoothing).
>
>
> ## 2) White-box attacks are practical
>
> We study adversarial robustness from the perspective of a defender, not as an adversary who attacks e.g. an application for their benefit. From this perspective, we identified three main reasons to study white-box attacks on GNNs at scale (besides the novelty): (1) white-box attacks are the most powerful threat model, (2) we can use white-box attacks to enhance the understanding of the stability of a model, and (3) strong attacks are required to assess the efficacy of defenses.
>
> (1) If we can show that an algorithm is robust w.r.t. to *strong* white-box attacks (the worst case), then we can also assume robustness w.r.t. less powerful threat models (i.e. the white box threat model is strictly the most powerful). Note that a (strong) attack provides an upper bound on the robustness of a GNN. Until *exact certification* is solved (i.e. no/small relaxation gap, which is presumably NP-hard [B, C, D]), we need these two complementary views. Otherwise, certification just provides a lower bound on the robustness.
>
> (2) A very practical use case for our attack is the *assessment of GNNs* on large-scale graphs. Adversarial attacks also enhance the understanding of how such an algorithm behaves to "worst-case noise". This is a desired property for the company and users in commercial applications. Please note that there are certainly multiple internet-scale applications of GNNs (*millions of people are affected by such algorithms in this very moment*, e.g. [37] or [E] where a scientist confirms the Google-internal use of PPRGo at scale). However, prior to this work, it is entirely unknown how robust or rather fragile GNNs on large graphs are. In Figure 8b, we show for PPRGo that already a tiny budget suffices to flip almost every node's prediction!
>
> (3) One of the findings in the image domain is that strong attacks are essential to "pave the way to more robust models" [A]. Even multiple black-box attacks are insufficient to reveal potential non-robustness [A].
>
> Additionally, to assess the efficacy of black-box attacks, it is best to compare with a white-box attack.
>
>
> ## References
>
> [A] - Florian Tramèr, Nicholas Carlini, and Wieland Brendel. 2020. On Adaptive Attacks to Adversarial Example Defenses. In NeurIPS 2020. 1633-1645.
>
> [B] - Maksym Andriushchenko, and Matthias Hein. 2020. Provably Robust Boosted Decision Stumps and Trees against Adversarial Attacks. In NeurIPS 2019.
>
> [C] - Huan Zhang, Tsui-Wei Weng, Pin-Yu Chen, Cho-Jui Hsieh, and Luca Danie. 2020. Efficient Neural Network Robustness Certification with General Activation Functions. In NeurIPS 2018.
>
> [D] - Tsui-Wei Weng, Huan Zhang, Hongge Chen, Zhao Song, Cho-Jui Hsieh, D. Boning, I. Dhillon, and L. Daniel. 2020. Towards Fast Computation of Certified Robustness for ReLU Networks. In ICML 2018. 1633-1645.
>
> [E] - Amol Kapoor. 2020. [Workshop - Mining and Learning with Graphs at Scale: Graph Neural Networks](https://gm-neurips-2020.github.io/). NeurIPS Workshop 2020.

---

### Official Review · Reviewer_CQeL · 2021-07-16

**Rating:** 7
**Confidence:** 2

**Summary:**

This paper addresses the problem of adversarial robustness for GNN regarding attack and defense on large graphs up to 100 million nodes. In terms of attack, they show that existing attack methods are weak for large graphs. To address this, they introduce the Masked Cross Entropy (MCE) which only considers correctly classified nodes on the attack. In addition, they also tackle the issue of handling large graphs which should be scaled with quadratic space complexity of adjacency matrix, and introduce the Projected Randomized Block Coordinate Descent (PR-BCD) based on Bernoulli sampling. In terms of defense, the authors propose the Soft median with differentiable relaxation of the Median.

**Limitations And Societal Impact:**

The presentation quality can be improved in revision, as noted in the Clarity section of the main review.

**Main Review:**

Originality
====
In this paper, the authors tackle the problem of GNN robustness in a large graph, which is important for GNN robustness but not addressed so far. This paper first points out the problem of existing attacks with their weakness in a large graph and suggests a theoretically guaranteed method with improved surrogate loss and scalable optimization method. This paper also provides the cue of scalable defense for a large graph.

Quality
====
The method sound convincing and the theoretical foundation seems solid. There are diverse comprehensive analyses that are helpful to understand the claim of the paper. The experimental setup covers the diverse sizes of graphs ranging from 2.8k to 111M against various attack and defense methods.

Significance
====
In experiments, they show that their attack and defenses are reasonably fast and require much less memory that can cover sufficiently large graphs like Papers 100M. In quantitative analysis, they show that the proposed attack significantly drops the performance against other margin losses in Figure 6 and Table 2 and the proposed Soft Median defense resists the attacks much better than the baselines.

Clarity
====
The problem and method are generally well-motivated and clear. However, the overall presentation is disappointing.
- In methodology sections (Section 2,3,4), subsections are rarely used and most topics are divided by only paragraph and bold title. It makes the paper looks fuzzy.
- There is no abstractive description of the overall methods. I suggest authors add the comprehensive figure to outline the entire method to help authors better understand the overall methods.
- The caption of Figure 7 is insufficient and Figure 7 is not referenced in the main paper.
I think most of the presentation issues are due to the page limit of the main paper. I hope it will be more clarified in revision.


**Time Spent Reviewing:**

15

---

> ### Author Response · Authors · 2021-08-10
> **Author Response to Reviewer CQeL**
>
> We thank you for the comments regarding the clarity of our work and we address them in the following.
>
> ### 1) Usage of subsections
>
> We will work with subsections as much as possible in a revision of this paper. As you mention correctly later, we gradually removed subsections to meet the space requirements.
>
> ### 2) Methods overview
>
> We will improve our overview in the introduction to ease understanding and navigation. We are thankful for further suggestions.
>
> ### 3) Caption of Figure 7
>
> Figure 7 is referenced in line 341. We suggest changing the caption to "Figure 7: PR-BCD (DICE dashed) on the large datasets (transfer) where the perturbed accuracy denotes the accuracy after attacking with budget $\Delta = \epsilon \cdot m$".

---

> > ### Comment · Reviewer_CQeL · 2021-08-23
> > **Thanks for the response**
> >
> > Thank you for the response. Most of my concerns regarding clarity are resolved by the author's response. I hope the authors address them in the camera-ready version if accepted.

---

### Official Review · Reviewer_hJTe · 2021-07-16

**Rating:** 6
**Confidence:** 5

**Summary:**

This paper studies adversarial attack and defense of GNNs at scale. The authors investigate the relationship between surrogate losses and accuracy, and scalable attack and defense algorithms. Some theoretical intuitions and extensive empirical evaluation are provided to support the claims.

**Limitations And Societal Impact:**

This paper has a good discussion on limitations and societal impact.

**Main Review:**

Strength
- Overall, this paper is clearly-written. While this paper wraps a lot of content, the high-level overview in the Introduction makes it easier for the reader to parse the content.
- As far as I know, adversarial attack and defense for large-scale GNNs have not been well-explored and this is an interesting topic.
- I find the analysis of the surrogate losses very interesting. The insight that the loss should guide the attack to focus on the correctly-classified boundary points could be helpful for the development of future attack methods.
- This paper has an extensive empirical evaluation.


Weakness
- The proposed attack is a white-box attack assuming almost everything (labels, gradients, etc) is available to the attacker. A white-box attack under the large-scale setup is rarely practical, which makes the proposed attack method less interesting.
- The assumption 2 is very unrealistic. The budget needed to flip the prediction of a node depends on the node features and its neighborhoods. Assumption 2 (and the proof of Proposition 1) assumes a uniform function g across all nodes. This is impossible, even approximately. With that said, while the significance of Proposition 1 is weakened due to the unrealistic assumption, the two properties defined in Definition 1 are still intuitively desirable.
- A table summarizing the theoretical computation complexity of the proposed methods and previous methods will be helpful.

Additional questions
- The study of surrogate losses does not seem to be relevant to the scale of GNNs. Is there anything that I missed?
- Could you elaborate on the reasons why MCE does not work well for PGD?
- Property (II) and property (B) essentially indicate a "diminishing-return" for the decrease of loss when the margin grows larger. This seems to align with the results in [22]. However, in A.2 the authors state that they mismatch. Could you elaborate more on the reasons? Also, the authors mention in A.2 that "our analysis suggests that the CW loss should be effective once the budget is large enough", but I did not find the relevant analysis. Could you point out where it is located?

Minor suggestions
- The naming of "budget-aware" loss in Definition 1 is a little bit confusing, since the definition itself does not involve the budget. Losses of this type favor correctly-classified boundary points, and has a diminishing slope when going further from the boundary. A more straightforward name such as "diminishing-slope" loss would be more informative.
- A typo: line 124, the subscript n should be i.


**Time Spent Reviewing:**

4

---

> ### Author Response · Authors · 2021-08-10
> **Author Response to Reviewer hJTe part 2: Additional questions**
>
> ## Q) Additional questions
>
> ### Q1) Relationship between scale and surrogate losses
>
> We identified the counter-intuitive behavior of CE and CW while running our attacks on large graphs with tiny budgets. This motivated us to study this topic in the first place. Moreover, the absolute difference is much more significant on our large graphs. For example, with a budget of $\epsilon=0.1$ on GCN on arXiv (Figure 15f) the accuracy degrades from 72\% to 23\% if we use the tanh margin loss. With CE loss it only drops to 44\% (*21\% difference*). On the small Cora ML dataset, we observe a drop from 82\% to 66\% with the tanh margin loss (everything else kept equal). With CE we only observe 75\% (*9\% difference*).
>
> Motivated by these observations, our analysis is intended for a small budget on large graphs ($\frac{\Delta}{n} \to 0$). As long as we do not scale $\Delta$ accordingly with $n$, the issue with previous losses becomes more and more significant the larger the graph is. On the other hand, you are also right that this behavior is also apparent on "small" graphs.
>
> ### Q2) Why does MCE not work well with PGD?
>
> The main reason is the zero gradient if classification margin $\psi < 0$ in the MCE (and CW) loss: $\text{CW} = \min(\max\_{c \ne c^\ast} \mathbf{z}\_{c} - \mathbf{z}\_{c^\ast}, 0)$ [4, 36]. To fully explain the reasons we need to dive into the PGD update step in epoch $t$:
>
> $\mathbf{p}\_{t} = \Pi\_{\mathbb{E}[\text{Bernoulli}(\mathbf{p}\_t)] \le \Delta} \left[ \mathbf{p}\_{t-1} + \alpha\_{t-1} \nabla \mathcal{L}(\mathbf{p}\_{t-1}, \dots) \right]\\,.$
>
> We can rewrite this expression to
>
> $\mathbf{p}\_{t} = \Pi\_{[0, 1]}  ( \mathbf{p}\_{t-1} \\,\\, \underbrace{+ \alpha\_{t} \nabla \mathcal{L}(\mathbf{p}\_{t-1}, \dots)}\_{\text{gradient update}} \underbrace{- \eta\_t \textbf{1}}\_{\text{correction}} )$
>
> where $\Pi\_{[0, 1]}$ clamps the values to the range $[0, 1]$ ($\Rightarrow \mathbf{p}\_{t} \in [0,1]^b$) and $\eta_t$ is chosen s.t. $\mathbb{E}[\text{Bernoulli}(\mathbf{p}\_t)] \le \Delta$ (i.e. $\sum \Pi\_{[0, 1]}(\mathbf{p}\_t) \le \Delta$). There are two competing terms: 1) the gradient update $\alpha\_{t-1} \nabla \mathcal{L}(\mathbf{p}\_{t-1}, \dots)$ and 2) the correction $\eta\_t \textbf{1}$ (typically lowers all weights in $\mathbf{p}\_{t}$). For reasonable parameter choices, the potential perturbations in $\mathbf{p}\_{t}$ are competing since our budget is limited (i.e. to maximize the loss we would like to flip more edges than budget we have). Then, after some epochs ($t > t\_0$), we will have $\eta\_t > 0$ and subtract $\eta\_t$ from each element in $\mathbf{p}\_{t-1}$.
>
> Now if we choose a loss $\mathcal{L}$ (e.g. CW or MCE loss) that has zero gradient, as soon as a node $v$ is misclassified ($\psi_v < 0$), the responsible edge(s) will not benefit from a "gradient update" anymore but $\eta_t > 0$ is still subtracted. So after some iterations node $v$ will be again correctly classified since the required edge flips in $\mathbf{p}_t$ lost weight/strength. This leads to instability.
>
> The symptoms are particularly visible in Figure 9e for the CW loss (after $t_0 = 25$ epochs the accuracy oscillates around 0.7). Moreover, the accuracy for the CW loss (Figure 9 d-f and Figure 10 d-f) are noisier than for the CE or tanh margin losses (other subfigures).
>
> In the revision, we will provide a figure with the loss/training curves for MCE in addition to the ones with CW. Moreover, we will include this discussion about the PGD update procedure.
>
> ### Q3) Diminishing-return
>
> #### Q3.1) Differences to [22]
>
> It is true that [22] also report a discrepancy between the accuracy and "the CW loss". Our intention in §A.2 was to simply acknowledge this and we agree that the word "mismatch" might be too strong (line 570). In addition to the reasons we mention in §A.2, we missed to make clear that we use a slightly different definition for the CW loss than in [22]. We follow Xu et al. [36] $\text{CW} = \min(\max\_{c \ne c^\ast} \mathbf{z}\_{c} - \mathbf{z}\_{c^\ast}, 0)$ (Eq. 6 in [36] with $\kappa=0$) while Ma et al. [22] use what we call margin loss: $\text{Margin} = \max\_{c \ne c^\ast} \mathbf{z}\_{c} - \mathbf{z}\_{c^\ast}$. In contrast to Ma et al., we stop attacking a node once it is misclassified.
>
> To show the differences to their analysis we now discuss the setup and approach of [22]. They select 1\% of nodes based on some centrality score and then gradually increase the feature perturbation of the selected nodes. For the "page rank like" score, they then realize that at some level the margin loss still increases while the accuracy remains constant. So it seems like that if the perturbation budget of the attacked nodes is large enough then their whole receptive field is successfully perturbed. Ma et al. then greedily avoid choosing neighboring nodes and justify this theoretically relying on homophily.
>
> In conclusion, they study how to spread the perturbed nodes over the graph. Instead, we discuss that e.g. with CE most of the budget is spent on nodes that are wrongly classified in the clean graph (Properties I and A). In contrast to Ma et al., we also consider the fact that e.g. the CW loss comes with the risk of unsuccessfully spending all/too much budget on high-confidence nodes (Properties II and B).
>
>
> #### Q3.2) CW loss with large budgets
>
> Considering Xu et al.’s [36] loss definition $\text{CW} = \min(\max\_{c \ne c^\ast} \mathbf{z}\_{c} - \mathbf{z}\_{c^\ast}, 0)$, the CW loss only encourages attacking nodes which are correctly classified. Hence, with a large enough budget, the CW loss should successfully attack a large fraction of nodes that were correctly classified before the attack (or all with unlimited budget). We are happy to make this more explicit in the revised version of our paper.
>
> ## Minor suggestions
>
> We are thankful for your thoughts. We agree and suggest dropping the name "budget-aware" since "diminishing-slope" only covers half of the properties.

---

> > ### Comment · Reviewer_hJTe · 2021-08-23
> > **Thanks for the response.**
> >
> > Dear authors,
> >
> > I appreciate the thorough responses. I'm fully satisfied by the further clarifications of W3 and all the additional questions, and hope they can be incorporated into the revised version.
> >
> > For W1 and W2, I reserve my opinions. However, I would like to make it clear that they are not major obstacles against the acceptance of this paper, as reflected in my overall rating. It is not necessary to make technical improvement on these issues, but I suggest the authors to at least include a discussion on these issues in the revised version.
> >
> > More concretely, for W1, while it is true that white-box attacks tend to be stronger than black-box attacks, it is also well-known that increased robustness often comes at the price of decreased accuracy. So in practice, designing defense techniques targeting at white-box attacks may not always be ideal. Better trade-off between robustness and accuracy may be achieved if we only consider realistic black-box attacks.
> >
> > For W2, I still consider the Assumption 2 unrealistic. Note that Assumption 2 assumes that the function g is uniformly independent of the node features (given $psi$) across all nodes. This is important as your proof for Proposition 1 critically rely on this property. In your response, you mentioned that having "a positive relationship" between $\psi$ and $\Delta$ implies that "the assumption holds in expectation". What's the formal definition of "holds in expectation"?

---

> > > ### Author Response · Authors · 2021-08-30
> > > **Author Response**
> > >
> > > Dear Reviewer,
> > >
> > > We thank you for the careful and comprehensive response. We will make sure to incorporate the valuable feedback provided.
> > >
> > > **W1 - black-/white-box attacks.** We agree with your points, noting additionally that the "optimal" threat model and trade-offs are application specific (e.g. safety-critical vs. ordinary applications).  We will include a discussion in the revised version.
> > >
> > > **W2 - Assumption 2.** By "holds in expectation" we specifically meant that we can rewrite Assumption 2 (and Proposition 1) as stated below. Here, we relaxed Assumption 2 / Proposition 1 s.t. the influence of the actual node attributes etc. is captured through randomness.
> > >
> > > > **Assumption 2** The _expected_ budget required to change the prediciton of node $i$ increases with the margin: $\mathbb{E}[\Delta_i | \psi_i] = g(|\psi_i|)$ for some increasing function $g(|\psi_i|) \ge 1.$
> > >
> > > > **Proposition 1** Let $\mathcal{L}'$ be the surrogate for the 0/1 loss $\mathcal{L}\_{0/1}$ used to attack a node classification algorithm $f\_{\theta}(\mathbf{A}, \mathbf{X})$ with a global budget $\Delta$.
> > > Additionally to Assumptions 1 and 2, suppose the adversary perturbs the chosen node until it is misclassified. We then obtain the global optimum of
> > > >
> > > > $\max\_{\tilde{\mathbf{A}}\text{ s.t. }\|\tilde{\mathbf{A}} - \mathbf{A}\|\_0 \le \Delta} \mathbb{E}[\mathcal{L}\_{0/1}(f\_{\theta}(\tilde{\mathbf{A}}, \mathbf{X}))]$
> > > >
> > > > through greedily attacking the nodes in order $\frac{\partial \mathcal{L}'}{\partial \mathbf{z}\_{c^*}}(\psi_0) \le \frac{\partial \mathcal{L}'}{\partial \mathbf{z}\_{c^*}}(\psi_1) \le \dots \le \frac{\partial \mathcal{L}'}{\partial \mathbf{z}\_{c^*}}(\psi_l)$ until the budget is exhausted $\Delta \le \sum_{i=0}^{l+1} \Delta_i$, if $\mathcal{L}'$ has the properties **(I)** $\frac{\partial \mathcal{L}'}{\partial \mathbf{z}\_{c^*}} |\_{\psi < 0} = 0$ and **(II)** $\frac{\partial \mathcal{L}'}{\partial \mathbf{z}\_{c^*}} |\_{\psi_0} < \frac{\partial \mathcal{L}'}{\partial \mathbf{z}\_{c^*}} |\_{\psi_1}$ for any $0 \le \psi\_0 < \psi_1$.
> > >
> > > Assumption 2 (as stated in the submitted paper) only needs to hold for a small fraction of nodes with low $\psi\_i$. For the empirical distribution of a two-layer GCN on Cora ML, $\mathbb{E}[\Delta\_i | \psi\_i] = 1$ and $\text{Var}[\Delta\_i | \psi\_i] = 0$ for the $22.9\%$ nodes with lowest margin $\psi\_i$. Hence, $\max\_{\tilde{\mathbf{A}}\text{ s.t. }\|\tilde{\mathbf{A}} - \mathbf{A}\|\_0 \le \Delta} \mathbb{E}[\mathcal{L}\_{0/1}(f\_{\theta}(\tilde{\mathbf{A}}, \mathbf{X}))] \approx \max\_{\tilde{\mathbf{A}}\text{ s.t. }\|\tilde{\mathbf{A}} - \mathbf{A}\|\_0 \le \Delta} \mathcal{L}\_{0/1}(f\_{\theta}(\tilde{\mathbf{A}}, \mathbf{X}))$ for small $\Delta$ (i.e. as stated in the submitted paper). We will make this clear in the revised paper.

---

> ### Author Response · Authors · 2021-08-10
> **Author Response to Reviewer hJTe part 1: Weakness**
>
> We thank you for the level of detail and particularly the thoughts on the surrogate losses. In our response, we focus 1) on highlighting the importance of white-box attacks *next* to other threat models. 2) we discuss the validity of Assumption 2. Then we follow up on the remaining points/questions.
>
> ## W) Weakness
>
> ### W1) White-box attacks are practical
>
> We study adversarial robustness from the perspective of a defender, not as an adversary who attacks e.g. an application for their benefit. From this perspective, we identified three main reasons to study white-box attacks on GNNs at scale (besides the novelty): (1) white-box attacks are the most powerful threat model, (2) we can use white-box attacks to enhance the understanding of the stability of a model, and (3) strong attacks are required to assess the efficacy of defenses.
>
> (1) If we can show that an algorithm is robust w.r.t. to *strong* white-box attacks (the worst case), then we can also assume robustness w.r.t. less powerful threat models (i.e. the white box threat model is strictly the most powerful). Note that a (strong) attack provides an upper bound on the robustness of a GNN. Until *exact certification* is solved (i.e. no/small relaxation gap, which is presumably NP-hard [A, B, C]), we need these two complementary views. Otherwise, certification just provides a lower bound on the robustness.
>
> (2) A very practical use case for our attack is the *assessment of GNNs* on large-scale graphs. Adversarial attacks also enhance the understanding of how such an algorithm behaves to "worst-case noise". This is a desired property for the company and users in commercial applications. Please note that there are certainly multiple internet-scale applications of GNNs (*millions of people are affected by such algorithms in this very moment*, e.g. [37] or [D] where a scientist confirms the Google-internal use of PPRGo at scale). However, prior to this work, it is entirely unknown how robust or rather fragile GNNs on large graphs are. In Figure 8b, we show for PPRGo that already a tiny budget suffices to flip almost every node's prediction!
>
> (3) One of the findings in the image domain is that strong attacks are essential to "pave the way to more robust models" [E]. Even multiple black-box attacks are insufficient to reveal potential non-robustness [E].
>
> Additionally, to assess the efficacy of black-box attacks, it is best to compare with a white-box attack.
>
> [A] - Maksym Andriushchenko, and Matthias Hein. 2020. Provably Robust Boosted Decision Stumps and Trees against Adversarial Attacks. In NeurIPS 2019.
>
> [B] - Huan Zhang, Tsui-Wei Weng, Pin-Yu Chen, Cho-Jui Hsieh, and Luca Danie. 2020. Efficient Neural Network Robustness Certification with General Activation Functions. In NeurIPS 2018.
>
> [C] - Tsui-Wei Weng, Huan Zhang, Hongge Chen, Zhao Song, Cho-Jui Hsieh, D. Boning, I. Dhillon, and L. Daniel. 2020. Towards Fast Computation of Certified Robustness for ReLU Networks. In ICML 2018. 1633-1645.
>
> [D] - Amol Kapoor. 2020. [Workshop - Mining and Learning with Graphs at Scale: Graph Neural Networks](https://gm-neurips-2020.github.io/). NeurIPS Workshop 2020.
>
> [E] - Florian Tramèr, Nicholas Carlini, and Wieland Brendel. 2020. On Adaptive Attacks to Adversarial Example Defenses. In NeurIPS 2020. 1633-1645.
>
> ### W2) Justification of Assumption 2.
>
> Intuitively, Assumption 2 states that the budget $\Delta$ required to successfully change the prediction of a node depends on its classification margin $\psi$. Recall that our setup targets large graphs and comparatively small budgets (lines 106-107, $\frac{\Delta}{n} \to 0$). In this case, the key job for an attacker is to spot "fragile nodes". Since $\Delta$ is small, Assumption 2 only needs to hold for nodes with a low positive classification margin $\psi$ and these nodes need to be at least as fragile as the remaining nodes. On Cora ML and a two-layer GCN, Assumption 2 holds exactly for the 22.9\% of nodes with lowest margin. That is, the 22.9% of nodes with lowest margin $\psi$ all require the lowest possible budget of one. For $\psi>22.9\\%$, we still observe a positive relationship between $\psi$ and $\Delta$ (i.e. the assumption holds in expectation).
>
> We also choose the classification margin as the sole input of the loss for three further reasons: (1) the classification margin $\psi$ implicitly depends on other influences. For example, a node with many similar neighbors (homophily) has likely a large margin $\psi$ and is presumably harder to perturb. (2) in the actual first-order optimization attack (as in §3), other influences (e.g. node features or the neighborhood) are automatically taken into account. However, (3) these simplifying assumptions allow a crisp and intuitive analysis of the loss.
>
> We will revise the motivation is §2 to enhance the reasoning why the assumptions are reasonable for the study of surrogate losses on large graphs with small budgets. Please let us know if our response does not fully address your points. In case you find it helpful, we can also provide an example where Assumption 2 holds exactly for the entire graph.
>
>
> ### W3) Table with theoretical complexities
>
> In the following, we summarize all approaches we use in our experiments. We assume that the number of features and hidden neurons is negligible. $k$ denotes the GDC/PPRGo hyperparameter for top-$k$-sparsification of the PPR matrix, $n$ is the number of nodes, and $m$ is the number of edges. We try to keep the overview simple and e.g. only list the most important hyperparameters. If a model preprocesses the adjacency matrix, we report the time complexities for preprocessing and GNN separately. For the attacks, we report the *additional* complexity (i.e. GNN excluded). We also use $b$ for the block size, $\Delta$ as the budget, and $E$ for the number of epochs. We chose $b=\mathcal{O}(m)$ for global attacks and $b=\mathcal{O}(1)$ for local attacks.
>
> | Architecture             | Memory Complexity          | Time Complexity (all nodes)                 |
> |--------------------------|----------------------------|---------------------------------------------|
> | Soft Median GDC (ours)   | $\mathcal{O}(k \cdot n)$   | $\mathcal{O}(n) + \mathcal{O}(k \cdot n)$   |
> | Soft Median PPRGo (ours) | $\mathcal{O}(k)$           | $\mathcal{O}(n) + \mathcal{O}(k \cdot n)$   |
> | Vanilla GCN              | $\mathcal{O}(m)$           | $\mathcal{O}(m)$                            |
> | Vanilla GDC              | $\mathcal{O}(k \cdot n)$   | $\mathcal{O}(n) + \mathcal{O}(k \cdot n)$   |
> | Vanilla PPRGo            | $\mathcal{O}(k)$           | $\mathcal{O}(n) + \mathcal{O}(k \cdot n)$   |
> | Soft Medoid GDC          | $\mathcal{O}(k^2 \cdot n)$ | $\mathcal{O}(n) + \mathcal{O}(k^2 \cdot n)$ |
> | SVD GCN                  | $\mathcal{O}(n^2)$         | $\mathcal{O}(m) + \mathcal{O}(n^2)$         |
> | Jaccard GCN              | $\mathcal{O}(m)$           | $\mathcal{O}(n^2) + \mathcal{O}(m)$         |
> | RGCN                     | $\mathcal{O}(m)$           | $\mathcal{O}(m)$                            |
>
>
>
> | Global Attack | Memory Complexity     | Time Complexity                  | Details                  |
> |---------------|-----------------------|----------------------------------|--------------------------|
> | PR-BCD (ours) | $\mathcal{O}(b)$      | $\mathcal{O}(E \cdot b \log(b))$ | $b \ge \Delta$           |
> | GR-BCD (ours) | $\mathcal{O}(b)$      | $\mathcal{O}(E \cdot b)$ | $b \ge \frac{\Delta}{E}$ |
> | FGSM          | $\mathcal{O}(n^2)$    | $\mathcal{O}(\Delta \cdot n^2)$  |                          |
> | PGD           | $\mathcal{O}(n^2)$    | $\mathcal{O}(E \cdot n^2)$       |                          |
> | DICE          | $\mathcal{O}(\Delta)$ | $\mathcal{O}(\Delta)$            |                          |
>
>
>
> | Local Attack  | Memory Complexity        | Time Complexity                          |
> |---------------|--------------------------|------------------------------------------|
> | PR-BCD (ours) | $\mathcal{O}(b \cdot k)$ | $\mathcal{O}(E \cdot b \log(b) \cdot k)$ |
> | Nettack       | $\mathcal{O}(\Delta)$    | $\mathcal{O}(\Delta \cdot n)$            |
> | DICE          | $\mathcal{O}(\Delta)$    | $\mathcal{O}(\Delta)$                    |

---

### Official Review · Reviewer_P3zQ · 2021-07-17

**Rating:** 6
**Confidence:** 3

**Summary:**

The contributions of the paper:

1) study the drawbacks of cross-entropy loss in the global attack of GNNs and propose alternative losses;

2) propose a scalable attacking algorithm based on the randomized block gradient descent algorithm. It also introduces a heuristic to enforce sparse adjacent matrix and avoid the computation on dense matrix;

3) propose a more efficient defense method based on an existing method, Soft Medoid.



**Main Review:**

Strength:
1. The major focus of this paper is to study the adversarial robustness of GNNs on a larger scale. This is important since most of the existing attack and defense methods for GNNs are limited to very small data.

2. The instigation and discussion on the ineffective of CE loss in the global attack of GNNs are interesting. It points out that attacking methods based on CE loss might waste attacking budgets, which provides useful insight for the development of stronger attacking methods.

3. The proposed scalable attacking method seems quite efficient and experiments support its effective and efficiency.

Weakness:

1. The author doesn't provide a sufficient explanation for the proposed tanh Margin loss so that this loss design is unclear. Moreover, it is unclear why PGD algorithm doesn't improve over CE. The explanation due to the learning dynamics of PGD and independence assumption  (line 129-133) are confusing and vague. It is better to make it more precise and clear.

2. The proposed attacking algorithm is a bit heuristic and doesn't provide any theoretical justification.

3. It is hard to understand the scalable defense in Section 4. The proposed Soft Median method is based on Soft Medoid but the paper doesn't provide sufficient detail to demonstrate the idea. Therefore, it is unclear how the proposed method improves the efficiency of Soft Medoid. In particular, note that the dimension-wise median has been explored and showed to be worse than Soft Medoid in Soft Medoid paper, which should be discussed. The idea of the differentiable relaxation of the median is unclear.

4. Some important related works are missing. For example, it is stated that the authors are not aware of any defense that scales to graphs significantly larger than PubMed. However, the defense on datasets such as obgn-arxiv and DP has been explored in the paper "GNNGUARD: Defending Graph Neural Networks against Adversarial Attacks", which needs to be discussed and even compared.

Overall, the paper provides insightful investigations on the attacking and defense of GNNs. The proposed methods push the research in this area to a much larger scale, which is a very important direction. However, the writing can be significantly improved to make every part more precise and clear.

**Time Spent Reviewing:**

5

---

> ### Author Response · Authors · 2021-08-10
> **Author Response to Reviewer P3zQ**
>
> We thank you for your constructive feedback and the specific pointers on how to improve the precision and clarity of our work. We now address your questions in the order you list them.
>
> ## 1) Surrogate Loss
>
> ### 1.1) Explanation of tanh margin loss
>
> In §2, we motivate the properties in Definition 2. Unfortunately, there is still an infinite number of functions obeying Definition 2 and there is no obvious best choice. Hence, we compared suitable functions that are common in the deep learning community. Following the principle of Occam’s Razor, we then chose tanh margin because it is common and hyperparameter-free. We also experimented with more complex alternatives. They did not provide benefits when using low budgets and hence we omitted them to avoid confusion. We argue that choosing one loss function for adversarial attacks comes with the issue that there are many plausible choices. Similarly, to the influential paper by Carlini and Wagner [4], we make the final choice empirically. We finally report the seven most interesting surrogate losses with varying properties (see Table 4). For a better overview of the losses, we plan to move Table 4 to §2.
>
>
> ### 1.2) Why does MCE not work well with PGD?
>
> The main reason is the zero gradient if classification margin $\psi < 0$ in the MCE (and CW) loss: $\text{CW} = \min(\max\_{c \ne c^\ast} \mathbf{z}\_{c} - \mathbf{z}\_{c^\ast}, 0)$ [4, 36]. To fully explain the reasons we need to dive into the PGD update step in epoch $t$:
>
> $\mathbf{p}\_{t} = \Pi\_{\mathbb{E}[\text{Bernoulli}(\mathbf{p}\_t)] \le \Delta} \left[ \mathbf{p}\_{t-1} + \alpha\_{t-1} \nabla \mathcal{L}(\mathbf{p}\_{t-1}, \dots) \right]\\,.$
>
> We can rewrite this expression to
>
> $\mathbf{p}\_{t} = \Pi\_{[0, 1]}  ( \mathbf{p}\_{t-1} \\,\\, \underbrace{+ \alpha\_{t} \nabla \mathcal{L}(\mathbf{p}\_{t-1}, \dots)}\_{\text{gradient update}} \underbrace{- \eta\_t \textbf{1}}\_{\text{correction}} )$
>
> where $\Pi_{[0, 1]}$ clamps the values to the range $[0, 1]$ ($\Rightarrow \mathbf{p}\_{t} \in [0,1]^b$) and $\eta_t$ is chosen s.t. $\mathbb{E}[\text{Bernoulli}(\mathbf{p}\_t)] \le \Delta$ (i.e. $\sum \Pi\_{[0, 1]}(\mathbf{p}\_t) \le \Delta$). There are two competing terms: 1) the gradient update $\alpha\_{t-1} \nabla \mathcal{L}(\mathbf{p}\_{t-1}, \dots)$ and 2) the correction $\eta\_t \textbf{1}$ (typically lowers all weights in $\mathbf{p}\_{t}$). For reasonable parameter choices, the potential perturbations in $\mathbf{p}\_{t}$ are competing since our budget is limited (i.e. to maximize the loss we would like to flip more edges than budget we have). Then, after some epochs ($t > t_0$), we will have $\eta_t > 0$ and subtract $\eta_t$ from each element in $\mathbf{p}_{t-1}$.
>
> Now if we choose a loss $\mathcal{L}$ (e.g. CW or MCE loss) that has zero gradient, as soon as a node $v$ is misclassified ($\psi_v < 0$), the responsible edge(s) will not benefit from a "gradient update" anymore but $\eta_t > 0$ is still subtracted. So after some iterations node $v$ will be again correctly classified since the required edge flips in $\mathbf{p}_t$ lost weight/strength. This leads to instability.
>
> The symptoms are particularly visible in Figure 9e for the CW loss (after $t_0 = 25$ epochs the accuracy oscillates around 0.7). Moreover, the accuracy for the CW loss (Figure 9 d-f and Figure 10 d-f) are noisier than for the CE or tanh margin losses (other subfigures).
>
> In the revision, we will provide a figure with the loss/training curves for MCE in addition to the ones with CW. Moreover, we will include this discussion about the PGD update procedure.
>
> ## 2) "The proposed attacking algorithm is a bit heuristic and doesn't provide any theoretical justification."
>
> Solving the combinatorial non-convex optimization problem (Eq. 1) is computationally very expensive. Even the continuously relaxed optimization problem is likely NP-hard (e.g. as stated in [A, B, C] for slightly different constraints). Hence, for an efficient and scalable attack, we need a practical algorithm that provides an approximate solution. Nevertheless, our attacks are plausibly rooted in the theory of first-order optimization as most other attacks (examples in the graph domain [33, 36]).
>
> As we discuss in lines 159-165, (Projected) Randomized Block Coordinate Descent is well studied in the field of large scale optimization and e.g. convergence guarantees have been derived [24, 25, 32]. We deviate from this literature solely in two facts (lines 189-195): 1) we resample at least half of the current block in each epoch and 2) we keep the non-zero edges also in the block of the subsequent batch. Without 1) we take the risk of not exploring enough and hence getting stuck in a local optimum. Without 2) we would not be able to obtain such a low worst-case space complexity. Despite the lack of guarantees, such reasonable heuristics (similarly to mini-batching or momentum) are widely accepted for such difficult optimization problems.
>
>
> [A] - Maksym Andriushchenko, and Matthias Hein. 2020. Provably Robust Boosted Decision Stumps and Trees against Adversarial Attacks. In NeurIPS 2019.
>
> [B] - Huan Zhang, Tsui-Wei Weng, Pin-Yu Chen, Cho-Jui Hsieh, and Luca Danie. 2020. Efficient Neural Network Robustness Certification with General Activation Functions. In NeurIPS 2018.
>
> [C] - Tsui-Wei Weng, Huan Zhang, Hongge Chen, Zhao Song, Cho-Jui Hsieh, D. Boning, I. Dhillon, and L. Daniel. 2020. Towards Fast Computation of Certified Robustness for ReLU Networks. In ICML 2018. 1633-1645.
>
> ## 3) Soft Median Defense
>
> ### 3.1) Explanation of Soft Median
>
> The Soft Median is a continuous relaxation of the optimization problem $\arg \min _{\mathbf{x}^{\prime} \in \mathbb{X}}\left\|\overline{\mathbf{x}}-\mathbf{x}^{\prime}\right\|$. We replace the argmin with the softmax and negated objective. This is inspired by the recent soft sorting technique [26], with a small simplification since we only need the "soft" sorting w.r.t. central element $\overline{\mathbf{x}}$. Hence, the Soft Median weights the inputs based on the distance to the dimension-wise median $\overline{\mathbf{x}}$. For $T \to 0$ it returns the instance that is closest to the dimension-wise median (see lines 267-268). Our Soft Median has an asymptotic complexity of $\mathcal{O}(k)$ with the number of inputs $k = |u \in \mathbb{N}(v) \cup v|$.
>
> In contrast, the Soft Medoid comes with quadratic complexity $\mathcal{O}(k^2)$ (lines 255-258). The key take-way is that they require the distance between every possible pair of nodes in the neighborhood of $v$: $\\{ \left\\|\mathbf{x}\_{i}-\mathbf{x}\_{j}\right\\|  \\, | \\, i,j \in \mathbb{N}(v) \cup v \\}$ (i.e. the distance matrix).
>
> ### 3.2) Dimension-wise median has been explored in [14]
>
> [14] does not compare Soft Median vs. Soft Medoid. They only compare to the "hard" dimension-wise median but find it not as strong as their Soft Medoid (see Table 2 in [14]). Similar to the finding in [14], we observe our Soft Median comes with a lower error if facing perturbed inputs (see tables below which reproduce and complement Figure 2 in [14]).
>
> The subsequent tables show the error $\|t(\mathbf{X}) - t(\tilde{\mathbf{X}})\|\_2$, for 50 samples from a centered ($t\_{\text{SM}}(\mathbf{X}) = 0$) bivariate normal distribution. The adversary is a point mass perturbation on the first axis over increasing fraction of outlier $\epsilon$.
>
> Table A: This table shows the bias for a perturbation with norm 1000.
>
> | Estimator            |   $\epsilon = 0.1$ |   $\epsilon = 0.2$ |   $\epsilon = 0.3$ |   $\epsilon = 0.4$ |
> |:---------------------|-------------------:|-------------------:|-------------------:|-------------------:|
> | Soft Median ($T=1$)  |         **0.11**   |           **0.16** |           **0.22** |           **0.29** |
> | Sample mean          |         100.00     |         200.00     |         300.00     |         400.00     |
> | dimension-wise median|           0.26     |           0.45     |           0.66     |           1.02     |
> | Soft Medoid ($T=50$) |           0.13     |           0.22     |           0.32     |           0.43     |
>
>
> Table B: This table shows the bias for a perturbation with norm 10.
>
> | Estimator            |   $\epsilon = 0.1$ |   $\epsilon = 0.2$ |   $\epsilon = 0.3$ |   $\epsilon = 0.4$ |
> |:---------------------|-------------------:|-------------------:|-------------------:|-------------------:|
> | Soft Median ($T=1$)  |           **0.12** |           **0.20** |           **0.31** |           **0.51** |
> | Sample mean          |               1.00 |               2.00 |               3.00 |               4.00 |
> | dimension-wise median|               0.27 |               0.45 |               0.66 |               1.02 |
> | Soft Medoid ($T=50$) |               0.13 |               0.26 |               0.74 |               3.30 |
>
>
> ## 4) Related work: GNNguard only uses subset of obgn-arxiv
>
> The GNNguard authors (cited with number [38]) only use a subset of arXiv covering 20\% of the nodes and only 6\% of the edges (papers published between 1971 and 2014). Moreover, in terms of number of nodes, the DP dataset is of similar size as PubMed (22.6k nodes vs. 19.7k nodes). We did not state those insignificant differences but will change this. Moreover, a comparison to theirs is not suitable since they study poisoning (train time) attacks, while we study evasion (test time) attacks (see lines 36-37).

---

> > ### Comment · Reviewer_P3zQ · 2021-08-23
> > **Discussion**
> >
> > Thank the authors for their detailed responses to my comments. The responses clarify some of my questions, although some other concerns (such as being heuristic and lack of theoretical guarantees) can not be easily fixed.
> >
> > Overall, I find the paper interesting, and the proposed algorithms seem to be reasonable and practically useful. I would like to suggest the authors carefully improve the writing and make the algorithm details clear.

---

### Decision · Program_Chairs · 2021-09-27

**Decision:**

Accept (Poster)

**Comment:**

This paper tackles the adversarial robustness of large-scale GNNs, which is practically important but has been overlooked. The authors provide a novel observation that the conventional cross-entropy (and Carlini-Wagner) loss guides the attacker to attack nodes that are already misclassified when attacking large GNNs, which results in the waste of the computing budget. Then, the authors show that under certain assumptions, a budget-aware loss can achieve the optimal solution, and propose a realization of such surrogate loss, Masked Cross Entropy (MCE), which only considers correctly classified samples, and thus is more efficient. The authors further propose scalable attacks and a scalable defense mechanism which modifies the aggregation function in the message passing framework. The experimental validation of the proposed attacks and the defenses show that they are indeed effective, and efficient in terms of computation and memory cost.

The following are the pros and cons of the paper mentioned in the initial reviews.

Pros
- The tackled problem of ensuring robustness of a large-scale GNNs against adversarial attacks, is an important yet unexplored problem.
- The study of the drawbacks of the cross-entropy loss when attacking GNNs is both novel and interesting.
- The proposed surrogate loss, Masked Cross Entropy, is both novel and effective for attacking large-scale GNNs, and has a potential to be further explored for other adversarial defense problems.
- The proposed attack and the defense methods are efficient and scalable, which makes the model practical.
- The paper is well-written and is accompanied with well-structured source code.

Cons

- The proposed attack and the defense methods are heuristic and lack theoretical justification.
- The assumption of white-box attack for large-scale GNNs is impractical.
- The assumption on the expected budget required to ensure the optimality of the solution with the budget-aware loss is unrealistic.
- The effectiveness of the proposed attacks are shown on a small-scale graph.
- Lack of discussion and experimental comparison against relevant baselines, such as Li et al. 20, which also considers defense against adversarial attacks on large graphs.
- Missing memory and time complexities of the different types of attacks.

While the initial reviews were a bit split, during the discussion period, the reviewers and the authors actively engaged in very thorough, constructive discussions. This cleared away many of the concerns from the reviewers, and the reviewers unanimously agreed to accept the paper at the end. The authors convinced the reviewers that the assumption of white-box attack is reasonable in the defender’s point of view, and acknowledged the reviewers’ arguments that the black-box attack could be more practical when considering the tradeoff between clean and robust accuracy. The authors provided a more detailed explanation of the assumption, asymptotic time and memory complexities, and detailed discussion and experimental comparison against Li et al. 20. The lack of strict theoretical guarantees for the proposed attacks and the defenses are not critical, as their effectiveness have been verified with extensive empirical analysis.

In sum, this is a strong paper that tackles a novel and practical problem, analyzes the problem with an existing approach, and proposes an effective solution backed up with both theoretical and empirical analysis, which makes it a clear accept. I praise the authors and the reviewers for their constructive and active discussions, and advise the authors to incorporate them into the revision, as well as the new results they provided in the responses.